# Entropy Meets Importance: A Unified Head Importance–Entropy Score for Stable and Efficient Transformer Pruning

## Abstract

Transformer-based models have achieved remarkable performance in NLP tasks. However, their structural characteristics—multiple layers and attention heads—introduce efficiency challenges in inference and deployment. To address these challenges, various pruning methods have recently been proposed. Notably, gradient-based methods using Head Importance Scores (HIS) have gained traction for interpretability, efficiency, and ability to identify redundant heads. However, HIS alone has limitations as it captures only the gradient-driven contribution, overlooking the diversity of attention patterns. To overcome these limitations, we introduce a novel pruning criterion, **HIES (Head Importance-Entropy Score)**, which integrates head importance scores with attention entropy, providing complementary evidence on per-head contribution. Empirically, HIES-based pruning yields up to 15.2% improvement in model quality and $2.04\times$ improvement in stability over HIS-only methods, enabling substantial model compression without sacrificing either accuracy or stability. Code will be released upon publication.

## 1 Introduction

Recent advances in Large Language Models (LLMs) have led to remarkable performance. In pursuit of better modeling of long-range dependencies, LLMs have scaled up both context lengths and attention head counts, guided by empirical scaling laws that correlate model capacity with performance (Kaplan et al., 2020; Chen et al., 2025b). This scaling, however, incurs substantial computational and memory costs during inference, resulting in prohibitive latency and energy consumption (Yang et al., 2020; Kim and Wu, 2020; Hoefler et al., 2021; Zhou et al., 2024). These constraints become critical barriers when LLMs are deployed to resource-constrained environments such as consumer-grade mobile devices or edge devices, for applications including real-time translation, intelligent voice assistants, and personalized recommendation systems.

To improve deployability of LLMs for resource-constrained environments, various pruning methods have been proposed (Ma et al., 2023; Yang et al., 2024). Typically, these methods selectively reduce computations by removing less important weights, channels, or attention heads. Among them, *head pruning* has gained considerable attention due to its structural simplicity, interpretability, and ability to directly target redundancy within the attention mechanism. Existing head pruning methods typically identifies less important heads based on Head Importance Score (HIS), which quantifies the gradient-based contribution of each head to the loss function. By leveraging gradient-based sensitivity to the loss, HIS prioritizes heads that have the most direct impact on accuracy of model inference.

However, HIS-based methods often exhibit limited stability in their performance. For clarity, prior works (Bair et al., 2024; Blanchet et al., 2024) motivate treating stability as a practical surrogate for robustness—namely, a model's resilience to input perturbations and pruning-induced distributional shifts. Such stability is crucial in real-world deployments where distribution shifts are common and aggressive compression is often required. In our observations, this instability appears to stem from two key factors. First, existing HIS-based methods solely rely on the loss gradient with respect to each head's output, which fails to capture token-level attention allocation or its alignment with the task's empirical distribution. Consequently, a concentrated head and a diffuse head can yield similar important scores, concealing their functionally distinct roles on the task-specific data manifold.

Second, a uniform, layer-agnostic criterion precludes layer-specific adaptation despite evidence that different layers require distinct attention behaviors (Artzy and Schwartz, 2024). Lacking such layer-specific characteristics often results in imbalanced pruning—preserving redundant heads in some layers while removing functionally critical ones in others. This imbalance not only degrades accuracy but also undermines stability, leading to unpredictable performance fluctuations across inputs or compression levels, particularly under aggressive pruning ratios.

This work aims to address the aforementioned limitations by proposing an *Entropy-Aware Pruning Criterion*, termed **HIES (Head Importance-Entropy Score)**, which jointly considers a head's gradient-based contribution to the loss and the distributional structure of its attention—specifically, the extent to which its attention is concentrated or dispersed across input tokens. We compute the HIS to quantify a head's loss relevance and Attention Entropy (AE) to measure how evenly a head distributes attention over input tokens. Their principled combination in HIES enables layer-adaptive pruning decisions and preserves functionally important heads. This allows for more balanced pruning across layers, improving both accuracy and stability under aggressive compression. Empirically, HIES yields up to a 15.2% improvement in model quality and $2.04\times$ improvement in stability over HIS-only methods. By preserving both accuracy and stability even under aggressive pruning ratios, HIES represents a more practical and robust solution compared to existing pruning methods. It is expected to offer more stable performance in resource-constrained environments.

## 2  BACKGROUND

**Attention head pruning.** To compress large language models efficiently, structured pruning methods (Han et al., 2015; Wang et al., 2019; Hou et al., 2020a; Ma et al., 2023; Ashkboos et al., 2024), which remove specific architectural components from Transformer models, have been widely adopted. Among these, attention head pruning has gained traction. This is largely because it directly reduces attention FLOPs and KV-cache memory while preserving the layer topology, thereby simplifying checkpoint compatibility and serving integration. Consequently, large-scale studies adopt head-level pruning as a practical axis in LLM compression pipelines (Jaradat et al., 2024; Muralidharan et al., 2024)[1]. Attention head pruning removes selected heads from a trained Transformer's multi-head attention with minimal impact on end-task performance (Vaswani et al., 2017). A widely adopted criterion is the HIS of Michel et al. (2019), which introduces mask variables $m_h \in \{0, 1\}$ multiplying the output of head $h$ and defines importance as the expected first-order loss increase under masking:

$$\text{HIS}_h = \mathbb{E}_{x \sim \mathcal{D}} \left| \frac{\partial \mathcal{L}(x)}{\partial m_h} \right| = \mathbb{E}_{x \sim \mathcal{D}} \left| \text{A}_h(x)^\top \frac{\partial \mathcal{L}(x)}{\partial \text{A}_h(x)} \right|, \tag{1}$$

where $\mathcal{D}$ denotes an input sample drawn from the data distribution $\mathcal{D}$, $\mathcal{L}(x)$ is the loss for sample $x$, and $\text{A}_h(x)$ is the output of head $h$. The second equality follows from the chain rule and the observation that gating scales the head's activation. Heads are then ranked by $\text{HIS}_h$ and pruned in ascending order of importance.

**Attention Entropy and Stability.** Zhai et al. (2023) quantify the concentration of each attention head's focus over input tokens via the entropy of its attention weight distribution $\text{AE}_h = (H(p^{(h)}) = -\sum_{i=1}^n p_i^{(h)} \log p_i^{(h)}$, where $p_i^{(h)}$ is the normalized attention probability assigned by head $h$ to the $i$-th input token subject to $\sum_{i=1}^n p_i^{(h)} = 1$. Higher entropy indicates a diffuse focus over the sequence, whereas lower entropy corresponds to highly concentrated attention patterns. Their empirical findings reveal a strong correlation between persistently low entropy (i.e., entropy collapse) and instability during training, including oscillations in the loss landscape and even divergence across various model scales and tasks.

## 3  MOTIVATION

Pruning Transformer models is most commonly driven by gradient-based criteria, such as HIS and variants used in recent pruning frameworks (e.g., LLM-Pruner) (Michel et al., 2019; Ma et al., 2023). While gradient-based methods are often effective at moderate sparsity, they exhibit sharp accuracy degradation once the pruning ratio exceeds a certain threshold, as shown in Fig. 1 (a). Such

---

[1]For more detailed discussions on related work, please refer to Appendix A.

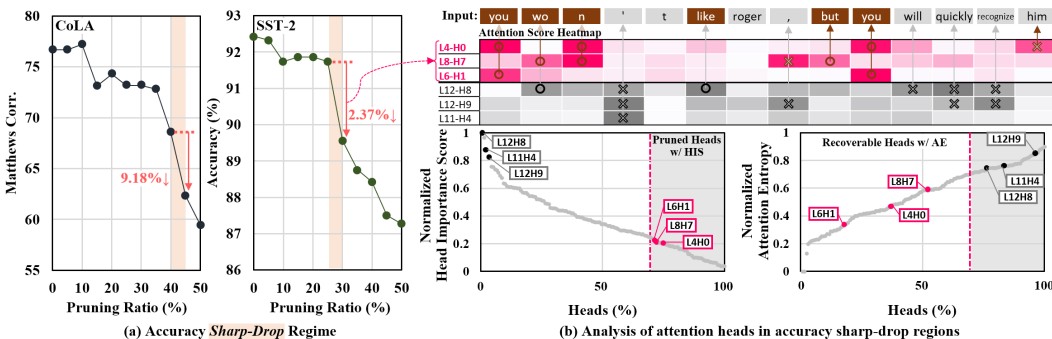

Figure 1: Analysis of accuracy degradation and head behaviors under HIS-based pruning. In our diagnostic study, we analyze the phenomena of pruning by HIS on BERT, focusing on detailed attention head behaviors during inference. (a) Accuracy curves of HIS-based pruning on CoLA and SST-2. The bold color highlights the sharp-drop regime of HIS-based pruning. (b) Head-level analysis on SST-2 with a validation example misclassified by the HIS-pruned model. The attention score heatmap shows heads on each column, where pruned low-HIS heads are indicated with colored layer–head labels and unpruned high-HIS heads with gray table. It then shows token-wise distributions, where tokens deemed important for classification (e.g., sentiment-discriminative tokens in SST-2) are marked with **O**, and non-critical tokens with **X**. The left plot shows the distribution of heads by normalized HIS, and the right plot shows the distribution by normalized AE, where pruned low-HIS heads are highlighted with red boxes and unpruned high-HIS heads with gray boxes. See Appendix C for experimental setup details.

*sharp-drops* have been widely observed across various attention variants and tasks (Ma et al., 2023; Mao et al., 2023; Ghattas et al., 2025), underscoring the generality of this phenomenon.

We focus on this "*sharp-drop*" regime and contrast two groups of heads. The first group consists of low-HIS heads that are pruned during the sharp-drop of accuracy. The second group consists of high-HIS heads that remain unpruned. The attention score heatmap in Fig. 1 (b) reveals that some pruned heads (red table) assign high attention scores—i.e., the weights computed by the softmax over token-token similarity that indicate how strongly a token attends to another—to sentiment-discriminative tokens (tokens relevant for classification). Nonetheless, these heads are pruned due to their low HIS values and end up causing the sharp accuracy drop observed in Fig. 1 (a). In contrast, some unpruned heads (gray table) often allocate strong attention to non-informative tokens. These heads, however, have high HIS values and thus remain unpruned, though they contribute little to overall model quality. These analysis results demonstrate that the gradient-based HIS is insufficient to capture the token-level attention score distributions (the detailed mathematical analysis is provided in Section 4.2.1), thereby resulting in suboptimal pruning decisions for heads focusing on decisive tokens.

The *sharp-drop* observed in pruning can be interpreted as a collapse of structural diversity in attention behaviors, caused by the elimination of heads that concentrate on decisive tokens. To capture and prevent such collapse of structural diversity in attention, we employ attention entropy, a measure widely used to prevent policy collapse in reinforcement learning (Bharadhwaj et al., 2020; Liu et al., 2021a; Wang et al., 2025). As shown in the bottom-right plots of Fig. 1 (b), incorporating AE helps retain low-entropy heads by recognizing their concentrated focus on decisive tokens, thereby preventing them from being pruned and mitigating the *sharp-drop* in accuracy.

Building on our analysis, we posit that attention entropy captures structural signals that reflect unstable behavior during deployment, leading to the following hypothesis:

> *Attention entropy serves as an indicator of inference-time stability, mitigating accuracy sharp-drops*

In particular, low-entropy heads may correlate with increased sensitivity to input perturbations, leading to unstable predictions under distribution shifts. This perspective motivates our investigation of entropy as a proxy for robustness and consistency during inference.

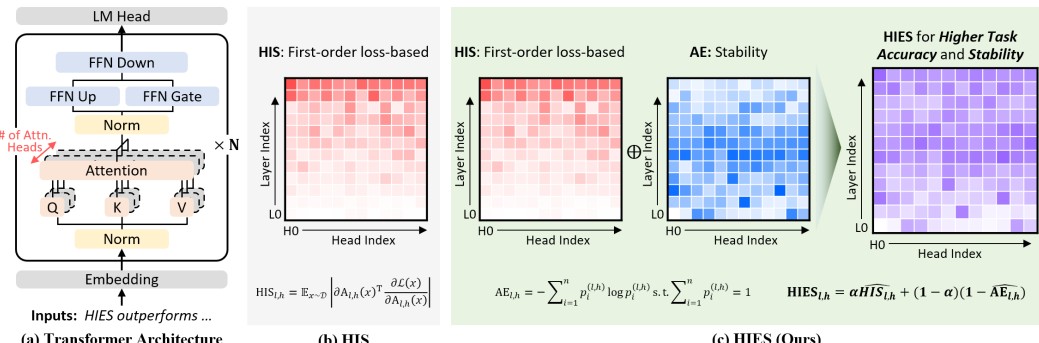

Figure 2: Design overview of Head Importance-Entropy Score (HIES). Darker cells correspond to values closer to 1, while lighter cells correspond to values closer to 0. Note the heatmap utilizes each metric across attention heads in BERT on the CoLA dataset.

## 4 PROPOSED METHOD

Fig. 2 provides an overview of our proposed *Head Importance–Entropy Score (HIES)*. Fig. 2 (a) outlines the Transformer architecture, where HIS computes the importance score of each of the $h$ attention heads within a Transformer block. Fig. 2 (b) displays the heatmap of the first-order, loss-based HIS. Combining the normalized HIS and AE yields the HIES heatmap in Fig. 2 (c), which integrates complementary signals provide a more stable assessment of attention heads across layers. This design captures the key intuition of HIES: HIS can be reinforced by AE, providing a more robust basis for pruning head selections. Section 4.1 formalizes HIES, and Section 4.2 develops a risk-decomposition analysis that clarifies the complementary roles of HIS and AE and motivates HIES as a robust criterion for head selection in pruning.

### 4.1 HEAD IMPORTANCE ENTROPY SCORE

We define the **Head Importance–Entropy Score (HIES)** as a weighted combination:

$$\text{HIES}_h = \alpha \widehat{\text{HIS}}_h + (1 - \alpha)(1 - \widehat{\text{AE}}_h), \quad \alpha \in [0, 1), \tag{1}$$

where $\alpha^2$ is a tunable hyperparameter.

**Min-Max Normalization** Directly comparing raw HIS and AE is inherently problematic, as the two metrics reside on different scales and encode distinct types of signals. To enable meaningful integration and ranking, we apply *min–max normalization* to both metrics, rescaling their values to the interval $[0, 1]$: $\widehat{\text{HIS}}_h = \frac{\text{HIS}_h - \min(\text{HIS})}{\max(\text{HIS}) - \min(\text{HIS})}$, $\widehat{\text{AE}}_h = \frac{\text{AE}_h - \min(\text{AE})}{\max(\text{AE}) - \min(\text{AE})}$. This distribution-agnostic normalization improves cross-criterion interpretability; lower normalized scores denote higher pruning priority. Prior studies show min–max scaling outperforms $z$-score standardization in stability and reproducibility across diverse tasks (de Amorim et al., 2022; Lima and Souza, 2023).

### 4.2 THEORETICAL ANALYSIS

We analyze pruning through a risk decomposition that combines a loss-increase term controlled by HIS, with a generalization-gap term upper-bounded in terms of AE via its token-wise deficit. We further show that the gradients of HIS and AE are orthogonal in expectation, indicating complementary axes: magnitude of contribution (HIS) and dispersion of attention (AE). This perspective motivates the composite importance measure HIES. By retaining heads with high HIES, we simultaneously minimize our theoretical bound and enhance pruning stability. Conceptually, this analysis formalizes importance-based selection into a principled framework and offers a rigorous rationale for HIES's safety and effectiveness.

---

[2]To determine the optimal combination of HIS and AE for each task, we adopt a task-specific tuning procedure, where the trade-off hyperparameter $\alpha$ is systematically explored under each compression setting. Sensitivity to $\alpha$ is analyzed in Appendix D.5.

### 4.2.1 LOSS-INCREASE CONTROL VIA HEAD IMPORTANCE (HIS)

**Setup.** Let $n$ be the sequence length, $|\mathrm{H}|$ the number of heads, $d_v$ the value dimension per head, and $d = |\mathrm{H}| \, d_v$ the model width (i.e., hidden size). For head $h$, let $A_h \in \mathbb{R}^{n \times d_v}$ denote the head output, i.e., the value-projected attention representation $A_h = \mathrm{softmax}\left(\frac{Q_h K_h^\top}{\sqrt{d_k}}\right) V_h$, where $Q_h = X W_h^Q \in \mathbb{R}^{n \times d_k}$, $K_h = X W_h^K \in \mathbb{R}^{n \times d_k}$, and $V_h = X W_h^V \in \mathbb{R}^{n \times d_v}$ are the query, key, and value projections of the input $X \in \mathbb{R}^{n \times d_{\mathrm{model}}}$, with parameter matrices $W_h^Q \in \mathbb{R}^{d_{\mathrm{model}} \times d_k}$, $W_h^K \in \mathbb{R}^{d_{\mathrm{model}} \times d_k}$, and $W_h^V \in \mathbb{R}^{d_{\mathrm{model}} \times d_v}$.

We then define $\mathbf{y} = \mathrm{Concat}(A_1, \ldots, A_{|\mathrm{H}|}) \in \mathbb{R}^{n \times d}$ as the pre-projection representation, which is subsequently projected through $W^O \in \mathbb{R}^{d \times d}$. Head removal is modeled by mask variables $m_h \in \{0, 1\}$: $\delta A_h = -(1 - m_h) A_h$ and $\delta \mathbf{y} = \mathrm{Concat}(\delta A_1, \ldots, \delta A_{|\mathrm{H}|})$. Formal definitions and implementation notes are deferred to Appendix B.1.2.

**Head Importance Score (HIS).** We define

$$\mathrm{HIS}_h := \mathbb{E}_{x \sim \mathcal{D}} \left| \frac{\partial \mathcal{L}(x)}{\partial m_h} \right| = \mathbb{E}_{x \sim \mathcal{D}} \left| \left\langle \nabla_{A_h(x)} \mathcal{L}(x), A_h(x) \right\rangle_F \right| = \mathbb{E}_{\mathcal{D}} \left[ \left| \left\langle \nabla_{A_h} \mathcal{L}, A_h \right\rangle_F \right| \right]. \quad (2)$$

This quantity is a first-order activation–gradient correlation whose absolute value prevents cross-sample cancellation, yielding the additive upper bound $\sum_h \mathrm{HIS}_h$ on the loss (cf. Appendix B.1.6).

**Lemma 1** (Loss-increase upper bound under head masking). *Let $\beta_y := \|\nabla_{\mathbf{y}}^2 \mathcal{L}\|_2$. For any mask variables* $\{m_h\}_{h=1}^{|\mathrm{H}|}$, $\Delta \mathcal{L} := \mathbb{E}_{\mathcal{D}}[\mathcal{L}(\mathbf{y} + \delta \mathbf{y}) - \mathcal{L}(\mathbf{y})] \leq \sum_{h=1}^{|\mathrm{H}|} (1 - m_h) \mathrm{HIS}_h + \frac{\beta_y}{2} \sum_{h=1}^{|\mathrm{H}|} (1 - m_h) \|A_h\|_F^2$. *Moreover, under* binary (sigmoid) cross-entropy *we have* $\|\nabla_{\mathbf{z}}^2 \mathcal{L}\|_2 \leq \frac{1}{4}$; *with the linear projection* $\mathbf{z} = \mathbf{y} W^O$, *this yields* $\beta_y \leq \frac{1}{4} \|W^O\|_2^2$, *hence*

$$\Delta \mathcal{L} \leq \sum_{h=1}^{|\mathrm{H}|} (1 - m_h) \mathrm{HIS}_h + \frac{1}{8} \|W^O\|_2^2 \sum_{h=1}^{|\mathrm{H}|} (1 - m_h) \|A_h\|_F^2. \quad (3)$$

*Remark.* For *multiclass softmax* cross-entropy, $\|\nabla_{\mathbf{z}}^2 \mathcal{L}\|_2 \leq \frac{1}{2}$ (cf. Appendix B.1.4); thus the quadratic coefficient becomes $\frac{1}{4}$ instead of $\frac{1}{8}$.

**Implication for pruning.** Eq. equation 3 shows that, for a fixed pruning fraction $\rho = \frac{1}{|\mathrm{H}|} \sum_h (1 - m_h)$, selecting heads with the smallest $\mathrm{HIS}_h$ minimizes the dominant first-order term, while the quadratic term is controlled by $\|W^O\|_2$ (or blockwise norms) and token-averaged activations. Under standard normalization, the quadratic contribution is typically dominated by the first-order term (cf. Appendix B.1.5), justifying the use of $\mathrm{HIS}_h$ as a practical surrogate importance for head pruning.

However, since $A_h = \mathrm{Attn}_h V_h$, $\mathrm{HIS}_h$ in Eq. equation 2 depends solely on $A_h$ and $\nabla_{A_h} \mathcal{L}$, without explicitly incorporating the distribution $\mathrm{Attn}_h$. Consequently, two heads with very different attention patterns (e.g., sharply focusing on tokens versus spreading over tokens) can yield similar $\mathrm{HIS}_h$ whenever the resulting $A_h$ is comparable. This is supported by our empirical analysis in Section 3.

### 4.2.2 GENERALIZATION GAP AND ATTENTION ENTROPY (AE)

**Setup.** Let $\mathcal{S} = \{(x_i, y_i)\}_{i=1}^N \sim \mathcal{D}$. We write $\mathbb{E}_{\mathcal{S}}$ and $\mathbb{E}_{\mathcal{D}}$ for empirical and population expectations. We assume the per-example loss $\ell$ is $L_\ell$-Lipschitz in its first argument, a standard assumption that yields stability-based generalization bounds (Bousquet and Elisseeff, 2002; Shalev-Shwartz and Ben-David, 2014; Hardt et al., 2016).

**Notation (attention entropy deficit).** For head $h$ and query token $t \in [n(x)]$, let $\boldsymbol{\alpha}_t^{(h)}(x) \in \Delta^{n(x)-1}$ denote the attention over keys and $H(\mathbf{p}) := -\sum_j p_j \log p_j$. We define the token-averaged, length-normalized deficit (AD)

$$\mathrm{AD}_h(x) := \frac{1}{n(x) \log n(x)} \sum_{t=1}^{n(x)} \left( \log n(x) - H(\boldsymbol{\alpha}_t^{(h)}(x)) \right) = 1 - \mathrm{AE}_h(x) \in [0, 1]. \quad (4)$$

**Main bound (loss–entropy link).** Let $M := \max_h \max_j \|V_h(j,:)\|_2$ and $C_{\mathrm{AE}} := \sqrt{8}\, M\, \sqrt{|\mathrm{H}|\rho \log n}$ for a representative effective length $n$. For the pruned model $f_{\mathcal{S},m}$, the expected generalization gap ($\mathcal{G}$) satisfies

$$\mathcal{G} := \mathbb{E}_{\mathcal{D}}\big[\ell(f_{\mathcal{S},m}(x), y)\big] \; - \; \mathbb{E}_{\mathcal{S}}\big[\ell(f_{\mathcal{S},m}(x), y)\big]$$

$$\leq \; 2L_\ell\, C_{\mathrm{AE}} \sqrt{\sum_{h=1}^{|\mathrm{H}|} (1 - m_h)\, \mathbb{E}_{\mathcal{S}}\big[\overline{\mathrm{AD}}_h(x)\big]} \; + \; \frac{B}{N}\,. \tag{5}$$

**Interpretation.** For a fixed pruning ratio $\rho$, pruning heads with smaller deficit (i.e., higher entropy) minimizes the bound's increase; pruning low-entropy (high-deficit) heads worsens it. Proof details, operator-norm assumptions, and variable-length handling are deferred to Appendix B.2.

### 4.2.3 Risk Upper Bound and HIES Minimization

**Composite Risk Bound.** Given the HIES defined above, the overall risk upper bound is

$$\mathcal{R}(m) := \sum_{h=1}^{|\mathrm{H}|} (1 - m_h)\, \mathrm{HIES}_h \tag{6}$$

**Pruning Objective (fixed budget).** Let $k := (1 - \rho)|\mathrm{H}|$ be the number of heads to retain. We solve the cardinality-constrained selection problem

$$\min_{m \in \{0,1\}^{|\mathrm{H}|}} \sum_{h=1}^{|\mathrm{H}|} (1 - m_h)\, \mathrm{HIES}_h \quad \text{s.t.} \sum_{h=1}^{|\mathrm{H}|} m_h = k. \tag{7}$$

**Lemma 2** (Optimality). *Selecting the $k$ heads with the largest HIES values (equivalently, pruning the $|H| - k$ heads with the smallest HIES values) yields the globally optimal mask $m^*$ that minimizes equation 6 subject to equation 7.*

### 4.2.4 Orthogonality and Complementarity

**Lemma 3** (Orthogonality). *Let*

$$u_h := \mathrm{sign}\big(\boldsymbol{\alpha}^{(h)\top} g_h\big)\, g_h, \qquad v_h := \mathbf{1} + \log \boldsymbol{\alpha}^{(h)}, \qquad \tilde{u}_h := P\, u_h, \;\; \tilde{v}_h := P\, v_h,$$

*where $P := I - \frac{1}{n}\mathbf{1}\mathbf{1}^\top$ projects onto $\{w : \mathbf{1}^\top w = 0\}$. Assume $\mathrm{Cov}(\tilde{u}_h, \tilde{v}_h) = 0$ (the cross-covariance matrix is zero). If, in addition, either $\mathbb{E}_{\mathcal{S}}[\tilde{u}_h] = 0$ or more generally $\langle \mathbb{E}_{\mathcal{S}}[\tilde{u}_h], \mathbb{E}_{\mathcal{S}}[\tilde{v}_h] \rangle = 0$, then*

$$\mathbb{E}_{\mathcal{S}}\Big[\big\langle \widetilde{\nabla}_{\boldsymbol{\alpha}^{(h)}} \mathrm{HIS}_h,\; \widetilde{\nabla}_{\boldsymbol{\alpha}^{(h)}} \mathrm{AE}_h \big\rangle\Big] \; = \; 0, \tag{8}$$

*i.e., the two gradient directions are orthogonal in expectation.*[3]

**Complementarity.** Because the gradients point along statistically orthogonal directions, HIS captures the magnitude of loss sensitivity whereas AE captures the dispersion of attention. Thus, they serve complementary roles: HIS emphasizes the magnitude of contribution, while AE characterizes distributional concentration. Combined, they balance pruning minimally influential heads and preserving heads important for generalization, underpinning HIES's effectiveness.

## 5 Experimental Results

### 5.1 Experimental Setup

**Model.** We use publicly available BERT$_{\mathrm{base}}$ checkpoints that have been fine-tuned and released by prior work (Devlin et al., 2019), and LLaMA-2$_{\mathrm{7B}}$ checkpoint from Hugging Face (Meta AI, 2023). To examine the generalizability to attention variants and tasks, we further employ ViT$_{\mathrm{Large}}$ (Dosovitskiy et al., 2020) and LLaVA-1.5$_{\mathrm{7B}}$ (Liu et al., 2023).

---

[3]Detailed derivations and preliminaries are deferred to Appendix B.4.

Table 1: Experimental results with BERT_base on natural language understanding task. We report percentage improvements in blue.

| Pruning Ratio | Method | SST-2 Accuracy | CoLA Matthews corr | MRPC F1 Score | QQP Accuracy | STS-B Pearson corr | QNLI Accuracy | MNLI Accuracy | RTE Accuracy | Average | |
|---|---|---|---|---|---|---|---|---|---|---|---|
| 0% | BERT_base | 92.43 | 76.69 | 91.35 | 91.27 | 94.02 | 91.54 | 84.57 | 72.56 | 86.80 | |
| | Random | 92.09 | 75.25 | 89.90 | 91.00 | 93.88 | 89.87 | 82.62 | 70.76 | 85.67 | 0.00% |
| | AD | 92.55 | 75.48 | 90.56 | 91.10 | 93.65 | 90.87 | 83.50 | 70.40 | 86.01 | +0.40% |
| | HIS | 91.74 | **77.21** | 90.65 | 91.23 | **94.04** | 90.63 | **84.04** | 71.84 | 86.42 | +0.88% |
| | L2 | 90.37 | 74.00 | 83.78 | 69.23 | 83.12 | 60.81 | 67.75 | 49.82 | 72.36 | -15.54% |
| 10% | LLM-Pruner (Channel) | 91.97 | 76.05 | 79.19 | 83.53 | 93.06 | 87.39 | 80.46 | 67.51 | 82.40 | -3.82% |
| | LLM-Pruner (Block) | 91.06 | 76.69 | 89.83 | 84.96 | 93.77 | 87.42 | 82.03 | 64.62 | 83.80 | -2.19% |
| | SliceGPT (w/o tune) | 51.38 | 49.86 | 81.46 | 63.18 | 58.92 | 53.91 | 36.84 | 49.46 | 55.63 | -35.07% |
| | SliceGPT (w/ tune) | 86.47 | 61.87 | 82.30 | 88.28 | 62.47 | 83.45 | 77.34 | 54.51 | 74.59 | -12.94% |
| | **HIES (ours)** | **92.66** | 75.48 | **91.04** | **91.93** | 94.00 | **91.03** | **84.04** | 71.84 | **86.50** | **+0.97%** |
| | Random | 90.29 | 69.02 | 85.27 | 84.00 | 92.90 | 79.80 | 75.15 | 60.29 | 79.59 | 0.00% |
| | AD | 86.58 | 50.00 | 84.53 | 84.57 | 81.94 | 67.82 | 77.00 | 56.68 | 73.64 | -7.47% |
| | HIS | 89.56 | 73.17 | **89.37** | 89.95 | 93.82 | 89.04 | 82.25 | 68.59 | 84.47 | +6.13% |
| | L2 | 86.58 | 67.52 | 81.58 | 64.83 | 76.52 | 51.09 | 56.00 | 50.90 | 66.75 | -16.10% |
| 30% | LLM-Pruner (Channel) | 88.53 | 70.36 | 86.89 | 81.23 | 92.50 | 67.38 | 66.67 | 64.26 | 77.23 | -2.97% |
| | LLM-Pruner (Block) | 88.99 | 73.93 | 84.76 | 80.09 | 93.40 | 82.68 | 78.33 | 66.79 | 81.12 | +1.92% |
| | SliceGPT (w/o tune) | 50.80 | 53.29 | 78.05 | 63.18 | 54.64 | 53.18 | 34.90 | 51.99 | 55.00 | -30.90% |
| | SliceGPT (w/ tune) | 83.49 | 60.14 | 81.80 | 85.80 | 60.89 | 67.36 | 75.50 | 54.87 | 71.23 | -10.49% |
| | **HIES (ours)** | **91.86** | **74.97** | 88.81 | **90.37** | **93.89** | **89.13** | **82.50** | **70.04** | **85.20** | **+7.04%** |
| | Random | 78.74 | 61.02 | 72.53 | 66.25 | 91.40 | 67.53 | 67.32 | 53.79 | 69.82 | 0.00% |
| | AD | 82.91 | 50.00 | 54.50 | 76.18 | 75.00 | 68.94 | 68.00 | 55.96 | 66.44 | -4.84% |
| | HIS | 87.27 | 59.48 | 86.52 | **85.91** | 92.61 | **82.68** | 78.67 | 62.82 | 79.50 | +13.84% |
| | L2 | 82.80 | 60.98 | 85.30 | 64.83 | 69.76 | 50.54 | 44.42 | 47.29 | 63.24 | -9.39% |
| 50% | LLM-Pruner (Channel) | 86.47 | 61.64 | 83.92 | 81.47 | 89.74 | 60.66 | 67.42 | 62.82 | 74.14 | +6.18% |
| | LLM-Pruner (Block) | 87.84 | **70.09** | 83.84 | 78.80 | **92.69** | 72.60 | 73.60 | 61.73 | 77.65 | +11.20% |
| | SliceGPT (w/o tune) | 50.92 | 52.79 | 81.22 | 63.19 | 47.70 | 50.98 | 34.93 | 50.90 | 54.08 | -22.54% |
| | SliceGPT (w/ tune) | 83.49 | 57.45 | 81.37 | 82.16 | 55.05 | 65.79 | 71.70 | 51.62 | 68.58 | -1.78% |
| | **HIES (ours)** | **90.71** | 68.52 | **86.80** | 85.73 | 92.65 | **82.68** | **79.00** | **65.34** | **81.43** | **+16.63%** |

**Datasets.** We evaluate on various widely-adopted benchmarks: GLUE (Wang et al., 2018), HellaSwag (Zellers et al., 2019), Winogrande (Sakaguchi et al., 2020), the AI2 Reasoning Challenge—ARC-e/ARC-c (Clark et al., 2018), OBQA (Mihaylov et al., 2018), ImageNet1k (Deng et al., 2009), CIFAR-100 (Krizhevsky, 2009), Food-101 (Bossard et al., 2014), Fashion MNIST (Xiao et al., 2017), VizWiz-VQA (Gurari et al., 2018), and MM-Vet (Yu et al., 2024).

**Baselines.**

- **Random**: Prune attention heads uniformly at random.
- **L2-Norm**: Prune attention heads with smaller weight magnitudes under the $\ell_2$ norm. This criterion leverages parameter norms as a direct measure of structural salience.
- **HIS** (Michel et al., 2019): Prune attention heads with the smallest head-importance first.
- **Attention Deficit (AD; $1-$Attention Entropy)** (Zhai et al., 2023): Prune attention heads with smaller attention entropy first, i.e., heads exhibiting more concentrated attention patterns.
- **LLM-Pruner (Channel-wise)** (Ma et al., 2023): Prune attention-layer channels based on first-order Taylor expansion of the loss. Importance scores are computed per attention channel, and pruning proceeds while preserving the most critical channels.
- **LLM-Pruner (Block-wise)** (Ma et al., 2023): Extend the channel-wise pruning strategy to whole attention blocks, guided by global importance ranking. Following the best-performing configuration reported in prior work, we retain the first three layers and the final layer, pruning the others. This variant also restricts pruning to attention layers.
- **SliceGPT**: Project activations onto principal components estimated from calibration data, removing directions corresponding to less important subspaces (Ashkboos et al., 2024). This preserves semantic subspaces while reducing redundancy.

## 5.2 MAIN RESULTS

We evaluate HIES using two key metrics: model quality and stability.[4] As reported in Table 1, HIES improves model quality by 8.21% on average. Table 2 further demonstrates a 3.3% average stability gain over HIS. Notably, at a pruning ratio of 50%, HIES achieves gains of up to 15.2% in model

---

[4]Experimental details are provided in Appendix 5.1.

Table 2: Stability results with BERT$_{base}$ on GLUE tasks. We report stability (%) against the unpruned model. Percentage improvements (in blue) are relative to HIS within each pruning ratio.

| Pruning Ratio | Method | SST-2 | CoLA | MRPC | QQP | STS-B | QNLI | MNLI | RTE | Average | |
|---|---|---|---|---|---|---|---|---|---|---|---|
| 10% | HIS | 97.71 | **96.55** | **94.36** | 97.30 | **99.47** | 95.79 | 95.66 | 94.22 | 96.63 | 0.00% |
| | L2 | 95.18 | 76.41 | 26.72 | 68.63 | 21.87 | 60.33 | 72.44 | 39.71 | 57.41 | -40.58% |
| | AD | **98.51** | 96.36 | 93.63 | **97.80** | 89.07 | 97.25 | **96.15** | **94.95** | 95.71 | -0.95% |
| | LLM-Pruner | 97.02 | 85.81 | 75.25 | 85.40 | 39.53 | 90.39 | 84.98 | 86.28 | 80.71 | -16.50% |
| | **HIES (ours)** | 98.17 | 96.36 | 94.12 | 97.80 | 96.67 | 97.25 | 95.74 | 93.50 | 96.45 | **-0.19%** |
| 30% | HIS | 94.38 | 90.03 | **90.03** | 94.77 | 78.93 | 92.95 | 91.56 | 87.36 | 90.25 | 0.00% |
| | L2 | 90.94 | 83.13 | 26.72 | 63.90 | 20.73 | 50.39 | 59.23 | 42.24 | 54.66 | -39.45% |
| | AD | 88.53 | 81.21 | 78.68 | 87.90 | 24.53 | 63.04 | 79.07 | 69.68 | 71.58 | -20.70% |
| | LLM-Pruner | 92.66 | 63.09 | 84.80 | 82.37 | 62.13 | 68.70 | 68.29 | 70.04 | 74.76 | -17.21% |
| | **HIES (ours)** | **97.13** | **93.38** | 89.46 | **94.97** | **86.67** | **93.23** | **91.56** | **88.81** | **91.90** | **+1.83%** |
| 50% | HIS | 90.02 | 40.84 | 83.33 | 88.80 | 69.13 | 85.54 | 81.47 | 78.70 | 77.23 | 0.00% |
| | L2 | 86.47 | 82.26 | 26.72 | 63.90 | 20.40 | 49.81 | 45.58 | 34.30 | 51.93 | -32.76% |
| | AD | 52.18 | 81.21 | 51.72 | 72.33 | 21.87 | 73.49 | 68.49 | 62.82 | 60.76 | -21.32% |
| | LLM-Pruner | 89.68 | 41.32 | 78.19 | 83.27 | 44.13 | 61.58 | 67.52 | 75.81 | 67.19 | -13.02% |
| | **HIES (ours)** | **95.07** | **83.13** | **84.56** | 88.27 | **75.20** | **85.54** | 78.29 | **81.23** | **83.66** | **+8.34%** |

quality and $2.04\times$ in stability compared to the best-performing baseline. These results corroborate our theoretical analysis, demonstrating that HIES preserves end-task performance while markedly enhancing robustness, both critical for reliable and efficient deployment.

## 5.3 EXTENDED EXPERIMENTAL RESULTS

### 5.3.1 HEAD REMOVAL PATTERNS (HEATMAP)

In our main results, HIES exhibits more stable performance gains than HIS at aggressive pruning ratios (e.g., $\geq$,30%). At lower ratios (e.g., $\leq$,10%), HIS and HIES perform comparably, while HIS achieves marginally higher accuracy in a few cases. We posit distinct prioritization. HIS, a gradient-based metric tends to retain redundant, low-risk heads at low sparsity, while HIES adds attention entropy to capture stability, thereby preserving specialized low-entropy heads that enhance robustness.

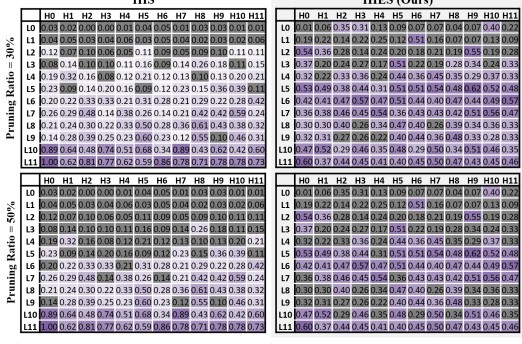

Figure 3: Head-level pruning patterns on the CoLA dataset across pruning ratios from 30% to 50% pruning ratios. Pruned heads are shaded in gray.

Pruning heatmap analysis provides empirical support for this distinction. As shown in Fig. 3, HIS (left panels) tends to remove heads primarily from the lower layers, producing an approximately bottom-up pattern. This behavior is guided by a one-step gradient saliency, which estimates the importance of each attention head based on a single backward pass through the model— this assigns higher importance to heads whose activations have a larger immediate effect on the loss. Meanwhile, HIES (right panels) yields a more dispersed selection spanning lower, middle, and upper layers. We attribute this to the entropy-aware term, which leverages structural properties of the attention distribution (concentration vs. dispersion) in addition to gradient sensitivity, thereby promoting diversity across layers in pruning decisions. Consequently, HIES exhibits more stable performance across pruning ratios, yielding flatter accuracy–sparsity curves than HIS. Additional results are provided in Appendix D.2.

### 5.3.2 SCALABILITY TO LARGER TRANSFORMER MODELS

On LLaMA-2$_{7B}$, we evaluate pruning on HellaSwag, Winogrande, ARC-Easy (ARC-e), ARC-Challenge (ARC-c), and Open Book Question Answering (OBQA), comparing HIES with HIS across pruning ratios of 10–60%. HIES improves accuracy by up to +10.54% and stability by up to +6.21% relative to HIS, averaged over tasks. This advantage persists uniformly across pruning ratios, indicating that the same pruning mechanism scales effectively to larger models with higher head counts. Notably, ARC-c is the most difficult benchmark in this suite, as reflected by its lowest base accuracy. Even under this challenging setting, HIES achieves consistent and often larger accuracy gains relative to HIS, underscoring its robustness not only on easier tasks but also on the hardest ones.

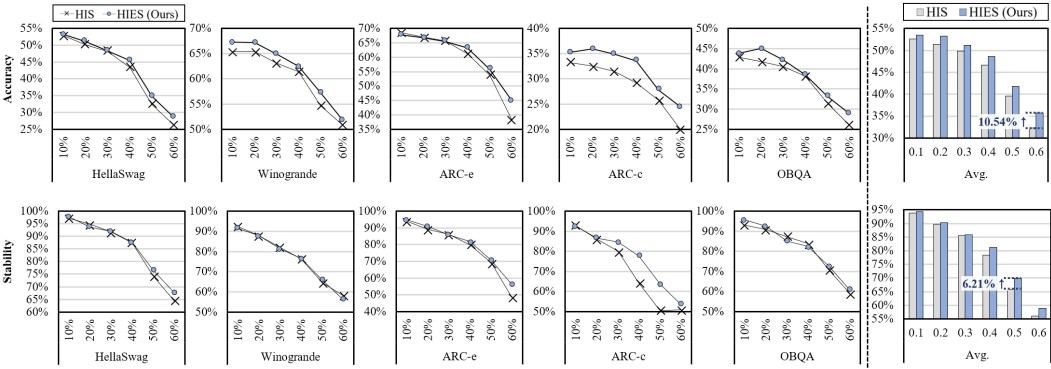

Figure 4: Comparison of HIES (Ours) and HIS on HellaSwag, Winogrande, ARC-e, ARC-c and OBQA across pruning ratios from 10% to 60% (x-axis). The top row reports task accuracy, while the bottom row reports stability. The rightmost panels summarize the mean over tasks.

Table 3: Experimental results with ViT$_{Large}$ on image classification benchmarks and LLaVA-1.5$_{Large}$ on multi-modal tasks. Relative improvements of HIES (Ours) over HIS are shown in blue.

| | Vision Transformer Model (ViT$_{Large}$) | | | | | | Vision-Langauge Model (LLaVA-1.5$_{7B}$) | | | | | |
|---|---|---|---|---|---|---|---|---|---|---|---|---|
| | Image Classification | | | | | | Visual Question Answering | | Complex Multimodal | | | |
| | ImageNet1k | | CIFAR-100 | | Avg. | | VizWiz-VQA | | MM-Vet | | Avg. | |
| | HIS | HIES (Ours) | HIS | HIES (Ours) | HIS | HIES (Ours) | | HIS | HIES (Ours) | HIS | HIES (Ours) | HIS | HIES (Ours) |
| 10% | 86.40% | 84.40% | 91.40% | **92.40%** | 88.90% | 88.40% -0.57% | 10% | 48.00% | 47.33% | **41.20%** | 39.40% | **44.60%** | 43.37% -2.84% |
| 20% | 44.80% | **80.20%** | 65.60% | **85.20%** | 55.20% | **82.70%** 33.25% | 30% | 44.00% | **47.00%** | 29.80% | **34.00%** | 36.90% | **40.50%** 8.89% |
| 30% | 6.20% | **55.40%** | 19.40% | **40.00%** | 12.80% | **47.70%** 73.17% | 50% | 32.67% | **39.67%** | 24.60% | **25.20%** | 28.64% | **32.43%** 11.71% |

### 5.3.3 GENERALIZABILITY TO ATTENTION VARIANTS AND TASKS

Table 3 evaluates the generalizability of HIES across attention variants and task domains, spanning vision classification with ViT$_{Large}$ and visual-language reasoning with LLaVA-1.5$_{7B}$.

First, across different attention-based variants, including ViT and LLaVA, HIES consistently shifts the region of sharp accuracy drop to more aggressive sparsity levels compared to HIS. In particular, at high pruning ratios where HIS accuracy collapses, HIES achieves 73.17% higher accuracy on ViT$_{Large}$ at 30% sparsity, compared to HIS (with 12.80% of accuracy). This indicates that HIES effectively captures structural importance regardless of the specific model configuration, enabling stable and reliable pruning.

Second, the advantages of HIES are further extended on complex multi-modal tasks. HIES shows improvement of 11.71% at 50% sparsity compared to HIS on VizWiz-VQA and MM-Vet. HIS frequently suffers from sharp accuracy drops on these tasks, as it relies solely on gradient-based head importance and fails to account for the structural diversity of attention heads. As a result, pruning based on HIS alone may remove heads that are essential for preserving cross-modal alignment and semantic grounding. In contrast, HIES leverages AE to capture these structural signals, resulting in more stable performance across challenging vision-language benchmarks. HIES also demonstrates robust and stable performance on downstream tasks (i.e., CIFAR-100, Food-101, and Fashion MNIST) — detailed results are provided in Appendix D.4.

Overall, these results confirm that combining HIS with AE produces pruning signals that generalize across architectures and modalities, preserving accuracy and stability. This underscores the potential of HIES as a broadly applicable criterion for efficient and reliable pruning of Transformer models.

## 6 CONCLUSION

In this paper, we present HIES, a novel pruning criterion that jointly leverages gradient-based head importance and attention entropy to better characterize per-head contributions. By combining complementary structural and behavioral signals, HIES outperforms HIS and other baselines, delivering both higher accuracy and greater inference-time stability. Beyond empirical gains, our analysis highlights the critical role of entropy. We believe HIES offers a principled direction for stable and efficient pruning of Transformer-based models, with potential to extend toward broader structured sparsity and large-scale model deployment.

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

# A  RELATED WORK

**Model Compression.** Language models (Devlin et al., 2018; Liu et al., 2019; Lewis et al., 2019) have rapidly advanced in scale and capability, intensifying the demand to reduce their parameter sizes and inference latency (Molchanov et al., 2017; Gordon et al., 2020). To compress large language models efficiently, various approaches have been explored, including pruning (Liu et al., 2021b; Kurtic et al., 2022; Xu et al., 2021), knowledge distillation (Sun et al., 2019; 2020; Pan et al., 2020), quantization (Bai et al., 2020; Yao et al., 2022; Zafrir et al., 2019; Chen et al., 2025a), and other techniques, like low-rank approximation methods (Saha et al., 2024; Li et al., 2023; Wong et al., 2025) or weight space decomposition methods such as (Ashkboos et al., 2024).

**Structured Pruning.** Among these, structured pruning—which removes entire architectural components rather than individual weights— can be performed at various granularities, such as whole layers (Fan et al., 2020), multi-head attention modules (Michel et al., 2019; Voita et al., 2019), or feed-forward networks (Lagunas et al., 2021; Hou et al., 2020b). Recently, attention head pruning has gained particular traction in the context of large language models. It directly reduces attention FLOPs and KV-cache size by lowering the number of active heads while preserving the layer topology, thereby simplifying checkpoint compatibility and serving integration. Consequently, several large-scale studies adopt head-level pruning as a practical axis in LLM compression pipelines (Ma et al., 2023; Jaradat et al., 2024; Muralidharan et al., 2024). Ma et al. (2023) propose a unified framework that integrates structured head pruning into the training pipeline of large language models, achieving substantial sparsity without accuracy degradation.

**Entropy in Reinforcement Learning and Transformers.** Entropy has long been employed in reinforcement learning (RL) as a means of encouraging exploration and preserving policy diversity. By introducing an entropy term into the policy objective, RL methods prevent premature convergence to deterministic strategies and mitigate policy collapse (Bharadhwaj et al., 2020; Liu et al., 2021a; Wang et al., 2025). Extensions of this idea include parameter-space entropy regularization to explicitly control diversity (Han et al., 2023), large-deviation interpretations of entropy-regularized RL (Arriojas et al., 2023), and state-distribution entropy regularization for improved robustness and generalization (Ashlag et al., 2025).

In Transformer architectures, attention entropy (AE) has been introduced to quantify the concentration of each head's focus across input tokens. Zhai et al. (2023) report that persistently low AE—an "entropy collapse" state—correlates strongly with training instabilities such as oscillations in the loss surface and divergence across model scales. Their theoretical analysis ties AE to the spectral norm of attention logits, and they propose $\sigma$Reparam to prevent collapse by enforcing a lower bound on entropy. More recent work identifies variance sensitivity in the softmax transformation as another source of entropy collapse (Anonymous, 2025). Together, these findings highlight attention entropy as a critical factor for stability, motivating its integration as a complementary signal in pruning frameworks.

## B PROOFS AND DETAILS

**Overview.** This appendix collects the formal analyses that underpin Section 4.2 and fixes notation and technical conventions used throughout. We first develop loss-increase control for head pruning via gradient-based head importance (HIS), derive operator-norm curvature bounds for the cross-entropy objective, and justify the first-order approximation by quantifying when quadratic terms are negligible (Appendix B.1).

We then turn to generalization: starting from token-averaged notation and a neighboring-dataset construction, we link perturbations of attention distributions to output deviations and establish an entropy–total variation inequality that couples attention entropy with stability, culminating in a stability–generalization connection and practical constraints (Appendix B.2).

Building on these components, we present a risk upper bound whose surrogate minimization yields the proposed HIES objective and clarifies its role as a principled pruning criterion (Appendix B.3). Finally, we prove an orthogonality result between the centered HIS and entropy directions, showing their complementarity and explaining why combining the two signals improves robustness across pruning regimes (Appendix B.4).

Collectively, these results provide (i) tight loss-control guarantees under operator-norm curvature, (ii) an entropy-based route from stability to generalization, and (iii) a unified risk-motivated justification for HIES.

### B.1 LOSS-INCREASE CONTROL VIA HEAD IMPORTANCE (SECTION 4.2.1)

#### B.1.1 LOSS BOUND WITH OPERATOR-NORM CONTROL

We write $\beta_y := \|\nabla_{\mathbf{y}}^2 \mathcal{L}\|_2$ for the operator-norm curvature at the representation $\mathbf{y}$.

Consider the second-order Taylor expansion in $\mathbf{y}$:

$$\mathcal{L}(\mathbf{y} + \delta\mathbf{y}) \leq \mathcal{L}(\mathbf{y}) + \langle \nabla_{\mathbf{y}}\mathcal{L}, \delta\mathbf{y} \rangle_F + \frac{1}{2}\delta\mathbf{y}^\top \nabla_{\mathbf{y}}^2 \mathcal{L}\,\delta\mathbf{y}.$$

Taking expectations and using the operator-norm bound yields

$$\Delta\mathcal{L} := \mathbb{E}_{\mathcal{D}}\big[\mathcal{L}(\mathbf{y} + \delta\mathbf{y}) - \mathcal{L}(\mathbf{y})\big] \leq \mathbb{E}\big[\langle \nabla_{\mathbf{y}}\mathcal{L}, \delta\mathbf{y}\rangle_F\big] + \frac{\beta_y}{2}\mathbb{E}\big[\|\delta\mathbf{y}\|_F^2\big].$$

Since $\delta\mathbf{y} = \mathrm{Concat}(\delta A_1, \ldots, \delta A_{|\mathrm{H}|})$, $\|\delta\mathbf{y}\|_F^2 = \sum_h \|\delta A_h\|_F^2$ and $\delta A_h = -(1 - m_h)A_h$, the quadratic term equals $\frac{\beta_y}{2}\sum_h (1 - m_h)\|A_h\|_F^2$. For the first-order term, using the absolute value in the HIS definition and head-wise triangle inequality,

$$\mathbb{E}\big[\langle \nabla_{\mathbf{y}}\mathcal{L}, \delta\mathbf{y}\rangle_F\big] = -\sum_h (1 - m_h)\,\mathbb{E}\big[\langle \nabla_{A_h}\mathcal{L}, A_h\rangle_F\big] \leq \sum_h (1 - m_h)\,\mathrm{HIS}_h,$$

hence

$$\Delta\mathcal{L} \leq \sum_{h=1}^{|\mathrm{H}|}(1 - m_h)\,\mathrm{HIS}_h + \frac{\beta_y}{2}\sum_{h=1}^{|\mathrm{H}|}(1 - m_h)\,\|A_h\|_F^2. \tag{9}$$

**Plug-ins (default: binary).**

**Binary (Sigmoid) CE.** $\beta_y \leq \frac{1}{4}\|W^O\|_2^2$,

$$\Rightarrow \quad \Delta\mathcal{L} \leq \sum_h (1 - m_h)\,\mathrm{HIS}_h + \frac{1}{8}\|W^O\|_2^2 \sum_h (1 - m_h)\,\|A_h\|_F^2.$$

**Multiclass softmax CE.** $\beta_y \leq \frac{1}{2}\|W^O\|_2^2$,

$$\Rightarrow \quad \Delta\mathcal{L} \leq \sum_h (1 - m_h)\,\mathrm{HIS}_h + \frac{1}{4}\|W^O\|_2^2 \sum_h (1 - m_h)\,\|A_h\|_F^2.$$

### B.1.2 Norms and Inner Products

We use token-averaged Frobenius norms and inner products: $\|A_h\|_F^2 = \frac{1}{n}\sum_{i=1}^n \|A_h(i)\|_2^2$ and $\langle U, V\rangle_F = \frac{1}{n}\sum_{i=1}^n \langle U(i), V(i)\rangle$ (*with batching*: replace $\frac{1}{n}$ by $\frac{1}{Bn}$).

### B.1.3 Estimating $\|W^O\|_2$ and Blockwise Norms via Power Iteration

The spectral norm of a matrix $M \in \mathbb{R}^{m\times n}$ is defined as

$$\|M\|_2 := \max_{\|x\|_2=1} \|Mx\|_2,$$

which measures the maximum $\ell_2$-amplification factor over all unit vectors. By the singular value decomposition (SVD), $M = U\Sigma V^\top$, where $\Sigma = \mathrm{diag}(\sigma_1, \sigma_2, \dots)$ with $\sigma_1 \geq \sigma_2 \geq \cdots \geq 0$, we have

$$\|M\|_2 = \sigma_{\max}(M),$$

i.e., the spectral norm equals the largest singular value. This follows since $U$ and $V$ are orthogonal and preserve the $\ell_2$-norm, so the maximization reduces to aligning $x$ with the right singular vector corresponding to $\sigma_{\max}$. Exact computation via full SVD costs $O(\min\{m^2 n, mn^2\})$, which is prohibitive for large $M$. Instead, we approximate $\sigma_{\max}(M)$ using the *power iteration* method: starting from a random unit vector $v_0$, iterate

$$u_t \leftarrow \frac{Mv_t}{\|Mv_t\|_2},$$

$$v_{t+1} \leftarrow \frac{M^\top u_t}{\|M^\top u_t\|_2}.$$

After $T$ iterations, $\|Mv_T\|_2$ converges to $\sigma_{\max}(M)$, and $v_T$ approximates the corresponding right singular vector. We apply this procedure to $W^O$ and, for a tighter quadratic term in our bound, to its head-wise column blocks $W^O_{(:,\mathcal{I}_h)}$, forming

$$\sum_h (1 - m_h)\|W^O_{(:,\mathcal{I}_h)}\|_2^2 \, \|A_h\|_F^2.$$

Here, $\mathcal{I}_h \subset \{1,\dots,d\}$ denotes the set of column indices in $W^O$ corresponding to the $d_v$ output dimensions of head $h$. Thus, $W^O_{(:,\mathcal{I}_h)} \in \mathbb{R}^{d\times d_v}$ is the column block of $W^O$ mapping the $d_v$-dimensional output of head $h$ to the $d$-dimensional model space.

### B.1.4 Cross-Entropy Curvature and Propagation to $\mathbf{y}$

**Logit-space Hessian (binary vs. multiclass).** **Binary (sigmoid) CE.** For a single logit $z$ with $p = \sigma(z)$,

$$\frac{d^2\mathcal{L}}{dz^2} = p(1-p) \leq \tfrac{1}{4},$$

hence $\|\nabla_{\mathbf{z}}^2 \mathcal{L}\|_2 \leq \frac{1}{4}$.

**Multiclass softmax CE.** For logits $\mathbf{z} \in \mathbb{R}^C$ and $\mathbf{p} = \mathrm{softmax}(\mathbf{z})$,

$$\nabla_{\mathbf{z}}^2 \mathcal{L} = \mathrm{diag}(\mathbf{p}) - \mathbf{p}\mathbf{p}^\top, \qquad \|\nabla_{\mathbf{z}}^2 \mathcal{L}\|_2 \leq \tfrac{1}{2}.$$

*Proof sketch.* For any unit vector $v$, $v^\top(\mathrm{diag}(p) - pp^\top)v = \sum_i p_i v_i^2 - (\sum_i p_i v_i)^2 = \mathrm{Var}_p(v)$. By Popoviciu's inequality, $\mathrm{Var}_p(v) \leq \frac{(\max_i v_i - \min_i v_i)^2}{4} \leq \frac{1}{2}$ for $\|v\|_2 = 1$. Tightness holds at $C = 2$, $p = (\frac{1}{2}, \frac{1}{2})$.

**Mapping through $W^O$.** With the immediate linear projection $\mathbf{z} = \mathbf{y}W^O$,

$$\nabla_{\mathbf{y}}^2 \mathcal{L} = (W^O)^\top \nabla_{\mathbf{z}}^2 \mathcal{L} \, W^O, \qquad \beta_y := \|\nabla_{\mathbf{y}}^2 \mathcal{L}\|_2 \leq \begin{cases} \frac{1}{4}\|W^O\|_2^2, & \text{binary CE}, \\ \frac{1}{2}\|W^O\|_2^2, & \text{multiclass softmax CE.} \end{cases}$$

(A blockwise refinement replaces $\|W^O\|_2$ by $\|W^O_{(:,\mathcal{I}_h)}\|_2$ head-wise.)

### B.1.5 WHY THE QUADRATIC TERM IS TYPICALLY NEGLIGIBLE

We first note

$$\text{HIS}_h \;=\; \mathbb{E}\big[\,|\langle \nabla_{A_h}\mathcal{L}, A_h\rangle_F|\,\big] \;=\; \mathbb{E}\big[\,|\cos\phi_h|\,\|\nabla_{A_h}\mathcal{L}\|_F\,\|A_h\|_F\,\big],$$

where the expectation is over $x\sim\mathcal{D}$ with token-averaging as in Appendix B.1.2. Assume there exists $g > 0$ such that, for all heads under consideration,

$$\text{HIS}_h \;\geq\; g\,\mathbb{E}\big[\|A_h\|_F\big],$$

e.g., define

$$g \;:=\; \min_{h\in\{1,\dots,\text{—H—}\}} \; \mathbb{E}\big[\,|\cos\phi_h|\,\|\nabla_{A_h}\mathcal{L}\|_F\,\big],$$

where $\cos\phi_h \frac{\langle \nabla_{A_h}\mathcal{L}, A_h\rangle_F}{\|\nabla_{A_h}\mathcal{L}\|_F\,\|A_h\|_F}$ denotes the cosine alignment between the head's gradient and activation.

Then

$$\frac{\text{quadratic}}{\text{first-order}} \;\leq\; \frac{\frac{\beta_y}{2}\sum_h(1-m_h)\,\mathbb{E}[\|A_h\|_F^2]}{\sum_h(1-m_h)\,\text{HIS}_h} \;\leq\; \frac{\beta_y}{2}\cdot\frac{\max_h\mathbb{E}[\|A_h\|_F]}{g},$$

where the second inequality uses $\sum_h(1-m_h)\mathbb{E}[\|A_h\|_F^2] \leq \big(\max_h\mathbb{E}[\|A_h\|_F]\big)\sum_h(1-m_h)\mathbb{E}[\|A_h\|_F]$ and the per-head lower bound $\text{HIS}_h \geq g\,\mathbb{E}[\|A_h\|_F]$.

Recalling $\beta_y \leq c\,\|W^O\|_2^2$ with $c = \frac{1}{4}$ for binary CE and $c = \frac{1}{2}$ for multiclass softmax CE (cf. Appendix B.1.4), we obtain

$$\frac{\text{quadratic}}{\text{first-order}} \;\leq\; \frac{c}{2}\,\|W^O\|_2^2\cdot\frac{\max_h\mathbb{E}[\|A_h\|_F]}{g}.$$

A blockwise refinement further tightens this by replacing $\|W^O\|_2^2$ with $\max_h\|W^O_{(:,\mathcal{I}_h)}\|_2^2$. Since (i) LayerNorm controls token-wise activation scales (thus $\max_h\mathbb{E}[\|A_h\|_F]$), and (ii) $g$ is bounded away from zero under non-degenerate alignment, the ratio is typically small. Hence the first-order term dominates in practice, while the second-order term remains explicitly controlled by the plug-in bounds in Appendix B.1.1.

### B.1.6 REMARKS ON HIS WITH ABSOLUTE VALUES

The absolute value in equation 2 is part of the definition to prevent cancellation across samples; consequently, the triangle inequality turns the first-order term into an additive upper bound $\sum_h(1-m_h)\,\text{HIS}_h$ (cf. Appendix B.1.1). If $\langle \nabla_{A_h}\mathcal{L}, A_h\rangle_F < 0$ on some samples, masking that head could locally decrease the loss; the metric remains conservative by construction.

## B.2 GENERALIZATION GAP AND ATTENTION ENTROPY (SECTION 4.2.2)

### B.2.1 NOTATION AND TOKEN AVERAGING

For head $h$ and query token $t\in\{1,\dots,n(x)\}$, let $\boldsymbol{\alpha}_t^{(h)}(x)\in\Delta^{n(x)-1}$ denote the attention distribution over keys, and $H(\mathbf{p}) := -\sum_j p_j\log p_j$ the entropy. Define the *token-averaged, length-normalized entropy* and *deficit* by

$$\text{AE}_h(x) := \frac{1}{n(x)\,\log n(x)}\sum_{t=1}^{n(x)} H\big(\boldsymbol{\alpha}_t^{(h)}(x)\big) \in [0,1],$$

$$\text{AD}_h(x) := \frac{1}{n(x)\,\log n(x)}\sum_{t=1}^{n(x)} \Big(\log n(x) - H\big(\boldsymbol{\alpha}_t^{(h)}(x)\big)\Big) \;=\; 1 - \text{AE}_h(x) \in [0,1].$$

For neighboring datasets $(\mathcal{S}, \mathcal{S}')$, write the symmetric aggregation

$$\overline{\mathrm{AD}}_h(x) := \tfrac{1}{2}\big(\mathrm{AD}_h(x) + \mathrm{AD}'_h(x)\big).$$

All token averages exclude padding positions and use the effective context length for causal masking(cf. Appendix B.2.1). Here $(\mathcal{S}, \mathcal{S}')$ are *neighboring* datasets that differ in one example.

### B.2.2 NEIGHBORING DATASETS AND WHY THEY APPEAR

We call two datasets $S = (z_1, \ldots, z_N)$ and $S' = (z_1, \ldots, z_{i-1}, z'_i, z_{i+1}, \ldots, z_N)$ *neighboring* if they differ in exactly one example.

**Why neighboring datasets?**

- **Symmetrization.** Introduce an i.i.d. ghost sample $S' \sim \mathcal{D}^N$ to rewrite the expected generalization gap as an average of sample-wise differences, e.g., $\mathbb{E}_{S,S'}\big[\tfrac{1}{N}\sum_{i=1}^N\big(\ell(f_S; z_i) - \ell(f_{S'}; z'_i)\big)\big]$, which is amenable to concentration and stability arguments.
- **Replace-one stability.** Measure sensitivity to a single replacement by comparing $f_S$ with $f_{S^{(i \leftarrow z')}}$, where $S^{(i \leftarrow z')}$ replaces $z_i$ by $z'_i$; under $\gamma$-uniform stability and bounded loss $B$, this yields $\mathbb{E}_S[\mathcal{G}(S)] \leq 2\gamma + \frac{B}{N}$.
- **Symmetric inequalities.** Our entropy–total variation (TV) control is symmetric in two distributions $(\alpha, \alpha')$; we thus aggregate via $\overline{\mathrm{AD}}_h(x) := \tfrac{1}{2}\big(\mathrm{AD}_h(x) + \mathrm{AD}'_h(x)\big)$, which streamlines notation and tightens constants in the perturbation bound.

### B.2.3 FROM ATTENTION PERTURBATION TO OUTPUT PERTURBATION

For token $t$, the head output is $a_h(t) = \big(\boldsymbol{\alpha}_t^{(h)}\big)^\top V_h \in \mathbb{R}^{d_v}$, hence for neighboring datasets,

$$\Delta_h(t) := a_h(t) - a'_h(t) = \big(\boldsymbol{\alpha}_t^{(h)} - \boldsymbol{\alpha}_t'^{(h)}\big)^\top V_h.$$

With $\|V_h\|_{\infty \to 2} := \max_j \|V_h(j,:)\|_2$ and $\|V_h\|_{\infty \to 2} \leq M$,

$$\|\Delta_h(t)\|_2 \leq \|\boldsymbol{\alpha}_t^{(h)} - \boldsymbol{\alpha}_t'^{(h)}\|_1 \cdot \|V_h\|_{\infty \to 2} \leq M\|\boldsymbol{\alpha}_t^{(h)} - \boldsymbol{\alpha}_t'^{(h)}\|_1. \tag{10}$$

Averaging over tokens and applying the mask $m_h$,

$$\|\Delta(x)\|_2 := \frac{1}{n(x)} \sum_{t=1}^{n(x)} \sum_{h=1}^{|\mathrm{H}|} (1 - m_h)\|\Delta_h(t)\|_2.$$

### B.2.4 ENTROPY–TOTAL VARIATION (TV) CONTROL

**Lemma 4** (Entropy–TV inequality). *For* $\mathbf{p}, \mathbf{q} \in \Delta^{n-1}$ *and* $\mathbf{u}$ *uniform,* $\|\mathbf{p} - \mathbf{q}\|_1^2 \leq 4\big[H(\mathbf{u}) - H(\mathbf{p}) + H(\mathbf{u}) - H(\mathbf{q})\big].$

*Proof.* Triangle inequality and $(a + b)^2 \leq 2(a^2 + b^2)$ give $\|\mathbf{p} - \mathbf{q}\|_1^2 \leq 2(\|\mathbf{p} - \mathbf{u}\|_1^2 + \|\mathbf{q} - \mathbf{u}\|_1^2)$. Pinsker w.r.t. $\mathbf{u}$ yields $\|\mathbf{p} - \mathbf{u}\|_1^2 \leq 2(\log n - H(\mathbf{p}))$ and likewise for $\mathbf{q}$. $\square$

Applying Lemma 4 to equation 10 token-wise and averaging,

$$\frac{1}{n(x)} \sum_{t=1}^{n(x)} \|\boldsymbol{\alpha}_t^{(h)} - \boldsymbol{\alpha}_t'^{(h)}\|_1 \leq \sqrt{\frac{1}{n(x)} \sum_{t=1}^{n(x)} \|\boldsymbol{\alpha}_t^{(h)} - \boldsymbol{\alpha}_t'^{(h)}\|_1^2} \leq \sqrt{8 \log n(x)}\sqrt{\overline{\mathrm{AD}}_h(x)}.$$

Therefore,

$$\|\Delta(x)\|_2 \leq M\sqrt{8 \log n(x)} \sum_{h=1}^{|\mathrm{H}|} (1 - m_h)\sqrt{\overline{\mathrm{AD}}_h(x)}.$$

By Cauchy–Schwarz and $\sum_h (1 - m_h) = |\mathrm{H}|\rho$,

$$\|\Delta(x)\|_2 \leq \underbrace{\sqrt{8}\,M\,\sqrt{|\mathrm{H}|\rho\,\log n(x)}}_{=:\,C_{\mathrm{AE}}(x)} \cdot \sqrt{\sum_{h=1}^{|\mathrm{H}|} (1 - m_h)\,\overline{\mathrm{AD}}_h(x)}. \tag{11}$$

### B.2.5 STABILITY AND GENERALIZATION

Let $\gamma := L_\ell \, \mathbb{E}_{\mathcal{S}}[\|\Delta(x)\|_2]$. By on-average replace-one stability (Bousquet and Elisseeff, 2002, Def. 6),

$$\mathbb{E}_{\mathcal{S}}\big[\mathcal{G}(\mathcal{S})\big] \leq 2\gamma, \qquad \mathcal{G}(\mathcal{S}) \leq 2\gamma + \tfrac{B}{N}.$$

Using equation 11 and Jensen for $\sqrt{\cdot}$,

$$\gamma \;\leq\; L_\ell \, \mathbb{E}_{\mathcal{S}}\big[C_{\mathrm{AE}}(x)\big] \cdot \sqrt{\sum_{h=1}^{|\mathrm{H}|}(1 - m_h) \, \mathbb{E}_{\mathcal{S}}\big[\overline{\mathrm{AD}}_h(x)\big]}.$$

Taking a representative $n$ (e.g., average/max effective length) yields the main-text constant $C_{\mathrm{AE}} = \sqrt{8}\, M \, \sqrt{|\mathrm{H}|\rho \, \log n}$ and Eq. equation 5.

### B.2.6 CONSTANTS AND PRACTICAL REMARKS

- **Operator norm.** $\|V_h\|_{\infty \to 2} := \max_j \|V_h(j,:)\|_2$; take $M := \max_h \|V_h\|_{\infty \to 2}$ (controlled by LayerNorm/weight norms).

- **Sequence length.** For padding/causal masking, replace $n(x)$ by the effective context length; averages exclude padded positions.

- **Deficit aggregation.** On-average: $\overline{\mathrm{AD}}_h = \frac{1}{2}(\mathrm{AD}_h + \mathrm{AD}'_h)$; Uniform: $\overline{\mathrm{AD}}_h = \max\{\mathrm{AD}_h, \mathrm{AD}'_h\}$.

- **Do not pool entropies.** Since using $H(\frac{1}{n}\sum_t \alpha_t)$ can underestimate deficit (Jensen) and weaken control, token-wise entropies are required.

## B.3 RISK UPPER BOUND AND HIES MINIMIZATION (SECTION 4.2.3)

*Proof.* Let $\mathrm{supp}(m) := \{h : m_h = 1\}$ denote the set of retained heads. Suppose an admissible mask $m'$ with $|\mathrm{supp}(m')| = k$ is not optimal. Then there exist $i \in \mathrm{supp}(m')$ and $j \notin \mathrm{supp}(m')$ such that $\mathrm{HIES}_j > \mathrm{HIES}_i$. Consider the mask $\tilde{m}$ that swaps $i$ and $j$ (retain $j$, prune $i$); the constraint in equation 7 is preserved. The objective in equation 6 changes by

$$\Delta\mathcal{R} = \big[\mathrm{HIES}_i\big] - \big[\mathrm{HIES}_j\big] < 0,$$

since $j$ was contributing to the sum (pruned) and $i$ was not (retained). Hence $\tilde{m}$ has a strictly smaller objective, contradicting the minimality of $m'$. Therefore retaining the $k$ heads with the largest HIES is optimal; equivalently, pruning the $|\mathrm{H}| - k$ smallest HIES is optimal. $\qquad\square$

## B.4 ORTHOGONALITY AND COMPLEMENTARITY (SECTION 4.2.4)

**Preliminaries.** For head $h$, let $\boldsymbol{\alpha}^{(h)} \in \Delta^{n-1}$ be the attention probability vector, $V_h \in \mathbb{R}^{n \times d_v}$ the value matrix, and

$$A_h \;=\; \boldsymbol{\alpha}^{(h)} V_h \;\in\; \mathbb{R}^{1 \times d_v}.$$

Define

$$g_h \;:=\; V_h\big(\nabla_{A_h}\mathcal{L}\big)^\top \;\in\; \mathbb{R}^n, \qquad \mathrm{HIS}_h \;=\; \big|\boldsymbol{\alpha}^{(h)\top} g_h\big|, \qquad \mathrm{AE}_h \;=\; -\sum_{j=1}^{n}\alpha_j^{(h)} \log \alpha_j^{(h)}.$$

**Gradients w.r.t. attention (interior points).** For $\alpha_j^{(h)} > 0$,

$$\nabla_{\boldsymbol{\alpha}^{(h)}}\mathrm{HIS}_h = \mathrm{sign}\big(\boldsymbol{\alpha}^{(h)\top} g_h\big)\, g_h, \qquad \nabla_{\boldsymbol{\alpha}^{(h)}}\mathrm{AE}_h = -\big(\mathbf{1} + \log \boldsymbol{\alpha}^{(h)}\big),$$

where $\log$ is applied elementwise. (At $\boldsymbol{\alpha}^{(h)\top} g_h = 0$, any subgradient in $\{s\, g_h : s \in [-1, 1]\}$ is valid; this does not affect the result in expectation.)

**Simplex projection.** Since $\boldsymbol{\alpha}^{(h)} \in \Delta^{n-1}$, we project onto the tangent space with $P := I - \frac{1}{n}\mathbf{1}\mathbf{1}^\top$ and define

$$\widetilde{\nabla}\mathrm{HIS}_h := P\,\nabla\mathrm{HIS}_h, \qquad \widetilde{\nabla}\mathrm{AE}_h := P\,\nabla\mathrm{AE}_h.$$

*Proof.* By definition, $u_h := \mathrm{sign}(\boldsymbol{\alpha}^{(h)\top} g_h)\, g_h$, $v_h := \mathbf{1} + \log\boldsymbol{\alpha}^{(h)}$, and $\tilde{u}_h := Pu_h$, $\tilde{v}_h := Pv_h$. Then

$$\widetilde{\nabla}_{\boldsymbol{\alpha}^{(h)}}\mathrm{HIS}_h = P\,\nabla_{\boldsymbol{\alpha}^{(h)}}\mathrm{HIS}_h = \tilde{u}_h, \qquad \widetilde{\nabla}_{\boldsymbol{\alpha}^{(h)}}\mathrm{AE}_h = P\,\nabla_{\boldsymbol{\alpha}^{(h)}}\mathrm{AE}_h = -\tilde{v}_h.$$

Hence

$$\mathbb{E}_{\mathcal{S}}\big[\,\langle\widetilde{\nabla}\mathrm{HIS}_h,\ \widetilde{\nabla}\mathrm{AE}_h\rangle\,\big] = \mathbb{E}_{\mathcal{S}}\big[\langle\tilde{u}_h, -\tilde{v}_h\rangle\big] = -\mathrm{tr}\Big(\mathbb{E}_{\mathcal{S}}\big[\tilde{u}_h\tilde{v}_h^\top\big]\Big).$$

Decomposing the second moment,

$$\mathbb{E}_{\mathcal{S}}\big[\tilde{u}_h\tilde{v}_h^\top\big] = \mathrm{Cov}(\tilde{u}_h, \tilde{v}_h) + \mathbb{E}_{\mathcal{S}}[\tilde{u}_h]\,\mathbb{E}_{\mathcal{S}}[\tilde{v}_h]^\top.$$

Under $\mathrm{Cov}(\tilde{u}_h, \tilde{v}_h) = 0$ and $\langle\mathbb{E}_{\mathcal{S}}[\tilde{u}_h], \mathbb{E}_{\mathcal{S}}[\tilde{v}_h]\rangle = 0$ (*or the stronger* $\mathbb{E}_{\mathcal{S}}[\tilde{u}_h] = 0$), we obtain

$$\mathbb{E}_{\mathcal{S}}\big[\,\langle\widetilde{\nabla}\mathrm{HIS}_h,\ \widetilde{\nabla}\mathrm{AE}_h\rangle\,\big] = 0.$$

$\square$

**Technical remarks.** (i) At $\boldsymbol{\alpha}^{(h)\top} g_h = 0$, use any subgradient of $|\cdot|$ for $\nabla_{\boldsymbol{\alpha}^{(h)}}\mathrm{HIS}_h$. (ii) Since $\boldsymbol{\alpha}^{(h)} = \mathrm{softmax}(\cdot)$, we have $\alpha_j^{(h)} > 0$, so $\log\boldsymbol{\alpha}^{(h)}$ (elementwise) is well-defined. (iii) "$\mathrm{Cov}(x, y) = 0$" denotes the *cross-covariance matrix* being zero, not merely componentwise uncorrelatedness. (iv) If one omits the projection $P$, the same argument applies with $u_h, v_h$ replacing $\tilde{u}_h, \tilde{v}_h$ under the analogous conditions $\mathrm{Cov}(u_h, v_h) = 0$ and $\langle\mathbb{E}_{\mathcal{S}}[u_h], \mathbb{E}_{\mathcal{S}}[v_h]\rangle = 0$.

## C  EXPERIMENTAL SETUP

### C.1  EXPERIMENTAL SETUP FOR MOTIVATION STUDY

We analyze accuracy degradation and head behaviors under HIS-based pruning. In our diagnostic study, we analyze the phenomena of pruning by HIS on BERT, focusing on detailed attention head behaviors during inference. Following prior work analyzing BERT's attention geometry and mechanisms (Clark et al., 2019; Rogers et al., 2020; Wang et al., 2024), we further explore attention head pruning dynamics.

### C.2  MODEL

Table 4: Summary of model parameters and architectures.

| Model | Parameters | # Layers | # Attention Heads | Architecture / Key Details |
|---|---|---|---|---|
| $\text{BERT}_{\text{base}}$ | 110M | 12 | 12 | Transformer encoder; pre-trained on masked language modeling and next sentence prediction tasks. |
| $\text{LLaMA-2}_{7B}$ | 7B | 32 | 32 | Transformer decoder-only; trained on large-scale text corpora for general-purpose language modeling. |
| $\text{ViT}_{\text{Large}}$ | 307M | 24 | 16 | Vision Transformer; patch-based image tokenization (16×16), pre-trained on ImageNet for image classification tasks. |
| $\text{LLaVA-1.5}_{7B}$ | 7B | 32 | 32 | Multi-modal LLaMA variant integrating a visual encoder; capable of joint image-text understanding and generation. |

### C.3  COMPUTING RESOURCES

Our experimental setup leverages two RTX 4090 GPUs with 24GB memory for NLU tasks using BERT and for image classification tasks using ViT. Experiments involving LLMs such as LLaMA and multi-modal VLMs such as LLaVA were conducted on H100 GPU with 80GB memory. For the MM-Vet benchmark, we evaluated model responses using the OpenAI API to handle open-ended answer scoring.

### C.4  DATASET STATISTICS

#### C.4.1  NATURAL LANGUAGE UNDERSTANDING TASK

We present the dataset statistics of GLUE (Wang et al., 2018) in Table 5.

#### C.4.2  IMAGE CLASSIFICATION TASK

Table 6 lists dataset statistics for the image classification task in the Computer Vision (CV) domain.

#### C.4.3  MULTI-MODAL VISION-LANGUAGE TASK

To evaluate the effectiveness of our pruning method on multi-modal vision-language models (VLMs), we used two benchmark datasets: **VizWiz-VQA** (Gurari et al., 2018) and **MM-Vet** (Yu et al., 2024). The evaluation was conducted using the $\text{LLaVA1.5}_{7B}$ model.

**VizWiz-VQA:** is designed for Visual Question Answering (VQA) in the context of assisting people who are blind. Each visual question originates from a real-world setting where blind users captured images and recorded spoken questions, and is accompanied by ten crowdsourced answers. The dataset poses two evaluation tasks: predicting the correct answer given an image and question, and detecting whether a question cannot be answered.

Table 5: Summary of the NLG benchmark.

| NLU Benchmark | | | | | | |
|---|---|---|---|---|---|---|
| Dataset | # Train | # Valid | # Test | # Label | Task | Evaluation Metric |
| Single-Sentence Classification (GLUE) | | | | | | |
| CoLA | 8,551 | 521 | 522 | 2 | Acceptability | Matthews corr |
| SST-2 | 66,349 | 1,000 | 872 | 2 | Sentiment | Accuracy |
| Pairwise Text Classification (GLUE) | | | | | | |
| MNLI | 392,702 | 9,832 | 9,815 | 3 | NLI | Accuracy |
| RTE | 2,490 | 138 | 139 | 2 | NLI | Accuracy |
| QQP | 362,846 | 1,000 | 40,431 | 2 | Paraphrase | Accuracy |
| MRPC | 3,668 | 204 | 204 | 2 | Paraphrase | F1 score |
| QNLI | 103,743 | 1,000 | 5,463 | 2 | QA/NLI | Accuracy |
| Pairwise Text Classification (GLUE) | | | | | | |
| STS-B | 5,749 | 750 | 750 | 1 | Similarity | Pearson corr |

Table 6: Summary of the CV benchmark.

| CV Benchmark | | | | | | |
|---|---|---|---|---|---|---|
| Dataset | # Train | # Valid | # Test | # Label | Task | Evaluation Metric |
| ImageNet1k | 1,281,167 | 50,000 | 100,000 | 1,000 | Classification | Accuracy |
| CIFAR-100 | 45,000 | 5,000 | 10,000 | 100 | Classification | Accuracy |
| Fashion MNIST | 54,000 | 6,000 | 10,000 | 10 | Classification | Accuracy |
| Oxford Flowers | 1,020 | 1,020 | 6,150 | 102 | Classification | Accuracy |

**MM-Vet:** is a benchmark intended to evaluate large multimodal models on complex tasks that require the integration of multiple vision-language capabilities. It defines six core VL skills and sixteen combinations of these skills, and employs an LLM-based evaluator to provide a unified scoring metric across diverse question types and answer formats. MM-Vet enables a systematic assessment of models' generalization, reasoning, and open-ended answer generation abilities.

By using both VizWiz-VQA and MM-Vet, we comprehensively evaluate our pruning method across real-world visual questions, complex multimodal reasoning, and diverse answer styles, providing a thorough assessment of its impact on the overall quality of the pruned model. Note that our evaluation is conducted on a subset of the datasets. To illustrate the nature of the datasets used in our evaluation, we provide example entries from both VizWiz-VQA and MM-Vet.

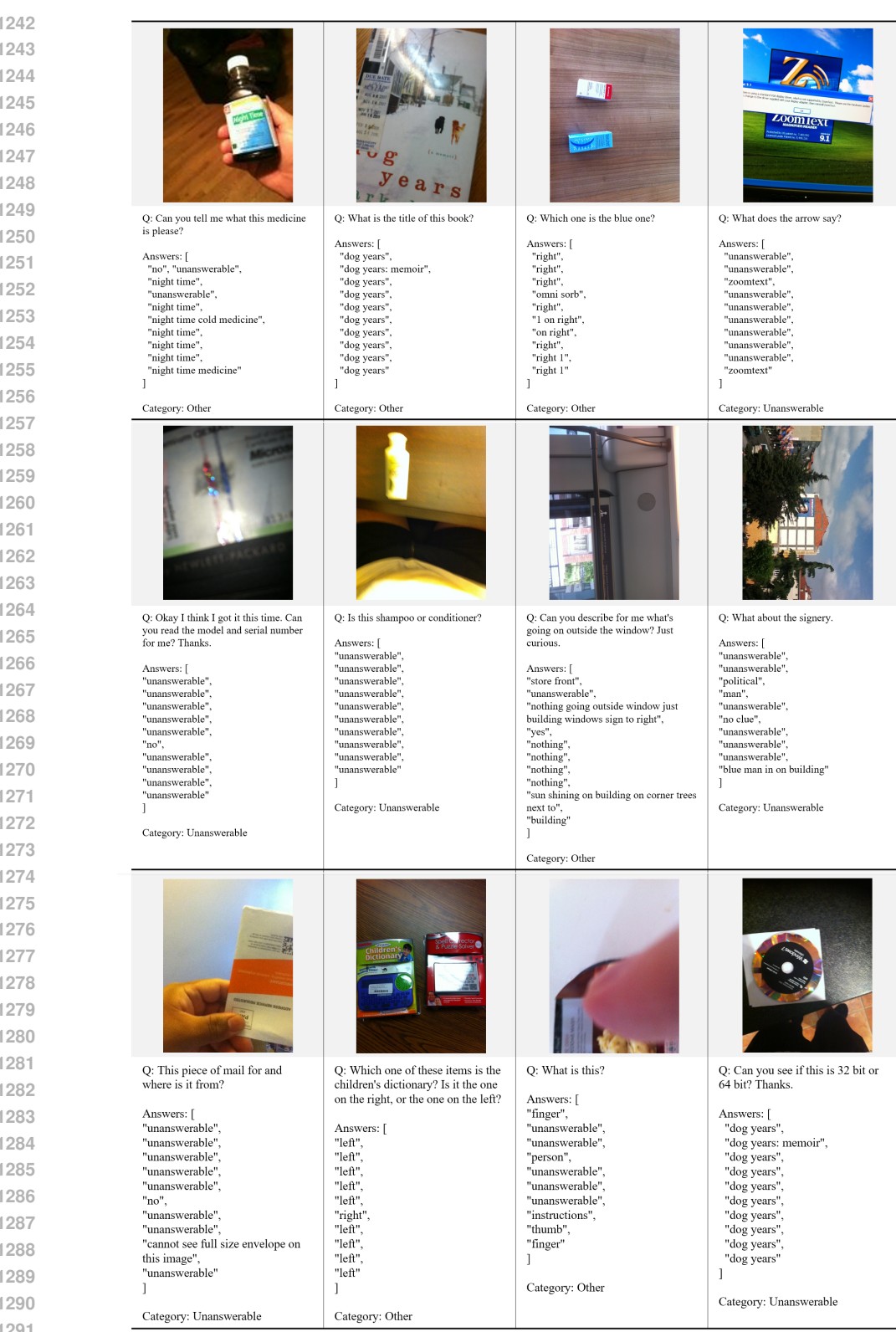

Figure 5: Examples from VizWiz-VQA showing visual questions asked by blind users and the corresponding answers from crowd workers. The examples include both questions that can be answered from the image and questions that cannot.

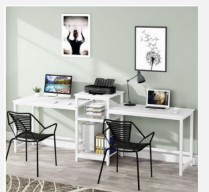

**Q:** How many gallons of supreme gasoline can I get with $50?

**GT:** 13.6 <OR> 13.7

**Required capabilities:** OCR, math

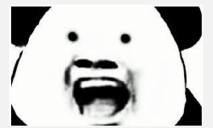

**Q:** On the right desk, what is to the left of the laptop?

**GT:** table lamp <OR> desk lamp

**Required capabilities:** Recognition, spatial awareness

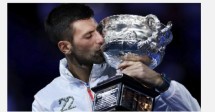

**Q:** What occasions would someone use this meme?

**GT:** This meme, commonly known as "Screaming Panda," is typically used to express shock, surprise, or fear. It could be used in response to a startling or unexpected event, or to convey a sense of panic or alarm. Some possible occasions where someone might use this meme include:- Reacting to a jump scare in a horror movie- Responding to a surprising plot twist in a TV show or book- Expressing shock at a news headline or current event- Conveying fear or anxiety about an upcoming deadline or exam- Showing surprise at an unexpected outcome in a sports game or other competition.

**Required capabilities:** Recognition, knowledge, language generation

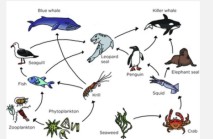

**Q:** In which country was this photo taken?

**GT:** Australia

**Required capabilities:** Recognition, knowledge

**Q:** Which are producers in this food web?
**GT:** Phytoplankton <AND> Seaweed
**Required capabilities:** OCR, knowledge, spatial awareness

**Q:** The table below gives information about the underground railway systems in six cities. Summarize the information by selecting and reporting the main features and make comparisons where relevant. You should write at least 150 words.

**GT:** The table shows data about the underground rail networks in six major cities. The table compares the six networks in terms of their age, size and the number of people who use them each year. It is clear that the three oldest underground systems are larger and serve significantly more passengers than the newer systems. The London underground is the oldest system, having opened in 1863. It is also the largest system, with 394 kilometers of route. The second largest system, in Paris, is only about half the size of the London underground, with 199 kilometers of route. However, it serves more people per year. While only third in terms of size, the Tokyo system is easily the most used, with 1927 million passengers per year. Of the three newer networks, the Washington DC underground is the most extensive, with 126 kilometers of route, compared to only 11 kilometers and 28 kilometers for the Kyoto and Los Angeles systems. The Los Angeles network is the newest, having opened in 2001, while the Kyoto network is the smallest and serves only 45million passengers per year.

**Required capabilities:** OCR, language generation, spatial awareness

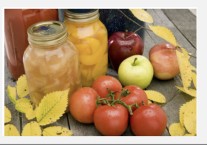

**Q:** How many tomatoes are there?
**GT:** 5
**Required capabilities:** Recognition

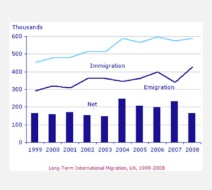

**Q:** The graph below shows the long-term international migration, UK, 1999-2008.Summarize the information by selecting and reporting the main features and make comparisons where relevant. You should write at least 150 words.

**GT:** The chart gives information about UK immigration, emigration and net migration between 1999 and 2008.Both immigration and emigration rates rose over the period shown, but the figures for immigration were significantly higher. Net migration peaked in 2004 and 2007.In 1999, over 450,000 people came to live in the UK, while the number of people who emigrated stood at just under 300,000. The figure for net migration was around160,000, and it remained at a similar level until 2003. From 1999 to 2004, the immigration rate rose by nearly 150,000 people, but there was a much smaller rise in emigration. Net migration peaked at almost 250,000 people in 2004. After 2004, the rate of immigration remained high, but the number of people emigrating fluctuated. Emigration fell suddenly in 2007, before peaking at about 420,000 people in 2008. As a result, the net migration figure rose to around 240,000in 2007, but fell back to around 160,000 in 2008.

**Required capabilities:** Recognition, OCR, language generation, spatial awareness

Figure 6: Eight example queries from the MM-Vet benchmark, each requiring different integrations of core vision–language capabilities to solve complicated multimodal tasks.

# D  ADDITIONAL EXPERIMENTAL RESULTS

## D.1  ORTHOGONALITY ANALYSIS

We provide an empirical sanity check supporting the assumptions above. Experiments use TinyBERT on SST-2 (`Vishnou/TinyBERT_SST2`). For each head we compute layerwise-normalized HIS and attention-entropy (AE) scores, stack them into vectors $u$ and $v$, and form the centered versions $\tilde{u}$ and $\tilde{v}$ by subtracting each vector's mean (a finite-sample proxy for projection onto the zero-sum subspace). The following sample statistics were obtained:

$$\widehat{\mathrm{Cov}}(\tilde{u}, \tilde{v}) = 0.030853, \qquad \overline{u} = 3.73 \times 10^{-9}, \qquad \overline{v} = 1.61 \times 10^{-8}.$$

Consequently,

$$\widehat{\mathbb{E}}\big[\langle \tilde{u}, -\tilde{v} \rangle\big] \;=\; -0.030853 \;=\; -\big(\widehat{\mathrm{Cov}}(\tilde{u}, \tilde{v}) + \overline{u}\,\overline{v}\big) \quad \text{(up to numerical precision)}.$$

The covariance magnitude is small on this batch, indicating weak coupling between the two directions and lending empirical support to the "*(near) uncorrelatedness*" assumption. We recommend reporting the same diagnostics averaged over multiple batches to reduce sampling noise and to provide confidence intervals.

## D.2 HEATMAP OF IMPORTANCE SCORES AND PRUNING RESULTS

**CoLA**

HIS

| | H0 | H1 | H2 | H3 | H4 | H5 | H6 | H7 | H8 | H9 | H10 | H11 |
|---|---|---|---|---|---|---|---|---|---|---|---|---|
| L0 | 0.04 | 0.03 | 0.01 | 0.01 | 0.02 | 0.05 | 0.06 | 0.02 | 0.04 | 0.04 | 0.02 | 0.02 |
| L1 | 0.05 | 0.06 | 0.05 | 0.05 | 0.07 | 0.04 | 0.07 | 0.05 | 0.03 | 0.04 | 0.03 | 0.08 |
| L2 | 0.13 | 0.08 | 0.11 | 0.07 | 0.06 | 0.12 | 0.10 | 0.06 | 0.10 | 0.11 | 0.12 | 0.12 |
| L3 | 0.09 | 0.15 | 0.11 | 0.11 | 0.12 | 0.17 | 0.10 | 0.15 | 0.27 | 0.19 | 0.12 | 0.16 |
| L4 | 0.20 | 0.33 | 0.17 | 0.09 | 0.13 | 0.22 | 0.13 | 0.14 | 0.11 | 0.14 | 0.21 | 0.22 |
| L5 | 0.24 | 0.10 | 0.15 | 0.21 | 0.17 | 0.10 | 0.13 | 0.23 | 0.16 | 0.37 | 0.39 | 0.12 |
| L6 | 0.21 | 0.23 | 0.33 | 0.34 | 0.22 | 0.32 | 0.29 | 0.22 | 0.30 | 0.23 | 0.29 | 0.43 |
| L7 | 0.26 | 0.30 | 0.49 | 0.15 | 0.39 | 0.27 | 0.15 | 0.22 | 0.42 | 0.43 | 0.59 | 0.25 |
| L8 | 0.22 | 0.25 | 0.30 | 0.23 | 0.34 | 0.50 | 0.29 | 0.37 | 0.62 | 0.43 | 0.39 | 0.33 |
| L9 | 0.15 | 0.29 | 0.40 | 0.26 | 0.24 | 0.60 | 0.24 | 0.13 | 0.55 | 0.11 | 0.47 | 0.32 |
| L10 | 0.89 | 0.64 | 0.49 | 0.75 | 0.51 | 0.69 | 0.35 | 0.90 | 0.44 | 0.62 | 0.43 | 0.60 |
| L11 | 1.00 | 0.62 | 0.81 | 0.77 | 0.63 | 0.59 | 0.86 | 0.78 | 0.71 | 0.78 | 0.78 | 0.73 |

HIES (Ours)

| | H0 | H1 | H2 | H3 | H4 | H5 | H6 | H7 | H8 | H9 | H10 | H11 |
|---|---|---|---|---|---|---|---|---|---|---|---|---|
| L0 | 0.01 | 0.06 | 0.35 | 0.31 | 0.13 | 0.09 | 0.07 | 0.07 | 0.04 | 0.07 | 0.40 | 0.22 |
| L1 | 0.19 | 0.22 | 0.14 | 0.22 | 0.25 | 0.12 | 0.51 | 0.16 | 0.07 | 0.07 | 0.13 | 0.09 |
| L2 | 0.54 | 0.36 | 0.28 | 0.14 | 0.24 | 0.20 | 0.18 | 0.21 | 0.19 | 0.55 | 0.19 | 0.28 |
| L3 | 0.37 | 0.20 | 0.24 | 0.27 | 0.17 | 0.51 | 0.22 | 0.19 | 0.28 | 0.34 | 0.24 | 0.33 |
| L4 | 0.32 | 0.22 | 0.33 | 0.36 | 0.24 | 0.44 | 0.36 | 0.45 | 0.35 | 0.29 | 0.37 | 0.33 |
| L5 | 0.53 | 0.49 | 0.38 | 0.44 | 0.31 | 0.51 | 0.51 | 0.54 | 0.48 | 0.62 | 0.52 | 0.48 |
| L6 | 0.42 | 0.41 | 0.47 | 0.57 | 0.47 | 0.51 | 0.44 | 0.40 | 0.47 | 0.44 | 0.49 | 0.57 |
| L7 | 0.36 | 0.38 | 0.46 | 0.45 | 0.45 | 0.36 | 0.43 | 0.43 | 0.42 | 0.51 | 0.56 | 0.47 |
| L8 | 0.30 | 0.30 | 0.40 | 0.26 | 0.34 | 0.47 | 0.40 | 0.26 | 0.39 | 0.34 | 0.36 | 0.33 |
| L9 | 0.32 | 0.31 | 0.27 | 0.26 | 0.22 | 0.40 | 0.44 | 0.36 | 0.48 | 0.33 | 0.28 | 0.33 |
| L10 | 0.47 | 0.52 | 0.29 | 0.46 | 0.35 | 0.48 | 0.29 | 0.50 | 0.34 | 0.51 | 0.46 | 0.35 |
| L11 | 0.60 | 0.37 | 0.44 | 0.45 | 0.41 | 0.40 | 0.45 | 0.50 | 0.47 | 0.43 | 0.45 | 0.46 |

**MRPC**

HIS

| | H0 | H1 | H2 | H3 | H4 | H5 | H6 | H7 | H8 | H9 | H10 | H11 |
|---|---|---|---|---|---|---|---|---|---|---|---|---|
| L0 | 0.04 | 0.04 | 0.02 | 0.02 | 0.03 | 0.09 | 0.08 | 0.03 | 0.06 | 0.05 | 0.03 | 0.02 |
| L1 | 0.13 | 0.18 | 0.12 | 0.17 | 0.27 | 0.13 | 0.13 | 0.16 | 0.11 | 0.10 | 0.11 | 0.20 |
| L2 | 0.53 | 0.27 | 0.20 | 0.16 | 0.16 | 0.25 | 0.38 | 0.18 | 0.31 | 0.42 | 0.21 | 0.50 |
| L3 | 0.34 | 0.42 | 0.23 | 0.40 | 0.41 | 0.43 | 0.25 | 0.31 | 0.40 | 0.50 | 0.35 | 0.40 |
| L4 | 0.47 | 0.32 | 0.23 | 0.44 | 0.28 | 0.52 | 0.21 | 0.24 | 0.31 | 0.42 | 0.35 | 0.27 |
| L5 | 0.65 | 0.30 | 0.28 | 0.36 | 0.48 | 0.31 | 0.28 | 0.49 | 0.78 | 0.65 | 0.53 | 0.25 |
| L6 | 0.32 | 0.27 | 0.39 | 0.31 | 0.74 | 0.60 | 0.57 | 0.23 | 0.74 | 0.47 | 0.33 | 0.72 |
| L7 | 0.49 | 0.29 | 0.77 | 0.16 | 0.85 | 0.42 | 0.18 | 0.13 | 0.52 | 0.55 | 0.67 | 0.44 |
| L8 | 0.29 | 0.39 | 0.75 | 0.66 | 0.34 | 0.40 | 0.53 | 0.34 | 0.23 | 1.00 | 0.34 | 0.56 |
| L9 | 0.70 | 0.23 | 0.22 | 0.43 | 0.21 | 0.32 | 0.98 | 0.28 | 0.54 | 0.65 | 0.23 | 0.57 |
| L10 | 0.44 | 0.37 | 0.33 | 0.41 | 0.21 | 0.19 | 0.24 | 0.56 | 0.26 | 0.37 | 0.51 | 0.32 |
| L11 | 0.32 | 0.15 | 0.39 | 0.10 | 0.27 | 0.46 | 0.10 | 0.11 | 0.89 | 0.43 | 0.11 | 0.29 |

HIES (Ours)

| | H0 | H1 | H2 | H3 | H4 | H5 | H6 | H7 | H8 | H9 | H10 | H11 |
|---|---|---|---|---|---|---|---|---|---|---|---|---|
| L0 | 0.04 | 0.05 | 0.17 | 0.18 | 0.02 | 0.08 | 0.04 | 0.02 | 0.08 | 0.03 | 0.21 | 0.11 |
| L1 | 0.14 | 0.27 | 0.10 | 0.16 | 0.35 | 0.12 | 0.29 | 0.15 | 0.09 | 0.07 | 0.15 | 0.19 |
| L2 | 0.64 | 0.31 | 0.21 | 0.12 | 0.16 | 0.22 | 0.36 | 0.16 | 0.28 | 0.56 | 0.17 | 0.43 |
| L3 | 0.40 | 0.34 | 0.23 | 0.39 | 0.34 | 0.50 | 0.23 | 0.25 | 0.34 | 0.53 | 0.32 | 0.40 |
| L4 | 0.45 | 0.28 | 0.26 | 0.47 | 0.24 | 0.50 | 0.25 | 0.31 | 0.33 | 0.35 | 0.35 | 0.32 |
| L5 | 0.60 | 0.36 | 0.27 | 0.37 | 0.40 | 0.35 | 0.35 | 0.50 | 0.68 | 0.68 | 0.53 | 0.31 |
| L6 | 0.39 | 0.33 | 0.41 | 0.38 | 0.67 | 0.57 | 0.56 | 0.28 | 0.62 | 0.48 | 0.39 | 0.73 |
| L7 | 0.49 | 0.34 | 0.70 | 0.28 | 0.83 | 0.45 | 0.28 | 0.26 | 0.51 | 0.55 | 0.63 | 0.45 |
| L8 | 0.35 | 0.40 | 0.72 | 0.58 | 0.39 | 0.42 | 0.51 | 0.39 | 0.30 | 0.84 | 0.40 | 0.54 |
| L9 | 0.65 | 0.32 | 0.32 | 0.42 | 0.32 | 0.39 | 0.88 | 0.38 | 0.51 | 0.61 | 0.32 | 0.53 |
| L10 | 0.40 | 0.42 | 0.37 | 0.42 | 0.31 | 0.27 | 0.32 | 0.52 | 0.31 | 0.41 | 0.46 | 0.38 |
| L11 | 0.37 | 0.25 | 0.38 | 0.23 | 0.33 | 0.46 | 0.23 | 0.23 | 0.78 | 0.40 | 0.23 | 0.34 |

**QNLI**

HIS

| | H0 | H1 | H2 | H3 | H4 | H5 | H6 | H7 | H8 | H9 | H10 | H11 |
|---|---|---|---|---|---|---|---|---|---|---|---|---|
| L0 | 0.02 | 0.02 | 0.01 | 0.01 | 0.02 | 0.04 | 0.06 | 0.02 | 0.04 | 0.03 | 0.01 | 0.02 |
| L1 | 0.23 | 0.24 | 0.17 | 0.26 | 0.35 | 0.19 | 0.16 | 0.25 | 0.16 | 0.16 | 0.24 | 0.32 |
| L2 | 0.81 | 0.51 | 0.33 | 0.21 | 0.27 | 0.41 | 0.67 | 0.27 | 0.48 | 0.65 | 0.35 | 0.69 |
| L3 | 0.58 | 0.62 | 0.34 | 0.63 | 0.27 | 0.72 | 0.38 | 0.46 | 0.70 | 0.84 | 0.68 | 0.66 |
| L4 | 0.55 | 0.41 | 0.36 | 0.52 | 0.34 | 0.76 | 0.34 | 0.36 | 0.52 | 0.48 | 0.43 | 0.33 |
| L5 | 0.71 | 0.26 | 0.36 | 0.36 | 0.46 | 0.46 | 0.36 | 0.49 | 0.70 | 0.79 | 0.68 | 0.30 |
| L6 | 0.41 | 0.34 | 0.58 | 0.27 | 0.74 | 0.70 | 0.54 | 0.39 | 0.86 | 0.60 | 0.38 | 0.70 |
| L7 | 0.66 | 0.38 | 0.47 | 0.20 | 0.70 | 0.45 | 0.16 | 0.16 | 0.57 | 0.62 | 0.77 | 0.21 |
| L8 | 0.28 | 0.62 | 0.33 | 0.51 | 0.29 | 0.48 | 0.51 | 0.50 | 0.28 | 0.65 | 0.78 | 0.72 |
| L9 | 0.44 | 0.37 | 0.31 | 0.45 | 0.20 | 0.28 | 0.56 | 0.38 | 0.69 | 0.19 | 0.31 | 0.58 |
| L10 | 0.31 | 0.16 | 0.58 | 0.43 | 0.26 | 0.09 | 0.24 | 0.32 | 0.35 | 0.49 | 0.59 | 0.73 |
| L11 | 0.31 | 0.35 | 0.50 | 0.30 | 0.44 | 0.61 | 0.29 | 0.30 | 1.00 | 0.44 | 0.29 | 0.46 |

HIES (Ours)

| | H0 | H1 | H2 | H3 | H4 | H5 | H6 | H7 | H8 | H9 | H10 | H11 |
|---|---|---|---|---|---|---|---|---|---|---|---|---|
| L0 | 0.53 | 0.55 | 0.15 | 0.14 | 0.63 | 0.56 | 0.65 | 0.61 | 0.51 | 0.64 | 0.09 | 0.32 |
| L1 | 0.55 | 0.42 | 0.54 | 0.59 | 0.46 | 0.64 | 0.17 | 0.68 | 0.69 | 0.73 | 0.56 | 0.76 |
| L2 | 0.51 | 0.65 | 0.66 | 0.74 | 0.63 | 0.78 | 0.89 | 0.71 | 0.79 | 0.41 | 0.79 | 0.90 |
| L3 | 0.66 | 0.93 | 0.68 | 0.79 | 0.63 | 0.63 | 0.74 | 0.89 | 0.96 | 0.78 | 0.87 | 0.78 |
| L4 | 0.76 | 0.75 | 0.67 | 0.63 | 0.77 | 0.79 | 0.59 | 0.49 | 0.72 | 0.81 | 0.66 | 0.50 |
| L5 | 0.75 | 0.37 | 0.56 | 0.54 | 0.78 | 0.58 | 0.42 | 0.53 | 0.77 | 0.65 | 0.72 | 0.45 |
| L6 | 0.52 | 0.58 | 0.70 | 0.38 | 0.77 | 0.74 | 0.63 | 0.70 | 1.00 | 0.66 | 0.50 | 0.63 |
| L7 | 0.75 | 0.54 | 0.56 | 0.35 | 0.63 | 0.55 | 0.29 | 0.29 | 0.65 | 0.66 | 0.73 | 0.38 |
| L8 | 0.48 | 0.85 | 0.48 | 0.69 | 0.47 | 0.61 | 0.65 | 0.69 | 0.55 | 0.76 | 0.83 | 0.77 |
| L9 | 0.63 | 0.54 | 0.56 | 0.64 | 0.42 | 0.49 | 0.69 | 0.52 | 0.80 | 0.45 | 0.55 | 0.84 |
| L10 | 0.55 | 0.34 | 0.76 | 0.64 | 0.49 | 0.29 | 0.46 | 0.56 | 0.51 | 0.63 | 0.69 | 0.88 |
| L11 | 0.58 | 0.63 | 0.73 | 0.54 | 0.70 | 0.77 | 0.57 | 0.56 | 0.98 | 0.66 | 0.58 | 0.67 |

**QQP**

HIS

| | H0 | H1 | H2 | H3 | H4 | H5 | H6 | H7 | H8 | H9 | H10 | H11 |
|---|---|---|---|---|---|---|---|---|---|---|---|---|
| L0 | 0.03 | 0.04 | 0.01 | 0.01 | 0.02 | 0.04 | 0.05 | 0.02 | 0.04 | 0.04 | 0.02 | 0.02 |
| L1 | 0.27 | 0.39 | 0.27 | 0.24 | 0.41 | 0.19 | 0.12 | 0.34 | 0.22 | 0.19 | 0.25 | 0.38 |
| L2 | 0.83 | 0.52 | 0.20 | 0.22 | 0.25 | 0.38 | 0.63 | 0.25 | 0.38 | 0.70 | 0.29 | 0.57 |
| L3 | 0.86 | 0.57 | 0.27 | 0.51 | 0.27 | 0.63 | 0.27 | 0.33 | 0.69 | 0.94 | 0.62 | 0.59 |
| L4 | 0.57 | 0.38 | 0.14 | 0.74 | 0.23 | 0.75 | 0.34 | 0.42 | 0.29 | 0.39 | 0.26 | 0.30 |
| L5 | 0.69 | 0.22 | 0.25 | 0.37 | 0.54 | 0.43 | 0.30 | 0.48 | 0.64 | 0.61 | 0.60 | 0.25 |
| L6 | 0.28 | 0.14 | 0.35 | 0.23 | 0.47 | 0.37 | 0.43 | 0.36 | 0.40 | 0.36 | 0.36 | 0.51 |
| L7 | 0.59 | 0.31 | 0.45 | 0.27 | 0.81 | 0.29 | 0.26 | 0.20 | 0.21 | 0.60 | 0.56 | 0.27 |
| L8 | 0.34 | 0.27 | 0.20 | 0.28 | 0.20 | 0.32 | 0.41 | 0.39 | 0.22 | 0.37 | 0.55 | 0.61 |
| L9 | 0.48 | 0.42 | 0.33 | 0.43 | 0.18 | 0.40 | 0.51 | 0.63 | 0.76 | 0.25 | 0.28 | 0.42 |
| L10 | 0.11 | 0.15 | 0.40 | 0.28 | 0.14 | 0.09 | 0.17 | 0.48 | 0.27 | 0.40 | 0.73 | 0.41 |
| L11 | 0.32 | 0.22 | 0.38 | 0.25 | 0.37 | 0.77 | 0.27 | 0.22 | 1.00 | 0.63 | 0.26 | 0.59 |

HIES (Ours)

| | H0 | H1 | H2 | H3 | H4 | H5 | H6 | H7 | H8 | H9 | H10 | H11 |
|---|---|---|---|---|---|---|---|---|---|---|---|---|
| L0 | 0.02 | 0.03 | 0.04 | 0.04 | 0.01 | 0.03 | 0.04 | 0.01 | 0.04 | 0.03 | 0.05 | 0.03 |
| L1 | 0.27 | 0.39 | 0.25 | 0.24 | 0.41 | 0.17 | 0.15 | 0.32 | 0.20 | 0.17 | 0.24 | 0.36 |
| L2 | 0.84 | 0.51 | 0.20 | 0.20 | 0.24 | 0.36 | 0.61 | 0.24 | 0.36 | 0.71 | 0.28 | 0.54 |
| L3 | 0.85 | 0.54 | 0.26 | 0.50 | 0.25 | 0.63 | 0.26 | 0.31 | 0.66 | 0.92 | 0.60 | 0.57 |
| L4 | 0.56 | 0.36 | 0.15 | 0.73 | 0.22 | 0.73 | 0.37 | 0.42 | 0.29 | 0.37 | 0.26 | 0.30 |
| L5 | 0.68 | 0.23 | 0.24 | 0.37 | 0.52 | 0.43 | 0.32 | 0.48 | 0.63 | 0.61 | 0.59 | 0.25 |
| L6 | 0.28 | 0.15 | 0.35 | 0.24 | 0.46 | 0.37 | 0.43 | 0.35 | 0.39 | 0.37 | 0.37 | 0.52 |
| L7 | 0.58 | 0.31 | 0.45 | 0.28 | 0.80 | 0.29 | 0.27 | 0.21 | 0.22 | 0.59 | 0.55 | 0.27 |
| L8 | 0.34 | 0.27 | 0.21 | 0.28 | 0.20 | 0.32 | 0.40 | 0.38 | 0.23 | 0.38 | 0.54 | 0.60 |
| L9 | 0.48 | 0.41 | 0.33 | 0.43 | 0.19 | 0.39 | 0.51 | 0.62 | 0.75 | 0.25 | 0.28 | 0.41 |
| L10 | 0.13 | 0.17 | 0.40 | 0.29 | 0.16 | 0.11 | 0.19 | 0.47 | 0.28 | 0.41 | 0.72 | 0.41 |
| L11 | 0.32 | 0.22 | 0.37 | 0.26 | 0.36 | 0.75 | 0.27 | 0.23 | 0.97 | 0.61 | 0.26 | 0.58 |

Figure 7: Heatmaps of head-importance scores across four GLUE tasks (CoLA, MRPC, QNLI, QQP). Left: HIS; Right: HIES (ours). Rows = layers (L0–L11); columns = heads (H0–H11).

We analyze the pruning patterns and performance dynamics of HIS- and HIES-based methods across varying sparsity levels. This section highlights the fundamental distinctions in head selection strategies and the underlying mechanisms responsible for the observed performance inversion.

## D.3 3D ANALYSIS OF ATTENTION HEAD IMPORTANCE SCORES

Figure 8: 3D Analysis of Attention Head Importance Scores

### D.3.1 DIFFERENCE IN PRUNING PATTERNS

Pruning heatmaps (Fig. 9) reveal systematic differences between the methods. HIS-based pruning tends to remove heads primarily from the lower layers, producing an approximately bottom-up pattern consistent with its one-step gradient saliency. In contrast, HIES yields a more dispersed selection spanning lower, middle, and upper layers. We attribute this to the entropy-aware term, which leverages structural properties of the attention distribution (concentration vs. dispersion) in addition to gradient sensitivity, thereby promoting diversity across layers in pruning decisions.

### D.3.2 PERFORMANCE INVERSION ACROSS SPARSITY REGIMES

We identify two distinct pruning regimes:

**Redundancy Regime ($\leq 10\%$ pruning).** In the early pruning phase, the model contains a substantial number of redundant heads. Here, gradient-based importance scores (HIS) are sufficient to identify and remove low-sensitivity heads, as they reflect the immediate (one-step) loss change. Consequently, HIS performs slightly better than HIES in both accuracy and stability under light pruning.

**Specialization Regime ($\geq 30\%$ pruning).** As pruning becomes more aggressive, redundant heads are mostly exhausted, and specialized heads begin to be targeted. In this regime, HIS alone struggles to distinguish critical heads from less important ones, as gradient magnitudes no longer capture long-term utility. In contrast, HIES leverages attention entropy to preferentially preserve highly concentrated (low-entropy) heads—which are typically more specialized—and prune high-entropy, less task-specific heads. This leads to superior accuracy and stability under higher pruning ratios.

**Summary**

- **Pruning $\leq 10\%$:** Redundancy regime $\Rightarrow$ HIS outperforms HIES.
- **Pruning $\geq 30\%$:** Specialization regime $\Rightarrow$ HIES outperforms HIS.

These findings demonstrate that HIS and HIES prioritize head preservation differently—HIS reflects short-horizon gradient sensitivity, whereas HIES incorporates extended inference-time stability by preserving low-entropy specialized heads.

**CoLA**

**Pruning Ratio**

**HIS** (left) / **HIES (Ours)** (right)

### 10%

**HIS**

| | H0 | H1 | H2 | H3 | H4 | H5 | H6 | H7 | H8 | H9 | H10 | H11 |
|---|---|---|---|---|---|---|---|---|---|---|---|---|
| L0 | 0.03 | 0.02 | 0.00 | 0.00 | 0.01 | 0.04 | 0.05 | 0.01 | 0.03 | 0.03 | 0.01 | 0.01 |
| L1 | 0.04 | 0.05 | 0.03 | 0.04 | 0.06 | 0.03 | 0.05 | 0.04 | 0.02 | 0.03 | 0.02 | 0.06 |
| L2 | 0.12 | 0.07 | 0.10 | 0.06 | 0.05 | 0.11 | 0.09 | 0.05 | 0.09 | 0.10 | 0.11 | 0.11 |
| L3 | 0.08 | 0.14 | 0.10 | 0.10 | 0.11 | 0.16 | 0.09 | 0.14 | 0.26 | 0.18 | 0.11 | 0.15 |
| L4 | 0.19 | 0.32 | 0.16 | 0.08 | 0.12 | 0.21 | 0.12 | 0.13 | 0.10 | 0.13 | 0.20 | 0.21 |
| L5 | 0.23 | 0.09 | 0.14 | 0.20 | 0.16 | 0.09 | 0.12 | 0.23 | 0.15 | 0.36 | 0.39 | 0.11 |
| L6 | 0.20 | 0.22 | 0.33 | 0.33 | 0.21 | 0.31 | 0.28 | 0.21 | 0.29 | 0.22 | 0.28 | 0.42 |
| L7 | 0.26 | 0.29 | 0.48 | 0.14 | 0.38 | 0.26 | 0.14 | 0.21 | 0.42 | 0.42 | 0.59 | 0.24 |
| L8 | 0.21 | 0.24 | 0.30 | 0.22 | 0.33 | 0.50 | 0.28 | 0.36 | 0.61 | 0.43 | 0.38 | 0.32 |
| L9 | 0.14 | 0.28 | 0.39 | 0.25 | 0.23 | 0.60 | 0.23 | 0.12 | 0.55 | 0.10 | 0.46 | 0.31 |
| L10 | 0.89 | 0.64 | 0.48 | 0.74 | 0.51 | 0.68 | 0.34 | 0.89 | 0.43 | 0.62 | 0.42 | 0.60 |
| L11 | 1.00 | 0.62 | 0.81 | 0.77 | 0.62 | 0.59 | 0.86 | 0.78 | 0.71 | 0.78 | 0.78 | 0.73 |

**HIES (Ours)**

| | H0 | H1 | H2 | H3 | H4 | H5 | H6 | H7 | H8 | H9 | H10 | H11 |
|---|---|---|---|---|---|---|---|---|---|---|---|---|
| L0 | 0.01 | 0.06 | 0.35 | 0.31 | 0.13 | 0.09 | 0.07 | 0.07 | 0.04 | 0.07 | 0.40 | 0.22 |
| L1 | 0.19 | 0.22 | 0.14 | 0.22 | 0.25 | 0.12 | 0.51 | 0.16 | 0.07 | 0.07 | 0.13 | 0.09 |
| L2 | 0.54 | 0.36 | 0.28 | 0.14 | 0.24 | 0.20 | 0.18 | 0.21 | 0.19 | 0.55 | 0.19 | 0.28 |
| L3 | 0.37 | 0.20 | 0.24 | 0.27 | 0.17 | 0.51 | 0.22 | 0.19 | 0.28 | 0.34 | 0.24 | 0.33 |
| L4 | 0.32 | 0.22 | 0.33 | 0.36 | 0.24 | 0.44 | 0.36 | 0.45 | 0.35 | 0.29 | 0.37 | 0.33 |
| L5 | 0.53 | 0.49 | 0.38 | 0.44 | 0.31 | 0.51 | 0.51 | 0.54 | 0.48 | 0.62 | 0.52 | 0.48 |
| L6 | 0.42 | 0.41 | 0.47 | 0.57 | 0.47 | 0.51 | 0.44 | 0.40 | 0.47 | 0.44 | 0.49 | 0.57 |
| L7 | 0.36 | 0.38 | 0.46 | 0.45 | 0.54 | 0.36 | 0.43 | 0.43 | 0.42 | 0.51 | 0.56 | 0.47 |
| L8 | 0.30 | 0.30 | 0.40 | 0.26 | 0.34 | 0.47 | 0.40 | 0.26 | 0.39 | 0.34 | 0.36 | 0.33 |
| L9 | 0.32 | 0.31 | 0.27 | 0.26 | 0.22 | 0.40 | 0.44 | 0.36 | 0.48 | 0.33 | 0.28 | 0.33 |
| L10 | 0.47 | 0.52 | 0.29 | 0.46 | 0.35 | 0.48 | 0.29 | 0.50 | 0.34 | 0.51 | 0.46 | 0.35 |
| L11 | 0.60 | 0.37 | 0.44 | 0.45 | 0.41 | 0.40 | 0.45 | 0.50 | 0.47 | 0.43 | 0.45 | 0.46 |

### 30%

**HIS**

| | H0 | H1 | H2 | H3 | H4 | H5 | H6 | H7 | H8 | H9 | H10 | H11 |
|---|---|---|---|---|---|---|---|---|---|---|---|---|
| L0 | 0.03 | 0.02 | 0.00 | 0.00 | 0.01 | 0.04 | 0.05 | 0.01 | 0.03 | 0.03 | 0.01 | 0.01 |
| L1 | 0.04 | 0.05 | 0.03 | 0.04 | 0.06 | 0.03 | 0.05 | 0.04 | 0.02 | 0.03 | 0.02 | 0.06 |
| L2 | 0.12 | 0.07 | 0.10 | 0.06 | 0.05 | 0.11 | 0.09 | 0.05 | 0.09 | 0.10 | 0.11 | 0.11 |
| L3 | 0.08 | 0.14 | 0.10 | 0.10 | 0.11 | 0.16 | 0.09 | 0.14 | 0.26 | 0.18 | 0.11 | 0.15 |
| L4 | 0.19 | 0.32 | 0.16 | 0.08 | 0.12 | 0.21 | 0.12 | 0.13 | 0.10 | 0.13 | 0.20 | 0.21 |
| L5 | 0.23 | 0.09 | 0.14 | 0.20 | 0.16 | 0.09 | 0.12 | 0.23 | 0.15 | 0.36 | 0.39 | 0.11 |
| L6 | 0.20 | 0.22 | 0.33 | 0.33 | 0.21 | 0.31 | 0.28 | 0.21 | 0.29 | 0.22 | 0.28 | 0.42 |
| L7 | 0.26 | 0.29 | 0.48 | 0.14 | 0.38 | 0.26 | 0.14 | 0.21 | 0.42 | 0.42 | 0.59 | 0.24 |
| L8 | 0.21 | 0.24 | 0.30 | 0.22 | 0.33 | 0.50 | 0.28 | 0.36 | 0.61 | 0.43 | 0.38 | 0.32 |
| L9 | 0.14 | 0.28 | 0.39 | 0.25 | 0.23 | 0.60 | 0.23 | 0.12 | 0.55 | 0.10 | 0.46 | 0.31 |
| L10 | 0.89 | 0.64 | 0.48 | 0.74 | 0.51 | 0.68 | 0.34 | 0.89 | 0.43 | 0.62 | 0.42 | 0.60 |
| L11 | 1.00 | 0.62 | 0.81 | 0.77 | 0.62 | 0.59 | 0.86 | 0.78 | 0.71 | 0.78 | 0.78 | 0.73 |

**HIES (Ours)**

| | H0 | H1 | H2 | H3 | H4 | H5 | H6 | H7 | H8 | H9 | H10 | H11 |
|---|---|---|---|---|---|---|---|---|---|---|---|---|
| L0 | 0.01 | 0.06 | 0.35 | 0.31 | 0.13 | 0.09 | 0.07 | 0.07 | 0.04 | 0.07 | 0.40 | 0.22 |
| L1 | 0.19 | 0.22 | 0.14 | 0.22 | 0.25 | 0.12 | 0.51 | 0.16 | 0.07 | 0.07 | 0.13 | 0.09 |
| L2 | 0.54 | 0.36 | 0.28 | 0.14 | 0.24 | 0.20 | 0.18 | 0.21 | 0.19 | 0.55 | 0.19 | 0.28 |
| L3 | 0.37 | 0.20 | 0.24 | 0.27 | 0.17 | 0.51 | 0.22 | 0.19 | 0.28 | 0.34 | 0.24 | 0.33 |
| L4 | 0.32 | 0.22 | 0.33 | 0.36 | 0.24 | 0.44 | 0.36 | 0.45 | 0.35 | 0.29 | 0.37 | 0.33 |
| L5 | 0.53 | 0.49 | 0.38 | 0.44 | 0.31 | 0.51 | 0.51 | 0.54 | 0.48 | 0.62 | 0.52 | 0.48 |
| L6 | 0.42 | 0.41 | 0.47 | 0.57 | 0.47 | 0.51 | 0.44 | 0.40 | 0.47 | 0.44 | 0.49 | 0.57 |
| L7 | 0.36 | 0.38 | 0.46 | 0.45 | 0.54 | 0.36 | 0.43 | 0.43 | 0.42 | 0.51 | 0.56 | 0.47 |
| L8 | 0.30 | 0.30 | 0.40 | 0.26 | 0.34 | 0.47 | 0.40 | 0.26 | 0.39 | 0.34 | 0.36 | 0.33 |
| L9 | 0.32 | 0.31 | 0.27 | 0.26 | 0.22 | 0.40 | 0.44 | 0.36 | 0.48 | 0.33 | 0.28 | 0.33 |
| L10 | 0.47 | 0.52 | 0.29 | 0.46 | 0.35 | 0.48 | 0.29 | 0.50 | 0.34 | 0.51 | 0.46 | 0.35 |
| L11 | 0.60 | 0.37 | 0.44 | 0.45 | 0.41 | 0.40 | 0.45 | 0.50 | 0.47 | 0.43 | 0.45 | 0.46 |

### 50%

**HIS**

| | H0 | H1 | H2 | H3 | H4 | H5 | H6 | H7 | H8 | H9 | H10 | H11 |
|---|---|---|---|---|---|---|---|---|---|---|---|---|
| L0 | 0.03 | 0.02 | 0.00 | 0.00 | 0.01 | 0.04 | 0.05 | 0.01 | 0.03 | 0.03 | 0.01 | 0.01 |
| L1 | 0.04 | 0.05 | 0.03 | 0.04 | 0.06 | 0.03 | 0.05 | 0.04 | 0.02 | 0.03 | 0.02 | 0.06 |
| L2 | 0.12 | 0.07 | 0.10 | 0.06 | 0.05 | 0.11 | 0.09 | 0.05 | 0.09 | 0.10 | 0.11 | 0.11 |
| L3 | 0.08 | 0.14 | 0.10 | 0.10 | 0.11 | 0.16 | 0.09 | 0.14 | 0.26 | 0.18 | 0.11 | 0.15 |
| L4 | 0.19 | 0.32 | 0.16 | 0.08 | 0.12 | 0.21 | 0.12 | 0.13 | 0.10 | 0.13 | 0.20 | 0.21 |
| L5 | 0.23 | 0.09 | 0.14 | 0.20 | 0.16 | 0.09 | 0.12 | 0.23 | 0.15 | 0.36 | 0.39 | 0.11 |
| L6 | 0.20 | 0.22 | 0.33 | 0.33 | 0.21 | 0.31 | 0.28 | 0.21 | 0.29 | 0.22 | 0.28 | 0.42 |
| L7 | 0.26 | 0.29 | 0.48 | 0.14 | 0.38 | 0.26 | 0.14 | 0.21 | 0.42 | 0.42 | 0.59 | 0.24 |
| L8 | 0.21 | 0.24 | 0.30 | 0.22 | 0.33 | 0.50 | 0.28 | 0.36 | 0.61 | 0.43 | 0.38 | 0.32 |
| L9 | 0.14 | 0.28 | 0.39 | 0.25 | 0.23 | 0.60 | 0.23 | 0.12 | 0.55 | 0.10 | 0.46 | 0.31 |
| L10 | 0.89 | 0.64 | 0.48 | 0.74 | 0.51 | 0.68 | 0.34 | 0.89 | 0.43 | 0.62 | 0.42 | 0.60 |
| L11 | 1.00 | 0.62 | 0.81 | 0.77 | 0.62 | 0.59 | 0.86 | 0.78 | 0.71 | 0.78 | 0.78 | 0.73 |

**HIES (Ours)**

| | H0 | H1 | H2 | H3 | H4 | H5 | H6 | H7 | H8 | H9 | H10 | H11 |
|---|---|---|---|---|---|---|---|---|---|---|---|---|
| L0 | 0.01 | 0.06 | 0.35 | 0.31 | 0.13 | 0.09 | 0.07 | 0.07 | 0.04 | 0.07 | 0.40 | 0.22 |
| L1 | 0.19 | 0.22 | 0.14 | 0.22 | 0.25 | 0.12 | 0.51 | 0.16 | 0.07 | 0.07 | 0.13 | 0.09 |
| L2 | 0.54 | 0.36 | 0.28 | 0.14 | 0.24 | 0.20 | 0.18 | 0.21 | 0.19 | 0.55 | 0.19 | 0.28 |
| L3 | 0.37 | 0.20 | 0.24 | 0.27 | 0.17 | 0.51 | 0.22 | 0.19 | 0.28 | 0.34 | 0.24 | 0.33 |
| L4 | 0.32 | 0.22 | 0.33 | 0.36 | 0.24 | 0.44 | 0.36 | 0.45 | 0.35 | 0.29 | 0.37 | 0.33 |
| L5 | 0.53 | 0.49 | 0.38 | 0.44 | 0.31 | 0.51 | 0.51 | 0.54 | 0.48 | 0.62 | 0.52 | 0.48 |
| L6 | 0.42 | 0.41 | 0.47 | 0.57 | 0.47 | 0.51 | 0.44 | 0.40 | 0.47 | 0.44 | 0.49 | 0.57 |
| L7 | 0.36 | 0.38 | 0.46 | 0.45 | 0.54 | 0.36 | 0.43 | 0.43 | 0.42 | 0.51 | 0.56 | 0.47 |
| L8 | 0.30 | 0.30 | 0.40 | 0.26 | 0.34 | 0.47 | 0.40 | 0.26 | 0.39 | 0.34 | 0.36 | 0.33 |
| L9 | 0.32 | 0.31 | 0.27 | 0.26 | 0.22 | 0.40 | 0.44 | 0.36 | 0.48 | 0.33 | 0.28 | 0.33 |
| L10 | 0.47 | 0.52 | 0.29 | 0.46 | 0.35 | 0.48 | 0.29 | 0.50 | 0.34 | 0.51 | 0.46 | 0.35 |
| L11 | 0.60 | 0.37 | 0.44 | 0.45 | 0.41 | 0.40 | 0.45 | 0.50 | 0.47 | 0.43 | 0.45 | 0.46 |

### 70%

**HIS**

| | H0 | H1 | H2 | H3 | H4 | H5 | H6 | H7 | H8 | H9 | H10 | H11 |
|---|---|---|---|---|---|---|---|---|---|---|---|---|
| L0 | 0.03 | 0.02 | 0.00 | 0.00 | 0.01 | 0.04 | 0.05 | 0.01 | 0.03 | 0.03 | 0.01 | 0.01 |
| L1 | 0.04 | 0.05 | 0.03 | 0.04 | 0.06 | 0.03 | 0.05 | 0.04 | 0.02 | 0.03 | 0.02 | 0.06 |
| L2 | 0.12 | 0.07 | 0.10 | 0.06 | 0.05 | 0.11 | 0.09 | 0.05 | 0.09 | 0.10 | 0.11 | 0.11 |
| L3 | 0.08 | 0.14 | 0.10 | 0.10 | 0.11 | 0.16 | 0.09 | 0.14 | 0.26 | 0.18 | 0.11 | 0.15 |
| L4 | 0.19 | 0.32 | 0.16 | 0.08 | 0.12 | 0.21 | 0.12 | 0.13 | 0.10 | 0.13 | 0.20 | 0.21 |
| L5 | 0.23 | 0.09 | 0.14 | 0.20 | 0.16 | 0.09 | 0.12 | 0.23 | 0.15 | 0.36 | 0.39 | 0.11 |
| L6 | 0.20 | 0.22 | 0.33 | 0.33 | 0.21 | 0.31 | 0.28 | 0.21 | 0.29 | 0.22 | 0.28 | 0.42 |
| L7 | 0.26 | 0.29 | 0.48 | 0.14 | 0.38 | 0.26 | 0.14 | 0.21 | 0.42 | 0.42 | 0.59 | 0.24 |
| L8 | 0.21 | 0.24 | 0.30 | 0.22 | 0.33 | 0.50 | 0.28 | 0.36 | 0.61 | 0.43 | 0.38 | 0.32 |
| L9 | 0.14 | 0.28 | 0.39 | 0.25 | 0.23 | 0.60 | 0.23 | 0.12 | 0.55 | 0.10 | 0.46 | 0.31 |
| L10 | 0.89 | 0.64 | 0.48 | 0.74 | 0.51 | 0.68 | 0.34 | 0.89 | 0.43 | 0.62 | 0.42 | 0.60 |
| L11 | 1.00 | 0.62 | 0.81 | 0.77 | 0.62 | 0.59 | 0.86 | 0.78 | 0.71 | 0.78 | 0.78 | 0.73 |

**HIES (Ours)**

| | H0 | H1 | H2 | H3 | H4 | H5 | H6 | H7 | H8 | H9 | H10 | H11 |
|---|---|---|---|---|---|---|---|---|---|---|---|---|
| L0 | 0.01 | 0.06 | 0.35 | 0.31 | 0.13 | 0.09 | 0.07 | 0.07 | 0.04 | 0.07 | 0.40 | 0.22 |
| L1 | 0.19 | 0.22 | 0.14 | 0.22 | 0.25 | 0.12 | 0.51 | 0.16 | 0.07 | 0.07 | 0.13 | 0.09 |
| L2 | 0.54 | 0.36 | 0.28 | 0.14 | 0.24 | 0.20 | 0.18 | 0.21 | 0.19 | 0.55 | 0.19 | 0.28 |
| L3 | 0.37 | 0.20 | 0.24 | 0.27 | 0.17 | 0.51 | 0.22 | 0.19 | 0.28 | 0.34 | 0.24 | 0.33 |
| L4 | 0.32 | 0.22 | 0.33 | 0.36 | 0.24 | 0.44 | 0.36 | 0.45 | 0.35 | 0.29 | 0.37 | 0.33 |
| L5 | 0.53 | 0.49 | 0.38 | 0.44 | 0.31 | 0.51 | 0.51 | 0.54 | 0.48 | 0.62 | 0.52 | 0.48 |
| L6 | 0.42 | 0.41 | 0.47 | 0.57 | 0.47 | 0.51 | 0.44 | 0.40 | 0.47 | 0.44 | 0.49 | 0.57 |
| L7 | 0.36 | 0.38 | 0.46 | 0.45 | 0.54 | 0.36 | 0.43 | 0.43 | 0.42 | 0.51 | 0.56 | 0.47 |
| L8 | 0.30 | 0.30 | 0.40 | 0.26 | 0.34 | 0.47 | 0.40 | 0.26 | 0.39 | 0.34 | 0.36 | 0.33 |
| L9 | 0.32 | 0.31 | 0.27 | 0.26 | 0.22 | 0.40 | 0.44 | 0.36 | 0.48 | 0.33 | 0.28 | 0.33 |
| L10 | 0.47 | 0.52 | 0.29 | 0.46 | 0.35 | 0.48 | 0.29 | 0.50 | 0.34 | 0.51 | 0.46 | 0.35 |
| L11 | 0.60 | 0.37 | 0.44 | 0.45 | 0.41 | 0.40 | 0.45 | 0.50 | 0.47 | 0.43 | 0.45 | 0.46 |

Figure 9: CoLA: heatmaps of head importance and pruning across sparsity levels. For each pruning ratio (10%, 30%, 50%, 70%), we show HIS (left) and HIES (right). Rows = layers (L0–L11); columns = heads (H0–H11). Dark/grey cells mark heads pruned at the target ratio.

## D.4 EXPERIMENTAL RESULTS ON DOWNSTREAM TASKS

Fig. 10 reports downstream evaluations of HIES versus the HIS baseline on CIFAR-100, Food-101, and Fashion-MNIST. Across all three benchmarks, HIES consistently sustains higher accuracy under aggressive pruning, whereas HIS exhibits rapid degradation once the pruning ratio exceeds 20%. On CIFAR-100, HIS collapses beyond moderate sparsity, while HIES exhibits slower degradation and retains substantially higher accuracy relative to HIS even at 40–50%. Food-101 reveals a similar trend, with HIES delivering substantial and consistent gains over HIS across all pruning levels. On Fashion-MNIST, HIS undergoes steep drops after 20% pruning, in contrast to the stable performance of HIES up to 50%. These results demonstrate that HIES reliably mitigates sharp-drop phenomena and delivers robust, stable improvements over HIS across heterogeneous downstream tasks.

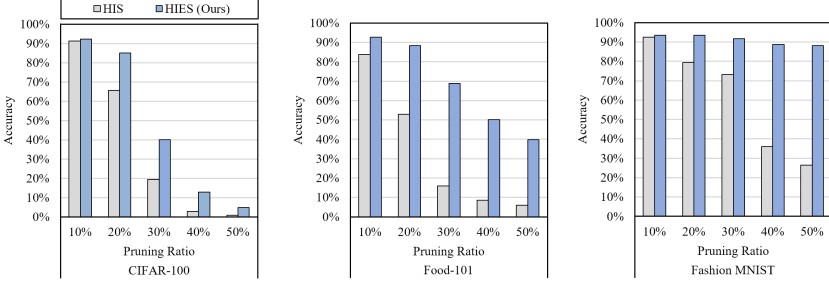

Figure 10: Evaluation of HIES on the image classification benchmarks. HIES consistently outperforms baseline, demonstrating robust and stable performance across downstream tasks.

## D.5 SENSITIVITY ANALYSIS - ABLATION ON $\alpha$

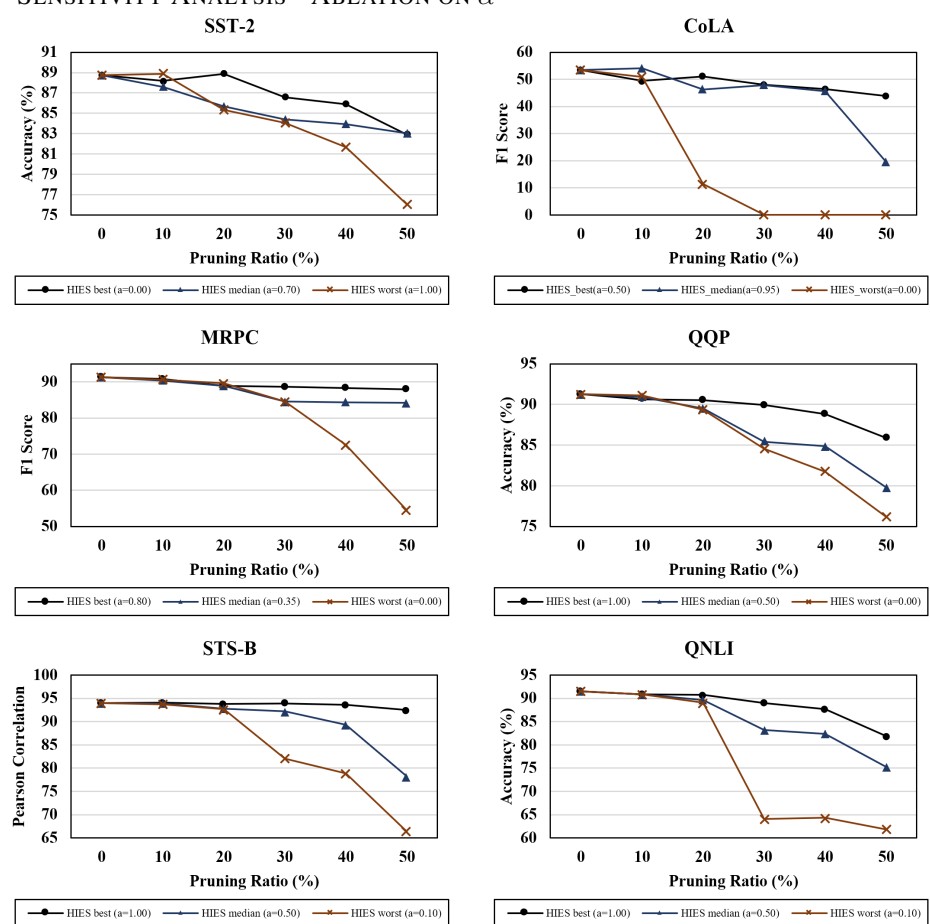

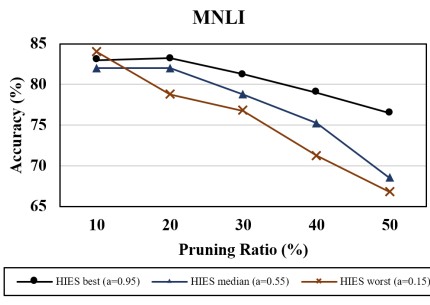 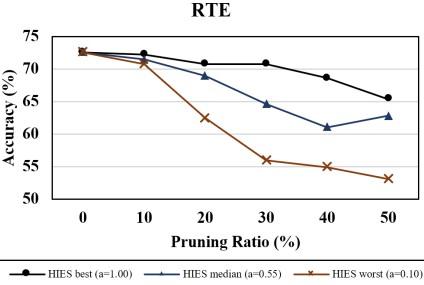

Figure 11: HIES sensitivity to the mixing coefficient $\alpha$ on GLUE. For each task, we sweep $\alpha$ and report three choices—$\alpha_{\text{best}}$, $\alpha_{\text{median}}$, $\alpha_{\text{worst}}$—selected by weighted AUC (wAUC) across pruning ratios. Curves plot performance versus pruning ratio for these three settings.

We sweep the mixing coefficient $\alpha \in [0, 1)$ that interpolates the gradient-based head-importance (HIS) and attention-entropy (AE) signals in HIES,

$$\text{HIES}_h(\alpha) = \alpha \widehat{\text{HIS}}_h + (1 - \alpha) \widehat{\text{AE}}_h.$$

As expected, larger $\alpha$ upweights HIS and preserves heads with strong task relevance, whereas smaller $\alpha$ upweights AE and retains low-entropy, focused heads. We choose a single $\alpha^\star$ on a held-out validation split and fix it for all reported experiments; the resulting accuracy–sparsity profiles are shown in Fig. 11.

