# A  REBUTTAL APPENDIX

**Additional Baseline Evaluation** (in response to *Reviewer pQ3d, Reviewer FG9H*)

We additionally compare HIES with existing head-pruning methods, including:

1. Voita et al. (2019), *Analyzing Multi-Head Self-Attention* (ACL 2019):
   - **Stage 1 (3 epochs):** Train gate (mask) parameters with BERT weights frozen using $L_0$ regularization.
   - **Stage 2 (9 epochs total):** For each pruning ratio (10%, 30%, 50%), fine-tune BERT independently for 3 epochs with the fixed mask.
2. Li et al. (2021), *Differentiable Subset Pruning of Transformer Heads* (TACL 2021):
   - **Joint DSP:** 3 epochs of joint optimization of mask parameters $w$ and BERT weights.
   - **Pipelined DSP:**
     - **Stage 1:** Train $w$ for 1 epoch.
     - **Stage 2:** Fine-tune BERT for 3 epochs with fixed masks.

Table R.1: Experimental results with BERT$_\text{base}$ on natural language understanding task. We report percentage improvements in blue. For each task, the best-performing method is shown in **bold**, and the second-best is underlined. Note that we include methods requiring additional training in the experiments, but exclude them solely when determining the **bold**/underline rankings to ensure a fair comparison.

| Pruning Ratio (%) | Method | SST-2 Accuracy | CoLA Matthews corr | MRPC F1 Score | MNLI Accuracy | QQP Pearson corr | QNLI Accuracy | STS-B Accuracy | RTE Accuracy | Average | |
|---|---|---|---|---|---|---|---|---|---|---|---|
| | Random | 92.09 | 75.25 | 89.90 | 91.00 | 93.88 | 89.87 | 82.62 | 70.76 | 85.67 | 0.00% |
| | AD | 92.55 | 75.48 | 90.56 | 91.10 | 93.65 | 90.87 | 83.50 | 70.40 | 86.01 | +0.40% |
| | HIS | 91.74 | **77.21** | 90.65 | 91.23 | **94.04** | 90.63 | **84.04** | **71.84** | 86.42 | +0.88% |
| | L2 | 90.37 | 74.00 | 83.78 | 69.23 | 83.12 | 60.81 | 67.75 | 49.82 | 72.36 | -15.54% |
| | Joint DSP | 91.97 | 77.65 | 89.33 | 77.68 | 81.73 | 89.46 | 86.57 | 59.93 | 81.79 | -4.53% |
| | Pipelined DSP (w/o tune) | 91.86 | 78.50 | 88.00 | 77.94 | 81.60 | 88.17 | 87.88 | 63.54 | 82.19 | -4.07% |
| | Pipelined DSP (w/ tune) | 92.20 | 78.53 | 89.97 | 78.61 | 81.91 | 88.69 | 88.42 | 64.98 | 82.91 | -3.22% |
| 10 | Voita.el (mask training + w/o tune) | 90.48 | 77.73 | 89.08 | 66.28 | 83.79 | 89.04 | 83.49 | 69.68 | 81.20 | -5.22% |
| | Voita.el(mask training + w/ tune) | 92.66 | 77.35 | 89.90 | 84.00 | 90.43 | 91.29 | 85.67 | 71.48 | 85.35 | -0.38% |
| | LLM-Pruner (Channel) | 91.97 | 76.05 | 79.19 | 83.53 | 93.06 | 87.39 | 80.46 | 67.51 | 82.40 | -3.82% |
| | LLM-Pruner (Block) | 91.06 | 76.69 | 89.83 | 84.96 | 93.77 | 87.42 | 82.03 | 64.62 | 83.80 | -2.19% |
| | SliceGPT (w/o tune) | 51.38 | 49.86 | 81.46 | 63.18 | 58.92 | 53.91 | 36.84 | 49.46 | 55.63 | -35.07% |
| | SliceGPT (w/ tune) | 86.47 | 61.87 | 82.30 | 88.28 | 62.47 | 83.45 | 77.34 | 54.51 | 74.59 | -12.94% |
| | **HIES (ours)** | **92.66** | 75.48 | **91.04** | **91.93** | 94.00 | **91.03** | **84.04** | **71.84** | **86.50** | **+0.97%** |
| | Random | 90.29 | 69.02 | 85.27 | 84.00 | 92.90 | 79.80 | 75.15 | 60.29 | 79.59 | 0.00% |
| | AD | 86.58 | 50.00 | 84.53 | 84.57 | 81.94 | 67.82 | 77.00 | 56.68 | 73.64 | -7.47% |
| | HIS | 89.56 | 73.17 | **89.37** | 89.95 | 93.82 | 89.04 | 82.25 | 68.59 | 84.47 | +6.13% |
| | L2 | 86.58 | 67.52 | 81.58 | 64.83 | 76.52 | 51.09 | 56.00 | 50.90 | 66.75 | -16.10% |
| | Joint DSP | 92.09 | 79.33 | 87.27 | 73.51 | 80.18 | 87.61 | 86.55 | 58.48 | 80.63 | +1.30% |
| | Pipelined DSP (w/o tune) | 91.51 | 75.07 | 84.34 | 76.54 | 78.50 | 84.09 | 86.50 | 61.73 | 79.79 | +0.25% |
| | Pipelined DSP (w/ tune) | 92.09 | 78.50 | 89.88 | 78.46 | 81.82 | 88.89 | 88.09 | 64.98 | 82.84 | +4.08% |
| 30 | Voita.el (mask training + w/o tune) | 89.56 | 77.26 | 89.47 | 68.17 | 83.63 | 88.71 | 85.67 | 64.98 | 80.93 | +1.69% |
| | Voita.el (mask training + w/ tune) | 92.20 | 77.61 | 90.49 | 83.76 | 90.26 | 90.65 | 88.83 | 72.92 | 85.84 | +7.85% |
| | LLM-Pruner (Channel) | 88.53 | 70.36 | 86.89 | 81.23 | 92.50 | 67.38 | 66.67 | 64.26 | 77.23 | -2.97% |
| | LLM-Pruner (Block) | 88.99 | 73.93 | 84.76 | 80.09 | 93.40 | 82.68 | 78.33 | 66.79 | 81.12 | +1.92% |
| | SliceGPT (w/o tune) | 50.80 | 53.29 | 78.05 | 63.18 | 54.64 | 53.18 | 34.90 | 51.99 | 55.00 | -30.90% |
| | SliceGPT (w/ tune) | 83.49 | 60.14 | 81.80 | 85.80 | 60.89 | 67.36 | 75.50 | 54.87 | 71.23 | -10.49% |
| | **HIES (ours)** | **91.86** | 74.97 | 88.81 | **90.37** | **93.89** | **89.13** | **82.50** | **70.04** | **85.20** | **+7.04%** |
| | Random | 78.74 | 61.02 | 72.53 | 66.25 | 91.40 | 67.53 | 67.32 | 53.79 | 69.82 | 0.00% |
| | AD | 82.91 | 50.00 | 54.50 | 76.18 | 75.00 | 68.94 | 68.00 | 55.96 | 66.44 | -4.84% |
| | HIS | 87.27 | 59.48 | 86.52 | **85.91** | 92.61 | **82.68** | 78.67 | 62.82 | 79.50 | +13.84% |
| | L2 | 82.80 | 60.98 | 85.30 | 64.83 | 69.76 | 50.54 | 44.42 | 47.29 | 63.24 | -9.39% |
| | Joint DSP | 90.94 | 76.26 | 82.40 | 71.02 | 78.52 | 81.16 | 86.68 | 58.48 | 78.18 | +11.98% |
| | Pipelined DSP (w/o tune) | 90.48 | 69.35 | 86.08 | 69.70 | 76.09 | 78.91 | 84.21 | 60.65 | 76.93 | +10.19% |
| | Pipelined DSP (w/ tune) | 91.28 | 79.28 | 89.88 | 77.51 | 81.53 | 87.79 | 87.78 | 63.54 | 82.32 | +17.91% |
| 50 | Voita.el (mask training + w/o tune) | 89.56 | 78.49 | 89.44 | 69.89 | 84.07 | 88.28 | 88.34 | 68.23 | 82.04 | +17.50% |
| | Voita.el (mask training + w/ tune) | 92.09 | 77.25 | 89.46 | 83.00 | 89.72 | 90.72 | 88.78 | 68.59 | 84.95 | +21.67% |
| | LLM-Pruner (Channel) | 86.47 | 61.64 | 83.92 | 81.47 | 89.74 | 60.66 | 67.42 | 62.82 | 74.14 | +6.18% |
| | LLM-Pruner (Block) | 87.84 | 70.09 | 83.84 | 78.80 | **92.69** | 72.60 | 73.60 | 61.73 | 77.65 | +11.20% |
| | SliceGPT (w/o tune) | 50.92 | 52.79 | 81.22 | 63.19 | 47.70 | 50.98 | 34.93 | 50.90 | 54.08 | -22.54% |
| | SliceGPT (w/ tune) | 83.49 | 57.45 | 81.37 | 82.16 | 55.05 | 65.79 | 71.70 | 51.62 | 68.58 | -1.78% |
| | **HIES (ours)** | **90.71** | 68.52 | **86.80** | 85.73 | 92.65 | **82.68** | **79.00** | **65.34** | **81.43** | **+16.63%** |

**Large-scale Reasoning Experiments** (in response to *Reviewer x54u*)

We further provide new large-scale reasoning experiments in Figure R.1, bringing the total to 32 tasks and offering a comprehensive evaluation of HIES across diverse benchmarks and modalities.

This includes natural language understanding (GLUE: SST-2, CoLA, MRPC, QQP, STS-B, QNLI, MNLI, RTE), textual reasoning (HellaSwag, Winogrande, ARC-e, ARC-c, OpenBookQA, PIQA), image classification (ImageNet, CIFAR-100, Food-101, Fashion-MNIST, Oxford Flowers), visual question answering (VizWiz-VQA), and complex multimodal reasoning (MM-Vet). In addition, these evaluations span diverse model families and sizes, including encoder-only models, decoder-only models, vision transformers, and vision–language models.

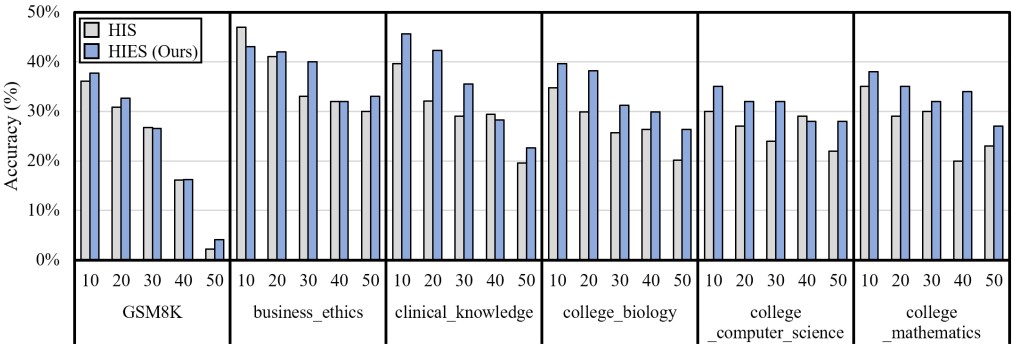

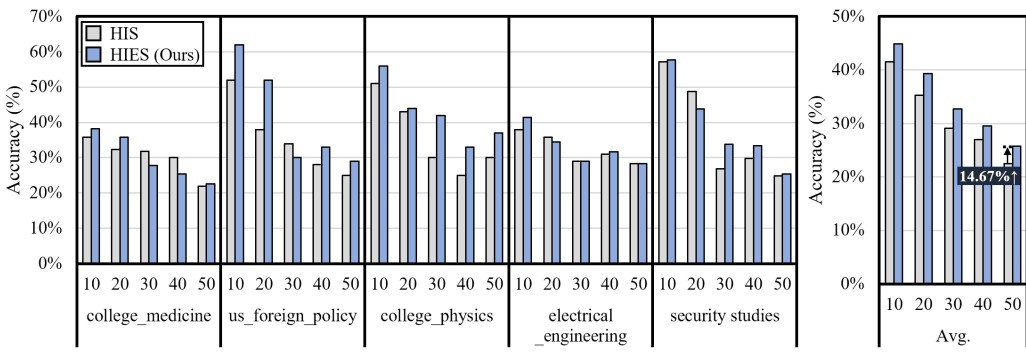

Figure R.1: Results on GSM8K (math word-problem reasoning) and MMLU (10-task knowledge reasoning) with LLaMA-2$_{7B}$.

**Comparison with SliceGPT on Decoder-Only Models** (in response to *Reviewer pQ3d*)

Figure R.2 shows the comparison between HIES and SliceGPT on decoder-only models.

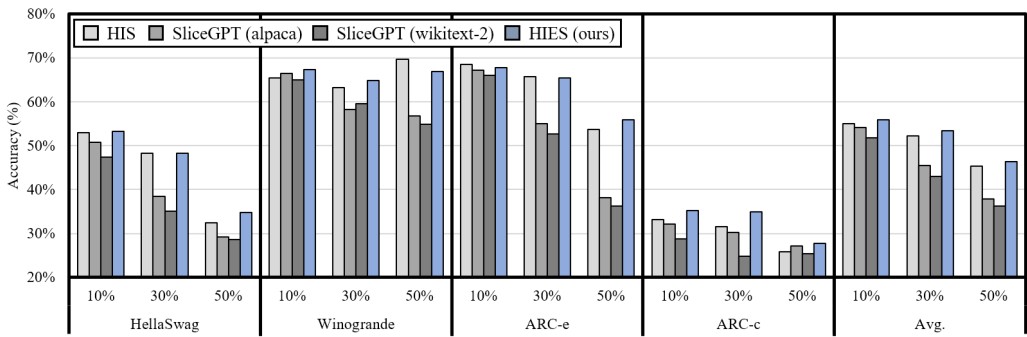

Figure R.2: Comparison of HIES and SliceGPT on LLaMA-2$_{7B}$ for HellaSwag, Winogrande, ARC-e, and ARC-c.

**Attention Entropy with L2-norm–based Importance Score** (in response to *Reviewer CUbs*)

Figure R.3 shows the impact of combining attention entropy (AE) with gradient L2-norm–based importance scores for LLaMA-2$_{7B}$ across five benchmarks: HellaSwag, Winogrande, ARC-e, ARC-c, and OBQA.

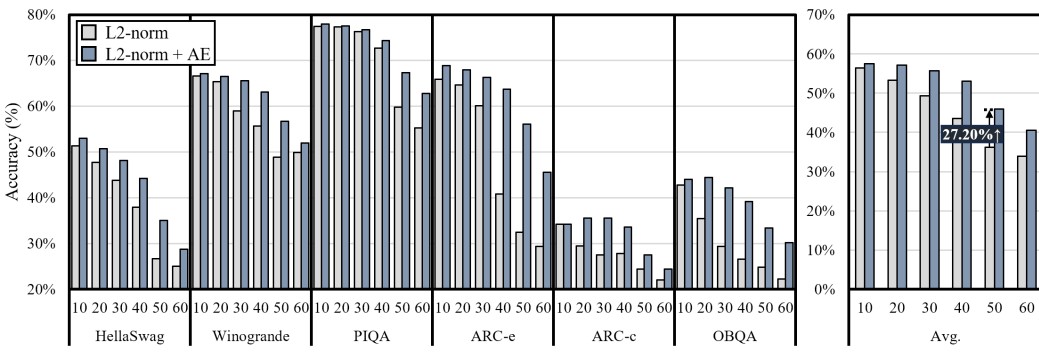

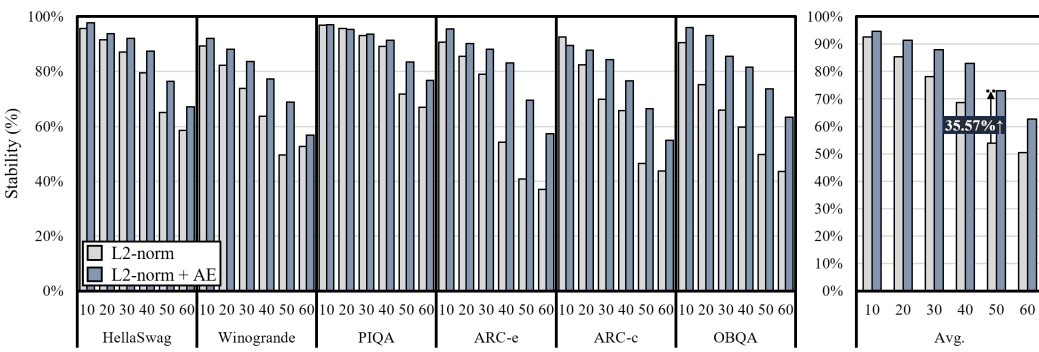

Figure R.3: Accuracy and stability improvements when combining Attention Entropy (AE) with gradient L2-norm–based importance scores, compared to using the L2-norm alone. We conduct experiments on LLaMA-2$_{7B}$ for HellaSwag, Winogrande, PIQA, ARC-e, ARC-c, and OBQA.

**Seed Sensitivity and Statistical Significance** (in response to *Reviewer CUbc, Reviewer FG9H*)

We evaluate the seed sensitivity of HIS and HIES, with results reported in Table R.2 and Table R.3 .

Table R.2: Accuracy and standard deviation in accuracy across 5 random seeds for BERT$_{base}$ on GLUE benchamrk.

| | Pruning Ratio (%) | HIS | | HIES (ours) | |
| --- | --- | --- | --- | --- | --- |
| | | Accuracy (%) | Standard Deviation | Accuracy (%) | Standard Deviation |
| CoLA | 10 | 74.18 | 1.60E-02 | 75.85 | 1.21E-02 |
| | 20 | 74.18 | 5.44E-04 | 73.04 | 1.09E-02 |
| | 30 | 72.70 | 6.91E-03 | 75.17 | 2.82E-03 |
| | 40 | 70.28 | 5.52E-03 | 70.20 | 3.68E-02 |
| | 50 | 60.37 | 2.18E-02 | 71.17 | 5.29E-03 |
| MNLI | 10 | 84.13 | 0.00E+00 | 84.06 | 1.64E-01 |
| | 20 | 82.93 | 9.13E-02 | 82.87 | 0.00E+00 |
| | 30 | 82.40 | 0.00E+00 | 82.29 | 3.58E-01 |
| | 40 | 81.46 | 2.39E-01 | 81.39 | 1.98E-01 |
| | 50 | 79.47 | 4.34E-01 | 78.87 | 3.27E-01 |
| MRPC | 10 | 90.29 | 4.58E-03 | 90.97 | 8.04E-04 |
| | 20 | 89.69 | 4.50E-03 | 89.17 | 3.55E-03 |
| | 30 | 88.79 | 2.36E-03 | 89.52 | 3.69E-03 |
| | 40 | 88.02 | 7.41E-03 | 87.12 | 9.06E-03 |
| | 50 | 86.85 | 1.65E-02 | 87.19 | 1.66E-02 |
| QNLI | 10 | 90.55 | 8.69E-02 | 90.51 | 1.34E-01 |
| | 20 | 90.23 | 3.48E-01 | 90.05 | 2.73E-01 |
| | 30 | 88.33 | 7.47E-01 | 88.15 | 7.43E-01 |
| | 40 | 84.75 | 1.45E+00 | 85.02 | 8.90E-01 |
| | 50 | 80.81 | 3.51E+00 | 82.83 | 1.74E+00 |
| QQP | 10 | 90.68 | 2.33E-02 | 91.26 | 8.00E-03 |
| | 20 | 90.56 | 1.52E-01 | 90.56 | 7.31E-02 |
| | 30 | 89.84 | 3.67E-01 | 89.72 | 3.45E-01 |
| | 40 | 88.07 | 1.84E-01 | 87.96 | 2.54E-01 |
| | 50 | 85.33 | 9.93E-01 | 85.90 | 6.97E-01 |
| RTE | 10 | 72.42 | 2.89E-01 | 72.13 | 2.70E-01 |
| | 20 | 71.91 | 5.78E-01 | 70.76 | 9.69E-01 |
| | 30 | 71.26 | 1.18E+00 | 71.34 | 1.47E+00 |
| | 40 | 67.94 | 2.10E+00 | 66.79 | 1.92E+00 |
| | 50 | 65.99 | 1.36E+00 | 65.42 | 1.56E+00 |
| SST-2 | 10 | 91.86 | 1.62E-01 | 92.64 | 4.59E-02 |
| | 20 | 91.77 | 2.22E-01 | 92.11 | 1.69E-01 |
| | 30 | 91.19 | 3.03E-01 | 92.20 | 1.92E-01 |
| | 40 | 88.67 | 1.62E+00 | 90.96 | 7.16E-01 |
| | 50 | 86.31 | 1.35E+00 | 90.55 | 1.24E+00 |
| STS-B | 10 | 87.85 | 1.88E-03 | 87.95 | 3.37E-03 |
| | 20 | 87.33 | 3.52E-03 | 87.35 | 2.06E-04 |
| | 30 | 86.82 | 2.15E-03 | 86.76 | 3.25E-03 |
| | 40 | 85.89 | 4.49E-03 | 85.87 | 4.41E-03 |
| | 50 | 83.38 | 8.95E-03 | 82.27 | 1.37E-02 |
| Avg. | 10 | 85.25 | 7.30E-02 | 85.67 | 7.98E-02 |
| | 20 | 84.82 | 1.75E-01 | 84.49 | 1.87E-01 |
| | 30 | 83.92 | 3.26E-01 | 84.39 | 3.90E-01 |
| | 40 | 81.89 | 7.00E-01 | 81.91 | 5.04E-01 |
| | 50 | 78.56 | 9.62E-01 | 80.53 | 6.99E-01 |

Table R.3: Accuracy and standard deviation in accuracy across 5 random seeds for LLaMA-2$_{7B}$ on HellaSwag, Winogrande, and ARC-c.

| Pruning Ratio (%) | HIS | | HIES (ours) | |
|---|---|---|---|---|
| | Accuracy (%) | Standard Deviation | Accuracy (%) | Standard Deviation |
| **HellaSwag** | | | | |
| 10 | 53.63 | 5.36E-01 | 53.98 0.66%↑ | 5.40E-01 |
| 20 | 51.70 | 5.17E-01 | 52.36 1.28%↑ | 5.24E-01 |
| 30 | 49.00 | 4.90E-01 | 49.84 1.71%↑ | 4.98E-01 |
| 40 | 45.27 | 4.53E-01 | 46.27 2.20%↑ | 4.63E-01 |
| 50 | 37.01 | 3.70E-01 | 36.13 2.37%↓ | 3.61E-01 |
| **Winogrande** | | | | |
| 10 | 66.38 | 6.64E-01 | 67.01 0.95%↑ | 6.70E-01 |
| 20 | 64.07 | 6.41E-01 | 66.27 3.42%↑ | 6.63E-01 |
| 30 | 63.03 | 6.30E-01 | 64.67 2.60%↑ | 6.47E-01 |
| 40 | 60.16 | 6.02E-01 | 61.06 1.50%↑ | 6.11E-01 |
| 50 | 56.16 | 5.62E-01 | 55.83 0.59%↓ | 5.58E-01 |
| **ARC-c** | | | | |
| 10 | 33.15 | 3.71E-03 | 35.05 5.73%↑ | 7.03E-03 |
| 20 | 33.15 | 5.57E-03 | 36.27 9.41%↑ | 0.00E+00 |
| 30 | 32.34 | 1.86E-03 | 35.46 9.64%↑ | 5.67E-03 |
| 40 | 29.69 | 1.86E-03 | 34.24 15.30%↑ | 1.15E-02 |
| 50 | 25.29 | 7.43E-03 | 28.95 14.48%↑ | 4.22E-02 |
| **Avg.** | | | | |
| 10 | 51.05 | 5.70E-03 | 52.01 1.88%↑ | 5.06E-03 |
| 20 | 49.64 | 5.12E-03 | 51.63 4.01%↑ | 2.72E-03 |
| 30 | 48.12 | 5.30E-03 | 49.99 3.88%↑ | 6.11E-03 |
| 40 | 45.04 | 5.49E-03 | 47.19 4.76%↑ | 8.82E-03 |
| 50 | 39.49 | 9.20E-03 | 40.30 2.07%↑ | 2.17E-02 |

**Effect of Calibration Dataset Size** (in response to *Reviewer pQ3d, Reviewer CUbs, Reviewer FG9H*)

We evaluate the effect of calibration dataset size on HIES performance, as reported in Table R.4 and 5.

Table R.4: Accuracy and standard deviation in accuracy for different calibration dataset sizes (1, 16, 32, 64, 128, 512, 1024) for BERT$_{base}$ on GLUE Benchmarks. Note that we use a default size of 32 in the main results.

| | Pruning Ratio (%) | HIS | | HIES (ours) | |
|---|---|---|---|---|---|
| | | Accuracy (%) | Standard Deviation | Accuracy (%) | Standard Deviation |
| CoLA | 10 | 74.46 | 1.33E-02 | 75.71 | 1.34E-02 |
| | 30 | 74.09 | 8.40E-03 | 75.10 | 3.04E-03 |
| | 50 | 62.01 | 3.37E-02 | 71.04 | 1.13E-02 |
| MNLI | 10 | 84.13 | 0.00E+00 | 84.13 | 0.00E+00 |
| | 30 | 82.40 | 0.00E+00 | 82.35 | 3.48E-01 |
| | 50 | 78.93 | 5.72E-01 | 78.65 | 1.98E-01 |
| MRPC | 10 | 90.52 | 2.82E-03 | 91.05 | 2.08E-03 |
| | 30 | 89.12 | 5.97E-03 | 89.61 | 2.94E-03 |
| | 50 | 86.45 | 1.55E-02 | 86.92 | 1.44E-02 |
| QNLI | 10 | 90.51 | 9.57E-02 | 90.54 | 1.25E-01 |
| | 30 | 88.25 | 5.97E-01 | 88.01 | 5.53E-01 |
| | 50 | 80.50 | 2.96E+00 | 81.97 | 1.39E+00 |
| QQP | 10 | 90.74 | 7.25E-02 | 91.24 | 4.16E-02 |
| | 30 | 90.12 | 2.31E-01 | 90.01 | 2.37E-01 |
| | 50 | 84.75 | 6.91E-01 | 85.60 | 4.29E-01 |
| RTE | 10 | 72.36 | 4.09E-01 | 72.25 | 3.25E-01 |
| | 30 | 71.17 | 1.13E+00 | 70.81 | 1.13E+00 |
| | 50 | 65.45 | 1.14E+00 | 65.50 | 2.13E+00 |
| SST-2 | 10 | 91.79 | 1.30E-01 | 92.56 | 1.68E-01 |
| | 30 | 90.78 | 4.53E-01 | 92.12 | 2.06E-01 |
| | 50 | 85.76 | 2.30E+00 | 90.33 | 1.16E+00 |
| STS-B | 10 | 87.98 | 7.08E-04 | 87.96 | 2.88E-03 |
| | 30 | 86.87 | 2.03E-03 | 86.75 | 2.64E-03 |
| | 50 | 83.53 | 6.39E-03 | 82.80 | 1.37E-02 |
| Avg. | 10 | 85.31 | 9.06E-02 | 85.68 | 8.48E-02 |
| | 30 | 84.10 | 3.04E-01 | 84.34 | 3.10E-01 |
| | 50 | 78.42 | 9.63E-01 | 80.35 | 6.68E-01 |

Table R.5: Accuracy and standard deviation in accuracy for different calibration dataset sizes (1, 16, 32, 64, 128, 512, 1024) for LLaMA-2$_{7B}$ on HellaSwag, Winogrande, and ARC-c. Note that we use a default size of 32 in the main results.

| Pruning Ratio (%) | | HIS | | HIES (ours) | |
|---|---|---|---|---|---|
| | | Accuracy (%) | Standard Deviation | Accuracy (%) | Standard Deviation |
| HellaSwag | 10 | 53.54 | 1.57E-03 | 54.09 1.02%↑ | 1.99E-03 |
| | 20 | 51.44 | 3.51E-03 | 52.37 1.82%↑ | 1.93E-03 |
| | 30 | 49.09 | 4.54E-03 | 50.22 2.30%↑ | 1.66E-03 |
| | 40 | 44.66 | 2.80E-03 | 45.59 2.09%↑ | 4.20E-03 |
| | 50 | 34.91 | 3.60E-03 | 34.43 1.36%↓ | 8.42E-03 |
| | 60 | 28.46 | 4.47E-03 | 28.91 1.57%↑ | 2.64E-03 |
| Winogrande | 10 | 65.87 | 5.51E-03 | 67.42 2.35%↑ | 3.14E-03 |
| | 20 | 64.31 | 9.97E-03 | 66.42 3.27%↑ | 4.20E-03 |
| | 30 | 63.21 | 5.59E-03 | 65.42 3.49%↑ | 5.50E-03 |
| | 40 | 60.38 | 9.65E-03 | 61.47 1.81%↑ | 5.67E-03 |
| | 50 | 56.17 | 7.89E-03 | 56.66 0.86%↑ | 5.77E-03 |
| | 60 | 51.23 | 8.90E-03 | 51.33 0.18%↑ | 1.06E-02 |
| ARC-c | 10 | 34.51 | 1.66E-02 | 34.41 0.29%↓ | 8.51E-03 |
| | 20 | 33.83 | 6.95E-03 | 35.68 5.46%↑ | 6.61E-03 |
| | 30 | 33.02 | 7.73E-03 | 34.75 5.24%↑ | 1.19E-02 |
| | 40 | 31.25 | 1.50E-02 | 33.47 7.10%↑ | 4.42E-03 |
| | 50 | 26.10 | 1.58E-02 | 26.86 2.92%↑ | 1.24E-02 |
| | 60 | 22.17 | 2.37E-02 | 23.64 6.65%↑ | 6.78E-03 |
| Avg. | 10 | 51.31 | 7.90E-03 | 51.97 1.29%↑ | 4.54E-03 |
| | 20 | 49.86 | 6.81E-03 | 51.49 3.27%↑ | 4.25E-03 |
| | 30 | 48.44 | 5.95E-03 | 50.13 3.49%↑ | 6.35E-03 |
| | 40 | 45.43 | 9.17E-03 | 46.85 3.12%↑ | 4.76E-03 |
| | 50 | 39.06 | 9.11E-03 | 39.32 0.66%↑ | 8.85E-03 |
| | 60 | 33.95 | 1.23E-02 | 34.63 1.98%↑ | 6.66E-03 |

**Extended Orthogonality Analysis** (in response to *Reviewer pQ3d*)

We provide an extended orthogonality analysis, with results shown in Table R. 6 and Table R. 7.

Table R.6: Orthogonality analysis using $BERT_{base}$ on the GLUE benchmark

| Task | $E[\tilde{u}]$ | $E[\tilde{v}]$ | $Cov(\tilde{u},\tilde{v})$ | $E[\langle\tilde{u},\tilde{v}\rangle]$ | Correlation |
|------|------|------|------|------|------|
| COLA | 6.94E-03 | 4.08E-01 | -1.40E-05 | 2.82E-03 | -1.54E-02 |
| SST2 | 6.94E-03 | 4.42E-01 | -3.60E-05 | 3.03E-03 | -3.41E-02 |
| MRPC | 6.94E-03 | 5.27E-01 | 2.41E-04 | 3.90E-03 | 2.32E-01 |
| STSB | 6.94E-03 | 4.30E-01 | -1.96E-04 | 2.79E-03 | -1.65E-01 |
| QQP | 6.94E-03 | 5.05E-01 | 1.56E-04 | 3.67E-03 | 1.95E-01 |
| MNLI | 6.94E-03 | 4.89E-01 | 1.73E-04 | 3.57E-03 | 2.27E-01 |
| QNLI | 6.94E-03 | 4.62E-01 | 1.53E-04 | 3.36E-03 | 2.23E-01 |
| RTE | 6.94E-03 | 5.05E-01 | 2.15E-04 | 3.72E-03 | 2.88E-01 |

Table R.7: Orthogonality analysis using $LLaMA-2_{7B}$ on 5

| Task | $E[\tilde{u}]$ | $E[\tilde{v}]$ | $Cov(\tilde{u},\tilde{v})$ | $E[\langle\tilde{u},\tilde{v}\rangle]$ | Correlation |
|------|------|------|------|------|------|
| HellaSwag | 9.77E-04 | 7.56E-01 | -1.47E-04 | 5.92E-04 | -1.44E-01 |
| Winogrande | 9.77E-04 | 7.66E-01 | -1.85E-04 | 5.63E-04 | -1.66E-01 |
| ARC-e | 9.77E-04 | 7.76E-01 | -1.42E-04 | 6.16E-04 | -1.44E-01 |
| ARC-c | 9.77E-04 | 7.71E-01 | -1.27E-04 | 6.26E-04 | -1.33E-01 |
| OBQA | 9.77E-04 | 8.01E-01 | -1.29E-04 | 6.53E-04 | -1.61E-01 |

**Behavior of the Mixing Coefficient $\alpha$ Under Increasing Pruning** (in response to *Reviewer pQ3d*)

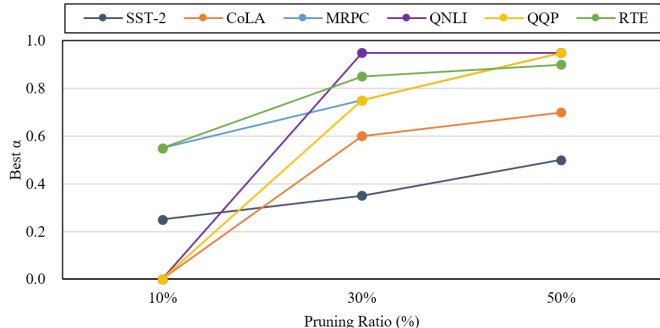

Figure R.4: Task-wise best $\alpha$ across pruning ratios on BERT with GLUE benchmarks.