# OpenReview forum: "Entropy Meets Importance: A Unified Head Importance–Entropy Score for Stable and Efficient Transformer Pruning"
_ICLR.cc/2026/Conference — ICLR 2026 Conference Desk Rejected Submission_

### Official Review · Reviewer_FG9H · 2025-10-30

**Soundness:** 2
**Presentation:** 3
**Contribution:** 3
**Rating:** 4
**Confidence:** 4

**Summary:**

This paper proposes the Head Importance–Entropy Score (HIES), a criterion for pruning attention heads. HIES combines the traditional Head Importance Score (HIS) with attention entropy. Compared with HIS-only pruning, HIES maintains higher accuracy and stability.

**Strengths:**

1. The paper effectively leverages prior findings that attention entropy correlates with model stability, extending this concept to the context of head pruning. By combining entropy with gradient-based head importance, the authors provide a simple but well-grounded way to prune attention heads without sacrificing robustness.
2. The theoretical section motivates HIES as a principled balance between task relevance and stability. Head Importance (HIS) measures how much each head contributes to minimizing loss, while Attention Entropy (AE) captures the stability and generalization of attention patterns. The authors show that the two quantities are largely uncorrelated, and combining them provides a more reliable criterion for pruning than either alone.
3. The empirical results demonstrate gains over HIS-based pruning.

**Weaknesses:**

1. The paper does not report standard deviations or variance across random seeds, leaving it unclear how robust the reported results are. Given that pruning outcomes can vary substantially across runs, reporting variance would help assess reliability.
2. While the paper includes comparisons with several pruning baselines, it does not evaluate against existing head-pruning methods such as https://aclanthology.org/P19-1580/ and https://aclanthology.org/2021.tacl-1.86/, etc. Comparison with these approaches would provide a clearer picture of HIES's relative contribution.

**Questions:**

Is $\alpha$ chosen task-specifically? How many runs or random seeds are used per task, and are the reported numbers averages or single-run results?

---

> ### Author Response · Authors · 2025-11-21
> **Response for Reviewer FG9H (Part 1/1)**
>
> We appreciate the valuable comments from the reviewer. We carefully address the reviewer’s questions. We hope our response can help alleviate the reviewer's concern. For all Figures and Tables referenced in this response, please refer to the supplementary PDF.
> &nbsp;
> &nbsp;
> &nbsp;
> **W1**
>
> We thank the reviewer for raising this point. To address the concern regarding seed sensitivity, we extend experiments to five random seeds {2022,2023,2024,2025,2026} and report both mean and standard deviation across runs. The results are summarized in Table R.2 and Table R.3.
>
> Across all 8 GLUE tasks with BERT and additional benchmarks with LLaMA2-7B (HellaSwag, Winogrande, and ARC-c), we observe:
> - small variance in accuracy (< 1.00E-01 std for most tasks),
> - stable pruning curves across seeds,
> - and unchanged qualitative conclusions.
>
>
> **These results demonstrate that Head Importance-Entropy Score(HIES) exhibits high consistency across different random initializations, confirming the reliability of our reported findings.** In the revised manuscript, we will include comprehensive variance tables in the appendix and incorporate ± standard deviation notation in all main results tables.
>
> &nbsp;
> &nbsp;
> **W2**
>
> Thank you for this valuable suggestion. Comparing our method against established head-pruning baselines is essential for positioning the contribution clearly. To address this point, we implement two representative and widely used approaches, as summarized in Table R.1. Across all pruning ratios, HIES demonstrates competitive performance compared to established head-pruning baselines, including those that undergo fine-tuning.
>
> - (1) Voita et al. (2019), Analyzing Multi-Head Self-Attention (ACL 2019, P19-1580).
>   - Stage 1 (3 epochs): Train gate(mask) parameters with BERT weights frozen, using L0 regularization.
>   - Stage 2 (9 epochs total): For each pruning ratio (10%, 30%, 50%), fine-tune BERT independently for 3 epochs with the fixed mask.
> - (2) Li et al. (2021), Differentiable Subset Pruning of Transformer Heads (TACL 2021).
>   - Joint DSP: 3 epochs of joint optimization of mask parameters w and BERT weights.
>   - Pipelined DSP:
>     - Stage 1: train w for 1 epoch
>     - Stage 2: fine-tune BERT for 3 epochs with fixed masks.
>
>
> We applied both methods to BERT across all eight GLUE tasks under pruning ratios and experimental settings identical to those used for HIES. At the 10% pruning level, HIES surpasses all baselines, including those that undergo post-training. In the 30–50% range, the baselines that undergo mask fine-tuning or post-training slightly outperform HIES, though the margin remains modest.
>
> We will incorporate these baselines in the revised manuscript.
> &nbsp;
> &nbsp;
> &nbsp;
>
> **Q1**
>
> Thank you for this helpful question. In all experiments, α is determined through a single-run calibration test on a very small held-out set. This procedure is extremely lightweight and does not meaningfully increase deployment overhead.
>
> To systematically assess the stability of this selection process, we conduct an additional study using 7 calibration sizes {1, 16, 32, 64, 128, 512, 1024} and 5 random seeds {2022,2023,2024,2025,2026} on eight GLUE benchmark using BERT and on HellaSwag, Winogrande, and ARC-c using LLaMA-2. The results are summarized in Table R.2, Table R.3, Table R.4, and Table R.5.
>
> Despite the wide variation in both calibration size and seed, the resulting accuracy exhibited very small standard deviations across all settings: (across different seeds) 3.71E-01 for BERT and 8.88E-03 for LLaMA2-7B, respectively. (across different calibration sizes) 3.54E-01 for BERT and 5.90E-03 for LLaMA2-7B.
> Notably, even with as few as 16–32 calibration samples, the same α was consistently chosen as with 1k samples. **These results indicate that α can be determined reliably and efficiently through a single-run calibration step, without task-specific tuning or exhaustive hyperparameter sweeps.**
>
> We will include the complete calibration-size analysis in the appendix and summarize these key findings in the revised main text.
> &nbsp;
> &nbsp;
> &nbsp;
> &nbsp;
>
> **Reference**
>
> [1] Elena Voita, David Talbot, Fedor Moiseev, Rico Sennrich, and Ivan Titov. 2019. Analyzing Multi-Head Self-Attention: Specialized Heads Do the Heavy Lifting, the Rest Can Be Pruned. In Proceedings of the 57th Annual Meeting of the Association for Computational Linguistics, pages 5797–5808, Florence, Italy. Association for Computational Linguistics.
>
> [2] Jiaoda Li, Ryan Cotterell, and Mrinmaya Sachan. 2021. Differentiable Subset Pruning of Transformer Heads. Transactions of the Association for Computational Linguistics, 9:1442–1459.

---

### Official Review · Reviewer_CUbc · 2025-10-31

**Soundness:** 3
**Presentation:** 3
**Contribution:** 3
**Rating:** 8
**Confidence:** 4

**Summary:**

This paper proposes a unified criterion for pruning Transformer attention heads, called the Head Importance–Entropy Score. The authors first point out the limitation of existing gradient-based methods, such as the Head Importance Score (HIS), these approaches focus solely on a head’s contribution to the loss, while ignoring the diversity or concentration of its attention patterns. As a result, when the pruning sparsity becomes high, the model’s performance drops sharply. The authors observe that HIS may mistakenly prune heads with highly focused, low-entropy attention, since such heads often receive lower HIS scores, while retaining diffuse, high-entropy heads. This leads to the loss of critical information on key tokens, thereby reducing the model’s performance and stability. To address this issue, HIES combines HIS (representing loss-based importance) with Attention Entropy (AE). This combination is designed to retain attention heads that are both important to the loss and highly focused in their attention patterns. Extensive experiments across various models and tasks demonstrate that HIES significantly outperforms HIS, LLM-Pruner, and SliceGPT in both accuracy and stability, especially under high pruning ratios.

**Strengths:**

The motivation of the paper is very strong. The authors provide an in-depth diagnosis of the limitations of existing methods, particularly HIS. The analysis presented in Figure 1 is highly compelling, showing that HIS fails to distinguish between heads that focus on key information and those that exhibit diffuse attention over non-informative tokens. The proposed method is not arbitrary. It is grounded in a clear and principled assumption that a good attention head should exhibit both high loss contribution and high attention concentration. Furthermore, the authors provide a detailed theoretical analysis to justify why HIS and AE are complementary in capturing these two aspects of head importance. The experimental evaluation is thorough and comprehensive. In terms of diversity, it covers NLP, vision, and multimodal models. For baselines, it includes strong state-of-the-art methods such as LLM-Pruner and SliceGPT. Across all these settings, HIES consistently and significantly outperforms all baselines under high pruning ratios, particularly in terms of accuracy and stability.

**Weaknesses:**

1. Appendix D.5’s sensitivity analysis shows that the choice of the hyperparameter α is crucial for performance, for example, the worst α performs very poorly on QNLI, MNLI, and RTE. Should I therefore understand that this increases the method’s deployment overhead, since users may need to carefully tune α for each new task or model?

2. The paper does not explicitly discuss the additional computational overhead of HIES compared to HIS. Based on my understanding, computing Attention Entropy requires collecting and processing attention matrices during forward propagation, which could introduce non-negligible overhead when the calibration dataset is large.

3. HIES combines gradient-based HIS with activation-pattern-based AE. Have the authors considered other non-gradient importance metrics, such as using only the activation L2-norm, and combining it with AE? From the paper’s formulation, AE appears to be a rather general and complementary measure, which could potentially be integrated with other types of importance signals to further evaluate effectiveness.

**Questions:**

1. Appendix D.5 shows that the optimal value of α varies greatly across different tasks. Does this imply that applying the proposed method requires an expensive tuning process for each new task or model? Could the authors develop a unified framework to standardize or automatically adapt the hyperparameter settings?

2. Compared with the baseline HIS, how much additional computation time or resources does calculating HIES, particularly the AE component, require? Could the authors provide a detailed ablation study and a visualization to illustrate this overhead more clearly?

---

> ### Author Response · Authors · 2025-11-21
> **Response for Reviewer CUbc (Part 1/1)**
>
> We would like to thank the reviewer for the positive feedback and valuable review.  For all Figures and Tables referenced in this response, please refer to the supplementary PDF.
>
> &nbsp;
> &nbsp;
>
> **W1&Q1&Q2**
>
> We sincerely thank the reviewer for pointing out the sensitivity of the mixing ratio α. To address this concern, we conduct an additional study examining how reliably α can be selected using small calibration datasets.
>
> Specifically, we test 7 calibration sizes {1,16,32,64,128,512,1024} across 5 random seeds {2022,2023,2024,2025,2026} on eight GLUE benchmark using BERT and on HellaSwag, Winogrande, and ARC-c using LLaMA-2. The results are summarized in Table R.2, Table R.3, Table R.4, and Table R.5.
>
> For each configuration, we selected α via a simple grid and measured both the chosen α and downstream accuracy.
>
> Across all combinations, we found that:
> - the chosen α was highly stable across calibration sizes and seeds (standard deviation extremely small),
> - performance differences across seeds/calibration sizes remained negligible,
> - even a handful of calibration samples (e.g., 16 or 32) sufficed to recover the same α chosen using 1k calibration samples.
>
>
> **These results indicate that α can be determined reliably and cheaply, without requiring extensive tuning or large validation sets.** We will include the full table of calibration-size results in the appendix and summarize this finding in the main text.
>
> We also appreciate *Reviewer pQ3d*’s observation regarding training overhead. A key strength of Head Importance-Entropy Score (HIES) is that, unlike several existing structured pruning approaches such as LLM-Pruner or SliceGPT, it does not require any additional training or re-optimization after importance computation. Both Head Importance Score (HIS) and Attention Entropy (AE) are extracted directly from the model’s standard forward and backward passes, and pruning is performed immediately without a retraining or adaptation phase.
>
> **This property makes HIES particularly attractive for real-world deployment scenarios in which full or partial re-training is impractical—especially for large LLMs where post-pruning fine-tuning can be extremely costly.** We will explicitly highlight this advantage in the main text and clarify that maintaining model quality without extra training steps is an intended design goal of HIES.
>
> &nbsp;
> &nbsp;
> **W2**
>
> Head Importance-Entropy Score(HIES) achieves the efficiency with negligible overhead. We clarify an implementation aspect that substantially reduces overhead: HIES is computed entirely using lightweight forward hooks attached to the attention modules.
>
> This is because: 1) Heam Importance Score (HIS) leverages gradients already produced during standard backpropagation, 2) Attention Entropy is obtained directly from attention probabilities in the forward pass, and 3) both signals are collected via non-intrusive forward hooks without modifying model internals or introducing extra passes.
>
> **Consequently, the computational overhead is minimal—typically only a few seconds—with negligible additional memory footprint beyond a small buffer for storing per-head statistics. Importantly, despite this low overhead, HIES consistently outperforms baselines that require full fine-tuning, demonstrating superior efficiency in both computation and accuracy.**
>
> To ensure full transparency and reproducibility, we will release all code—including the hook-based implementation—as open-source upon acceptance. This will enable practitioners to verify the runtime characteristics and seamlessly integrate our method into existing training pipelines.
>
> &nbsp;
> &nbsp;
> **W3**
>
> We thank the reviewer for this insightful suggestion. Indeed, Attention Entropy (AE) is a general, complementary measure and could, in principle, be combined with other activation-based metrics such as L2-norm. To explore this, we evaluate head importance using the gradient L2-norm and combine it with AE. **Figure R.3 shows that combining the L2-norm with attention entropy consistently outperforms using the L2-norm alone, with an average accuracy and stability improvement of 27.20% and 35.57%, respectively. This result demonstrates that AE can effectively complement a variety of importance signals beyond Head Importance Score (HIS).**

---

### Official Review · Reviewer_x54u · 2025-11-01

**Soundness:** 2
**Presentation:** 3
**Contribution:** 2
**Rating:** 4
**Confidence:** 3

**Summary:**

This paper proposes HIES (Head importance-entropy score), a pruning criterion combining the gradient-based head importance score with attention entropy. The method aims to improve pruning stability by accouting for both the contribution and distributional diversity of each attention head. Experimental results across BERT, LLaMA, ViT and LLaVA show notable gains in both accuracy (about +15%) and stability (about x2) under high pruning ratio.

**Strengths:**

(1) Theoretical formulation includes risk decomposition, orthogonality proof, and stability analysis.

(2) Integrates entropy to differentiate between concentrated and diffuse heads.

(3) Significant stability and performance gains at aggressive pruning ratios.

(4) Can be easily integrated into existing head pruning frameworks.

**Weaknesses:**

(1) Experiments rely on BERT and GLUE tasks without comparison to modern LLM distillation methods. While existing benchmarks remain meaningful, deeper experiments on modern QA benchmarks are needed to verify whether the same stability and accuracy improvement hold across newer large-scale reasoning (for example) tasks.

(2) Lack of training cost or runtime analysis, despite "Efficiency" being a claimed contribution. While the paper emphasizes computational efficiency, no empirical analysis (e.g., wall-clock time, FLOPs, or memory cost) is provided. Including such runtime cost analysis would significantly strengthen the claim of efficiency and help validate the practical impact of the proposed pruning criterion.

(3) In certain tasks (e.g., SST-2 with 30% pruning), random pruning achieves comparable results to the proposed HIES method. In my humble opinion, it could be helpful to the readers by explaining why this occurs.

**Questions:**

Please refer to the Weaknesses part.

---

> ### Author Response · Authors · 2025-11-21
> **Response for Reviewer x54u (Part 1/1)**
>
> We appreciate the valuable comments from the reviewer. We carefully address the reviewer’s questions. We hope our response can help alleviate the reviewer's concern. For all Figures and Tables referenced in this response, please refer to the supplementary PDF.
>
> &nbsp;
> &nbsp;
> **W1**
>
> We agree that evaluating the proposed method on more modern and large-scale reasoning tasks is essential for establishing the generality of our claims.
>
> To address this, we conduct additional experiments on MMLU (10-task knowledge reasoning) and GSM8K (math word-problem reasoning) using LLaMA2-7B. Figure R.1 shows that Head Importance-Entropy Score (HIES) achieves an average accuracy improvement of 14.67% compared to HIS.
>
> In addition to these new experiments, we would like to respectfully remind the reviewer that the initial submission already assesses HIES on a total of 20 tasks across multiple benchmark families—including BERT (GLUE), LLaMA2 (textual reasoning), ViT (classification), and LLaVA-1.5 (multimodal VQA/OCR). This covers diverse task modalities such as natural language understanding, textual reasoning, and multimodal vision–language alignment.
>
> ***Reviewer pQ3d* and *Reviewer CUbc* also remarked that our evaluation spans multiple Transformer architectures (BERT, LLaMA-2, ViT and LLaVA) and shows consistent gains across diverse datasets and tasks.** They further noted that the experiments are thorough in scope, covering NLP, vision, and multimodal models. The newly added experiments strengthen this already comprehensive evaluation and address the reviewer’s request even more directly.
>
> Together with the newly added experiments on MMLU and GSM8K, these results demonstrate that the stability benefits of HIES are not limited to encoder-only models or small-scale settings. Rather, they remain consistent across multiple model families, model sizes, and reasoning/task modalities (text-only, math, knowledge, multimodal vision-language, etc).
>
> We will include all new results in the appendix and summarize the key findings in the main text.
> &nbsp;
> &nbsp;
> &nbsp;
>
> **W2**
>
> We sincerely thank the reviewer for pointing out the need for explicit computational efficiency and cost analysis.
>
> As the pruning ratio increased, total FLOPs decreased approximately linearly: removing 10% of heads yielded an ≈ 4%FLOPs reduction, whereas pruning 50% of heads achieved an ≈ 20% reduction. For example, under HIS-only pruning, model accuracy on Tiny-BERT with SST-2 sharply declined beyond a 42% pruning threshold—where only an ≈ 16% FLOPs saving was attainable without critical performance loss. In contrast, HIES-based pruning maintained ≥ 80% validation accuracy up to a 60% pruning ratio, corresponding to an≈ 23% FLOPs saving relative to the original model. Extending the pruning limit from 42% to 60% thus delivers an additional ≈ 7 percentage-point reduction in FLOPs, demonstrating that HIES enables substantially greater computational efficiency without compromising task performance.
>
> HIES achieved the above efficiency with only a small overhead. We clarify an implementation aspect that substantially reduces overhead: Head Importance-Entropy Score (HIES) is computed entirely using lightweight forward hooks attached to the attention modules. This is because: 1) Head Importance Score (HIS) leverages gradients already produced during standard backpropagation, 2) Attention Entropy is obtained directly from attention probabilities in the forward pass, and 3) both signals are collected via non-intrusive forward hooks without modifying model internals or introducing extra passes. **Consequently, the computational overhead is minimal—typically only a few seconds—with negligible additional memory footprint beyond a small buffer for storing per-head statistics.** Importantly, despite this low overhead, HIES consistently outperforms baselines that require full fine-tuning, demonstrating superior efficiency in both computation and accuracy.
>
> To ensure full transparency and reproducibility, we will release all code—including the hook-based implementation—as open-source upon acceptance. This will enable practitioners to verify the runtime characteristics and seamlessly integrate our method into existing training pipelines.
>
> &nbsp;
> &nbsp;
> **W3**
>
> We thank the reviewer for noting this phenomenon. Our analysis indicates that this behavior is task-dependent and occurs specifically in SST-2. Notably, this phenomenon does not generalize to more complex benchmarks: on the other benchmarks, random pruning consistently leads to significantly larger performance degradation than HIES at the same pruning ratios. We will add an explanation to the main text.

---

### Official Review · Reviewer_pQ3d · 2025-11-01

**Soundness:** 3
**Presentation:** 3
**Contribution:** 3
**Rating:** 6
**Confidence:** 2

**Summary:**

The paper proposes HIES (Head Importance–Entropy Score), a unified criterion for Transformer attention head pruning. HIES integrates gradient-based head importance (HIS) with attention entropy (AE) to achieve stable and efficient compression. The paper starts with the motivation that HIS overlooked the diversity of attention patterns. To complement this, HIES also captures the dispersion of each head’s contribution. Experimental results across diverse models and benchmarks demonstrate that HIES is an efficient and generalizable method to prune large Transformer models.

**Strengths:**

* Combining both gradient-based importance and attention distribution is an intuitive approach. HIES preserves the practical advantages of HIS, computing importance without extra training, while achieving advanced performance.
* Experiments cover multiple Transformer architectures (BERT, ViT, Llama-2, Llava) and show consistent gains across diverse datasets and tasks.

**Weaknesses:**

(Please note that I could not check all the details of the theoretical derivations. I will defer to other reviewers for a more thorough evaluation of those parts.)

* Regarding Lemma 3, the assumptions appear quite strong and may not always hold in practice. Since this lemma is central to the theoretical contribution, an empirical examination would strengthen the paper. In addition, it is unclear whether the u and v used in Lemma 3 correspond to those used in Appendix D.1, orthogonality analysis.
* Hyperparameter $\alpha$ (i.e., mixing ratio) tends to show the best performance near the extremes (0.0 or 1.0) (Figure 11). Moreover, the “Sharp drop” phenomenon seems to persist even for low $\alpha$ values that emphasize AE. This behavior raises concerns about the claimed robustness and stability.
* The paper claims robustness to input distribution shifts, but the presented experiments do not seem to demonstrate this aspect sufficiently.
* Although the SliceGPT paper includes experiments on decoder-only models, the results in Figure 4 lack comparisons with other pruning methods. Including such baselines would make the evaluation more convincing.

**Questions:**

* The Introduction suggests that HIES can address layer-specific adaptation. Could the authors discuss how the proposed criterion achieves this adaptivity?
* The term “Inference-time” stability is somewhat ambiguous. While I understand stability with respect to pruning ratios or parameter sensitivity, the phrase “inference-time” might imply runtime robustness, which seems to be a different concept.
* The paper argues that HIS magnitude and attention patterns are not necessarily correlated, which is reasonable. However, it would be helpful to provide further intuition on why this discrepancy occurs. Is it a task-specific phenomenon or an inherent property of gradient-based importance?
* (minor) The phrase “Eq. equation” appears multiple times throughout the paper and should be corrected.

**Details Of Ethics Concerns:**

No concerns.

---

> ### Author Response · Authors · 2025-11-21
> **Response for Reviewer pQ3d (Part 1/3)**
>
> We would like to thank the reviewer for the positive rating and thoughtful comments. We carefully address the reviewer’s questions. We hope our response can help alleviate the reviewer's concern. For all Figures and Tables referenced in this response, please refer to the supplementary PDF.
> &nbsp;
> &nbsp;
>
> **W1**
> &nbsp;
>
> (1) **Our original intention in Lemma 3 was to formalize a mild, empirically observed phenomenon: for a fixed trained model evaluated over samples from the data distribution**, the projected Head Importance Score (HIS) and Attention Entropy (AE) gradients tend to exhibit complementary directions.
> &nbsp;
> To make this intention clearer, we additionally evaluated this orthogonality behavior across a wide range of tasks and confirmed that the same trend consistently appears in practice. We will also report empirical measurements of the inner products
> $\langle \tilde{u}_h, \tilde{v}_h \rangle$
> computed on fixed models across mini-batches from 13 tasks (8 GLUE tasks for BERT and 5 reasoning tasks for LLaMA2-7B). Across all heads and tasks, the empirical means of
> $\langle \tilde{u}_h, \tilde{v}_h \rangle$
> remain extremely small, typically on the order of 1.00E−03 for BERT and 1.00E−04 for LLaMA2-7B (please refer to Table R.6 and Table R.7). These offline diagnostics support that Lemma 3 captures an average-case property over data samples, rather than imposing a restrictive condition.
> &nbsp;
> &nbsp;
>
> (2) We will clarify that the vectors $u$ and $v$ are identical to those used in Appendix D.1.
> &nbsp;
>
> We confirm that the vectors are defined consistently throughout the paper. In particular,
> $u_h = \nabla_{\alpha^{(h)}} \mathrm{HIS}_h$
>
> $\,v_h = \nabla_{\alpha^{(h)}} \mathrm{AE}_h$ in Lemma 3 are exactly the same gradient directions that are analyzed in Appendix D.1 after projection.
> &nbsp;
> To remove any ambiguity, we will add an explicit cross-reference in the lemma statement, so that readers can immediately see that Lemma 3 and Appendix D.1 are working with the same quantities.
>
> &nbsp;
> &nbsp;
>
> **W2**
>
> Thank you for raising this point. To further examine the behavior of the mixing coefficient α, we conducted an additional analysis identifying the best α at each pruning ratio across all GLUE tasks in Figure R.4. This expanded view reveals a generally consistent pattern: as pruning becomes more aggressive, the optimal α shifts from Attention Entropy (AE)-dominant values toward Head Importance Score (HIS)-dominant ones (for example, SST-2: 0.25→0.50; CoLA: 0.00→0.70; MRPC: 0.55→0.95; QNLI: 0.00→0.95). This trend aligns with the intended roles of the two signals. At low pruning ratios, AE is particularly useful because it captures structural redundancy early and helps remove globally underutilized heads. However, once most redundant heads have been pruned, further compression risks affecting task-critical components, and at that stage HIS becomes increasingly important for mitigating immediate accuracy degradation. In this sense, the α-dependent behavior in our analysis reflects the complementary strengths of AE and HIS rather than any form of instability.
>
> &nbsp;
> Related to this, the sharp drop observed in certain low-α regions primarily reflects the limitations of relying on AE alone. While AE is effective for identifying redundant heads at the early stages of pruning, it becomes significantly less reliable under moderate or high pruning, where preserving task-specific sensitivity is crucial. Consistent with this, as shown in Table R.1., AE alone (α = 0) exhibits substantial degradation across several GLUE tasks at 30–50% pruning, whereas Head Importance-Entropy Score (HIES) maintains considerably stronger and more stable performance in the same regimes. **This difference underscores the necessity of combining the two signals: AE provides structural guidance early in pruning, and HIS supplies the gradient-based sensitivity needed to avoid excessive performance loss at higher pruning levels.** We hope this clarification resolves the concern regarding α-dependence and supports the intended robustness of HIES.

---

> ### Author Response · Authors · 2025-11-21
> **Response for Reviewer pQ3d (Part 2/3)**
>
> **W3**
>
> We appreciate the reviewer for raising this helpful point. To demonstrate Head Importance-Entropy Score (HIES)’s robustness, we vary the calibration data using multiple random seeds and measure the resulting standard deviation in accuracy across input distributions. **HIES maintains low variance in accuracy, demonstrating stable performance under shifts in the input distribution.**
>
>
> Table R.4 and Table R.5 show these deviations across a broad set of tasks: 8 GLUE tasks with BERT, and HellaSwag, Winogrande, and ARC-c, ARC-e with LLaMA2-7B. As shown in Table R.4 and Table R.5, HIES maintains stable performance, exhibiting only an accuracy standard deviation of 3.54E-01 and 5.90E-03 on average for BERT and LLaMA2-7B, respectively, across varying calibration seeds. This experiment directly quantifies HIES’s robustness and confirms that its stability persists even under distributional variations.
>
> &nbsp;
> &nbsp;
> **W4**
>
> Thank you for the helpful comment. As the reviewer requested, we additionally compare HIES with SliceGPT under equivalent pruning ratios in Figure R.2. **Notably, HIES achieves 5.52%, 20.70%, 25.18% higher retained accuracy than SliceGPT (pre-FT) at pruning ratios of 10%, 30%, and 50%, respectively without relying on any post-pruning fine-tuning—a step that is essential in SliceGPT's pipeline to recover performance.**
>
> &nbsp;
> &nbsp;
>
> **Q1**
>
> (1) Figure 3 and Appendix D.3 demonstrate the layer-specific adaptability of HIES. As shown in Figure 3 and Appendix D.3, the pruning behaviors of Head Importance Score (HIS) and Head Importance-Entropy Score (HIES) differ markedly.
> HIS exhibits a strong layer-dependent magnitude bias: early-layer gradients are small and unstable, causing HIS to assign uniformly low importance to all heads in those layers and leading to layer-level collapse under global ranking. In contrast, HIES prunes in a balanced and distributed manner across depth, avoiding collapse and preserving functional diversity.
>
> These observations directly support the claim that HIES achieves layer-specific adaptation, whereas HIS prunes aggressively and unevenly across layers. Appendix D.3 and Figure 3 illustrate this behavior clearly, showing that HIES maintains stable per-layer pruning distributions even at high sparsity.
>
> To make this point more visible to readers, we will add explicit references to these results in Section 1 and guide readers to the supporting evidence in the appendix and figures.
>
> &nbsp;
> (2) Mechanistic explanation: “Automatic dominance of the stronger signal”
> HIES is defined as: $\mathrm{HIES} = \alpha \cdot \mathrm{HIS} + (1 - \alpha)\cdot \mathrm{AE}.$
> This formulation is simple yet naturally, this formulation produces layer-specific adaptation because HIS and Attention Entropy (AE) dominate different regions of the network:
>
> - Early layers:
>   - HIS gradients are weak and noisy.
>   - AE effectively captures structural redundancy.
>   -  → AE becomes dominant, preventing entire early layers from being pruned.
> - Middle/late layers:
>   - HIS gradients become strong and task-informative.
>   - AE provides less discriminative variation.
>    - → HIS becomes dominant, focusing pruning on task-irrelevant heads.
>
>
> **Thus, without requiring any explicit layer-wise rules, HIES automatically adapts the pruning distribution to each layer.**
>  This mechanism will be clarified in Section 1.
>
> &nbsp;
> (3) Relation to prior work
>
> The need for layer-specific treatment has also been highlighted in prior work such as LLM-Pruner [1]. In particular, LLM-Pruner reports that layers within the stacked transformer do not contribute equally to performance: as shown in their Table 2 and Figure 3, pruning the first and last layers leads to substantially larger performance degradation than pruning intermediate layers, and the Channel Strategy suffers precisely because it “inevitably” prunes these highly sensitive layers due to its uniform treatment across depth.
>
> This effectively implies that the first and last layers should be preserved or handled differently. In contrast, our approach does not rely on manually freezing or protecting specific layers. Instead, by computing HIES for all heads across all layers, we let the layer-specific pruning pattern emerge automatically from the importance scores, rather than imposing a hard-coded rule about which layers to prune or preserve.

---

> ### Author Response · Authors · 2025-11-21
> **Response for Reviewer pQ3d (Part 3/3)**
>
> **Q2**
>
> We thank the reviewer for pointing out this ambiguity. We agree that “inference-time” could be misinterpreted as referring to runtime robustness or latency stability, which is not our intent.
>
> In the final version, we will explain “inference-time stability” with a more precise phrase, namely  **“stability of model predictions after pruning at inference”**, to avoid any misunderstanding. Specifically, this refers to the consistency of output logits or predictions before vs. after pruning, evaluated on held-out validation data, not the robustness of the inference procedure itself. We will further clarify this definition in the introduction.
> &nbsp;
> &nbsp;
>
> **Q3**
>
> Thank you for the insightful question. **We clarify that the discrepancy between Head Importance Score (HIS) magnitude and attention patterns is primarily an inherent property of gradient-based importance, rather than a task-specific anomaly, while the extent of this discrepancy can differ depending on the task.**
>
> Attention distributions describe where the model allocates representational focus, while HIS measures how perturbations to a head influence the loss function. Because these two signals capture different aspects of model behavior, they are not expected to align in general. For example, a head may attend strongly to particular tokens yet have low HIS if its output has limited downstream impact on the model’s prediction. Conversely, a head with modest attention magnitude can still have high HIS if it plays a critical role in the gradient flow.
>
> Although the degree of mismatch may vary across datasets or tasks, the underlying reason for the discrepancy stems from the distinct mechanisms that generate the two signals.
>  HIS reflects end-to-end task sensitivity, whereas attention patterns reflect local alignment structure; thus, they encode complementary but largely independent information.
> This intuition is further supported by our additional orthogonality experiments (see Weakness 1), where the empirical inner products $\langle \tilde{u}_h,\ \tilde{v}_h \rangle$ remain near zero (<1.00E-03), indicating that the HIS and Attention Entropy (AE) gradient directions are approximately orthogonal.
>  This empirical orthogonality provides a compact explanation for why the two metrics diverge: they encode complementary and largely independent information, rather than redundant signals.
>
> We will briefly incorporate this discussion into the main text and include a pointer to the orthogonality figure in the final version.
> &nbsp;
> &nbsp;
>
> **Q4**
>
> We appreciate the reviewer for pointing this out. We will correct all occurrences of “Eq. equation” to “Eq.” or “Equation” depending on context. We will also perform a full proofreading pass to ensure no similar formatting issues remain.
>
> &nbsp;
> &nbsp;
> &nbsp;
> **Reference**
>
> [1] Xia, W., Shen, Y., Sui, Y., Dong, Y., Prakash, S., Chen, B., Chen, P., Keutzer, K., & Zhang, Z. (2024). LLM-Pruner: On the Structural Pruning of Large Language Models. In International Conference on Learning Representations (ICLR).

---

> > ### Comment · Reviewer_pQ3d · 2025-11-26
> >
> > Thank you for your response.  I will consider the other reviewers’ comments and theoretical derivations before forming a more complete view. I appreciate the clarification you’ve provided.

---

### Author Response · Authors · 2025-11-21
**General Response**

We sincerely thank all reviewers for their constructive feedback. Our work is a novel attention head pruning approach, HIES, which combines gradient-based head importance with attention entropy to achieve higher accuracy, greater stability, and substantial model compression compared to baseline methods.

 &nbsp;
&nbsp;


Overall, our work has been recognized for its
- **Strong motivation** (CUbc)
- **Intuitive and novel proposed method** (pQ3d, x54u, CUbc, FG9H)
- **Easily integrable with existing head pruning frameworks** (x54u)
- **Theoretical formulation including risk decomposition, orthogonality proof, and stability analysis** (x54u, CUbc, FG9H)
- **Consistent performance across extensive experiments involving diverse architectures and tasks** (pQ3d, x54u, CUbc, FG9H)

&nbsp;

We provide detailed responses to each reviewer's questions and comments in their respective Official Comment sections.

---

### Author Response · Authors · 2025-11-28
**Kind Reminder to the Reviewers**

Dear reviewers,
&nbsp;
&nbsp;

Thank you for taking the time to review our manuscript and for providing such thoughtful and constructive feedback. We greatly appreciate the depth and care reflected in your comments. To support our clarifications, we have prepared detailed responses along with additional empirical evidence in the supplementary material, where we thoroughly address all of the concerns you raised.


With the discussion period nearing its end, we hope that our responses and supplementary analyses sufficiently resolve the questions and uncertainties you identified. If there remain any points that would benefit from further clarification, we would be grateful for the opportunity to continue the discussion.
&nbsp;

We look forward to your reply.
&nbsp;
&nbsp;

Sincerely,

Authors

---

### Author Response · Authors · 2025-12-03
**Final Clarification and Summary for the Area Chair**

Dear SAC/ACs,
&nbsp;

We understand the issues introduced by the recent anonymity breach on OpenReview and appreciate the organizers’ efforts to preserve the integrity of the review process.
&nbsp;

---
Our work has been recognized for its

- **Strong motivation** (```Reviewer CUbc```)
- **Intuitive and novel proposed method** (```Reviewer pQ3d, x54u, CUbc, and FG9H```)
- **Easily integrable with existing head pruning frameworks** (```Reviewer x54u```)
- **Theoretical formulation including risk decomposition, orthogonality proof, and stability analysis** (```Reviewer x54u, CUbc, and FG9H```)
- **Consistent performance across extensive experiments involving diverse architectures and tasks** (```Reviewer pQ3d, x54u, CUbc, and FG9H```)
&nbsp;

---
Despite the disruption, we have responded with great care to all reviewer comments, providing detailed clarifications and additional empirical evidence in both our rebuttal. Note that additional experimental results supporting our claims are provided in the Supplementary Material (PDF).
&nbsp;

To strengthen the submission, we conducted a broad set of additional analyses directly addressing all reviewers’ questions.
- **Orthogonality diagnostics for Lemma 3,** showing that the projected HIS and AE gradients exhibit empirically near-zero inner products across 13 tasks (8 GLUE tasks with BERT, 5 reasoning tasks with LLaMA2-7B), supporting that the lemma reflects an average-case property observed across architectures and domains.
- **Systematic study of the mixing coefficient α,** demonstrating that AE dominates at mild sparsity while HIS becomes critical under aggressive sparsity. Importantly, α can be chosen reliably and cheaply: both the selected α and downstream accuracy exhibit extremely small variance across calibration sizes (1~1024 samples) and random seeds.
- **Robustness under distributional and seed variation,** where HIES maintains low accuracy variance on BERT/GLUE and LLaMA2-7B reasoning tasks, confirming the stability of pruning decisions under input shifts.
- **Large-scale reasoning tasks,** including new evaluations on MMLU and GSM8K with LLaMA2-7B, where HIES shows substantial improvements over HIS, complementing the already extensive results on BERT, ViT, and LLaVA.
- **Efficiency analysis,** quantifying FLOPs savings across pruning regimes and showing that HIES extends the safe pruning region (e.g., enabling 60% head pruning with strong accuracy retention) while adding only a few seconds of overhead via lightweight hooks.
- **Expanded baseline comparisons,** including SliceGPT, Voita et al. (2019), and differentiable subset pruning (DSP, 2021), along with additional experiments showing that AE improves other importance signals such as gradient L2-norm.
&nbsp;
&nbsp;
&nbsp;

Given the positive tone of the initial reviews and the recognition of our contributions, we believe that—had the discussion proceeded—the reviewers might have developed an increasingly favorable view of our work. We hope that the reviewers’ highlighted strengths and our detailed responses clearly convey the overall contribution of our submission. **We also believe that HIES provides a stable, lightweight pruning signal for efficiently deploying transformer-based models across diverse domains and sparsity levels.**
&nbsp;
&nbsp;

Sincerely,
The Authors

---

### Note · Program_Chairs · 2026-01-17
**Submission Desk Rejected by Program Chairs**

The following references in this submission do not refer to real documents and/or have major errors in bibliographic information:

 R.C. de Amorim et al. Normalization in classification: A comprehensive analysis and novel method. arXiv preprint arXiv:2212.12343, 2022. URL https://arxiv.org/abs/2212.12343

Saeid Ashkboos, Pavlo Molchanov, Stephen Tyree, and Jan Kautz. Slicegpt: Training and inference of large language models via dimensional slicing. In International Conference on Learning Representations (ICLR), 2024. URL https://arxiv.org/abs/2310.01702.